# Multi-annotator Deep Learning:
# A Probabilistic Framework for Classification

**Marek Herde**                                                 *marek.herde@uni-kassel.de*
*Intelligent Embedded Systems*
*University of Kassel*
*Kassel, Hesse, Germany*

**Denis Huseljic**                                              *dhuseljic@uni-kassel.de*
*Intelligent Embedded Systems*
*University of Kassel*
*Kassel, Hesse, Germany*

**Bernhard Sick**                                               *bsick@uni-kassel.de*
*Intelligent Embedded Systems*
*University of Kassel*
*Kassel, Hesse, Germany*

**Reviewed on OpenReview:** *https://openreview.net/forum?id=MgdoxzImlK*

## Abstract

Solving complex classification tasks using deep neural networks typically requires large amounts of annotated data. However, corresponding class labels are noisy when provided by error-prone annotators, e.g., crowdworkers. Training standard deep neural networks leads to subpar performances in such multi-annotator supervised learning settings. We address this issue by presenting a probabilistic training framework named multi-annotator deep learning (MaDL). A downstream ground truth and an annotator performance model are jointly trained in an end-to-end learning approach. The ground truth model learns to predict instances' true class labels, while the annotator performance model infers probabilistic estimates of annotators' performances. A modular network architecture enables us to make varying assumptions regarding annotators' performances, e.g., an optional class or instance dependency. Further, we learn annotator embeddings to estimate annotators' densities within a latent space as proxies of their potentially correlated annotations. Together with a weighted loss function, we improve the learning from correlated annotation patterns. In a comprehensive evaluation, we examine three research questions about multi-annotator supervised learning. Our findings show MaDL's state-of-the-art performance and robustness against many correlated, spamming annotators.

## 1 Introduction

Supervised *deep neural networks* (DNNs) have achieved great success in many classification tasks (Pouyanfar et al., 2018). In general, these DNNs require a vast amount of annotated data for their successful employment (Algan & Ulusoy, 2021). However, acquiring top-quality class labels as annotations is time-intensive and/or financially expensive (Herde et al., 2021). Moreover, the overall annotation load may exceed a single annotator's workforce (Uma et al., 2021). For these reasons, multiple non-expert annotators, e.g., crowdworkers, are often tasked with data annotation (Zhang, 2022; Gilyazev & Turdakov, 2018). Annotators' missing domain expertise can lead to erroneous annotations, known as noisy labels. Further, even expert annotators cannot be assumed to be omniscient because additional factors, such as missing motivation, fatigue, or an annotation task's ambiguity (Vaughan, 2018), may decrease their performances. A popular annotation

quality assurance option is the acquisition of multiple annotations per data instance with subsequent aggregation (Zhang et al., 2016), e.g., via majority rule. The aggregated annotations are proxies of the *ground truth* (GT) labels to train DNNs. Aggregation techniques operate exclusively on the basis of annotations. In contrast, model-based techniques use feature or annotator information and thus work well in low-redundancy settings, e.g., with just one annotation per instance (Khetan et al., 2018). Through predictive models, these techniques jointly estimate instances' GT labels and *annotators' performances* (APs) by learning and inferring interdependencies between instances, annotators, and their annotations. As a result, model-based techniques cannot only predict GT labels and APs for training instances but also for test instances, i.e., they can be applied in transductive and inductive learning settings (Vapnik, 1995).

Despite ongoing research, several **challenges** still need to be addressed in multi-annotator supervised learning. To introduce these challenges, we exemplarily look at the task of animal classification in Fig. 1. Eight annotators have been queried to provide annotations for the image of a jaguar. Such a query is difficult because jaguars have remarkable similarities to other predatory cats, e.g., leopards. Accordingly, the obtained annotations indicate a strong disagreement between the leopard and jaguar class. Simply taking the majority vote of these annotations results in leopard as a wrongly estimated GT label. Therefore, advanced multi-annotator supervised learning techniques leverage annotation information from other (similar) annotated images to estimate APs. However, producing accurate AP estimates is difficult because one needs to learn many annotation patterns. Otherwise, the estimated GT labels will be biased, e.g., when APs are exclusively modeled as a function of annotators. In this case, we cannot identify annotators who are only knowledgeable about certain classes or regions in the feature space. Another challenge in multi-annotator supervised learning concerns potential (latent) correlations between annotators. In our animal annotation task, we illustrate this issue by visualizing three latent groups of similarly behaving annotators. Although we assume that the annotators work independently of each other, they can still share common or statistically correlated error patterns (Chu et al., 2021). This is particularly problematic if a group of ordinary persons strongly outvotes a much smaller group of professionals. Considering prior information about the annotators, i.e., annotator features or metadata (Zhang et al., 2023), can help to identify these groups. Moreover, prior information enables a model to inductively learn performances for annotators who have provided few or no annotations. In our example, zoological interest could be a good indicator for this purpose. While the inductive learning of APs for annotators poses an additional challenge to the already complex task, its use may be beneficial for further applications, e.g., optimizing the annotator selection in an active learning setting (Herde et al., 2021) or training annotators to improve their own knowledge (Daniel et al., 2018).

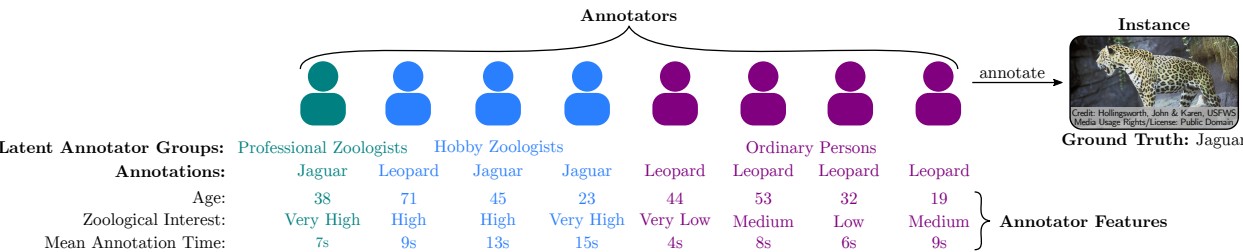

Figure 1: Animal classification as an illustration of a multi-annotator supervised learning problem.

In this article, we address the above challenges by making the following **contributions**:

- We propose *multi-annotator deep learning* (MaDL) as a probabilistic and modular classification framework. In an end-to-end training via a weighted maximum-likelihood approach, it learns embeddings of annotators to account for possible correlations among them.

- We specify six properties concerning the estimation of APs and application scenarios for categorizing related multi-annotator supervised learning techniques.

- Associated with these properties, we formulate three *research questions* (RQs), which we experimentally investigate, including comparisons of MaDL to related techniques.

The **remainder of this article** is structured as follows: In Section 2, we formally introduce the problem setting of supervised learning from multiple annotators. Subsequently, we identify central properties of multi-annotator supervised learning techniques as a basis for categorizing related works and pointing out their differences to MaDL in Section 3. Section 4 explains the details of our MaDL framework. Experimental evaluations of MaDL and related techniques are presented regarding RQs associated with the aforementioned properties in Section 5. Finally, we conclude and give an outlook regarding future research work in Section 6.

## 2 Problem Setting

In this section, we formalize the assumptions and objectives of multi-annotator supervised learning for classification tasks.

**Prerequisites:** Without loss of generality, we represent a data instance as a vector $\mathbf{x} := (x^{(1)}, ..., x^{(D)})^{\mathrm{T}}$, $D \in \mathbb{N}_{>0}$ in a $D$-dimensional real-valued input or feature space $\Omega_X := \mathbb{R}^D$. The $N \in \mathbb{N}_{>0}$ instances jointly form a matrix $\mathbf{X} := (\mathbf{x}_1, ..., \mathbf{x}_N)^{\mathrm{T}}$ and originate from an unknown probability density function $\mathrm{Pr}(\mathbf{x})$. For each observed instance $\mathbf{x}_n \sim \mathrm{Pr}(\mathbf{x})$, there is a GT class label $y_n \in \Omega_Y := \{1, ..., C\}$. Each GT label $y_n$ is assumed to be drawn from an unknown conditional distribution: $y_n \sim \mathrm{Pr}(y \mid \mathbf{x}_n)$. We denote the GT labels as the vector $\mathbf{y} := (y_1, ..., y_N)^{\mathrm{T}}$. These GT labels are unobserved since there is no omniscient annotator. Instead, we consider multiple error-prone annotators. For the sake of simplicity, we represent an annotator through individual features as a vector $\mathbf{a}_m \in \Omega_A := \mathbb{R}^O, O \in \mathbb{N}_{>0}$. If no prior annotator information is available, the annotators' features are defined through one-hot encoded vectors, i.e., $\Omega_A := \{\mathbf{e}_1, ..., \mathbf{e}_M\}$ with $\mathbf{a}_m := \mathbf{e}_m$, to identify each annotator uniquely. Otherwise, annotator features may provide information specific to the general annotation task, e.g., the zoological interest when annotating animal images or the years of experience in clinical practice when annotating medical data. Together, the $M \in \mathbb{N}_{>0}$ annotators form a matrix $\mathbf{A} := (\mathbf{a}_1, ..., \mathbf{a}_M)^{\mathrm{T}}$. We denote the annotation assigned by annotator $\mathbf{a}_m$ to instance $\mathbf{x}_n$ through $z_{nm} \in \Omega_Y \cup \{\otimes\}$, where $z_{nm} = \otimes$ indicates that an annotation is unobserved, i.e., not provided. An observed annotation is assumed to be drawn from an unknown conditional distribution: $z_{nm} \sim \mathrm{Pr}(z \mid \mathbf{x}_n, \mathbf{a}_m, y)$. Multiple annotations for an instance $\mathbf{x}_n$ can be summarized as a vector $\mathbf{z}_n := (z_{n1}, ..., z_{nM})^{\mathrm{T}}$. Thereby, the set $\mathcal{A}_n := \{m \mid m \in \{1, ..., M\} \wedge z_{nm} \in \Omega_Y\}$ represents the indices of the annotators who assigned an annotation to an instance $\mathbf{x}_n$. Together, the annotations of all observed instances form the matrix $\mathbf{Z} := (\mathbf{z}_1, ..., \mathbf{z}_N)^{\mathrm{T}}$. We further assume there is a subset of annotators whose annotated instances are sufficient to approximate the GT label distribution, i.e., together, these annotated instances allow us to correctly differentiate between all classes. Otherwise, supervised learning is hardly possible without explicit prior knowledge about the distributions of GT labels and/or APs. Moreover, we expect that the annotators independently decide on instances' annotations and that their APs are time-invariant.

**Objectives:** Given these prerequisites, the first objective is to train a downstream GT model, which approximates the optimal GT decision function $y_{\mathrm{GT}} : \Omega_X \to \Omega_Y$ by minimizing the expected loss across all classes:

$$y_{\mathrm{GT}}(\mathbf{x}) := \underset{y' \in \Omega_Y}{\arg\min} \left( \mathbb{E}_{y|\mathbf{x}} \left[ L_{\mathrm{GT}}(y, y') \right] \right). \tag{1}$$

Thereby, we define the loss function $L_{\mathrm{GT}} : \Omega_Y \times \Omega_Y \to \{0, 1\}$ through the zero-one loss:

$$L_{\mathrm{GT}}(y, y') := \delta(y \neq y') := \begin{cases} 0, & \text{if } y = y', \\ 1, & \text{if } y \neq y'. \end{cases} \tag{2}$$

As a result, an optimal GT model for classification tasks can accurately predict the GT labels of instances.

**Proposition 1.** *Assuming $L_{\mathrm{GT}}$ to be the zero-one loss in Eq. 2, the Bayes optimal prediction for Eq. 1 is given by:*

$$y_{\mathrm{GT}}(\mathbf{x}) = \underset{y' \in \Omega_Y}{\arg\max} \left( \mathrm{Pr}(y' \mid \mathbf{x}) \right). \tag{3}$$

When learning from multiple annotators, the APs are further quantities of interest. Therefore, the second objective is to train an AP model, which approximates the optimal AP decision function

$y_{\mathrm{AP}} : \Omega_X \times \Omega_A \to \{0, 1\}$ by minimizing the following expected loss:

$$y_{\mathrm{AP}}(\mathbf{x}, \mathbf{a}) \coloneqq \operatorname*{arg\,min}_{y' \in \{0,1\}} \left( \mathbb{E}_{y|\mathbf{x}} \left[ \mathbb{E}_{z|\mathbf{x},\mathbf{a},y} \left[ L_{\mathrm{AP}} \left( y', L_{\mathrm{GT}} \left( y, z \right) \right) \right] \right] \right). \tag{4}$$

Defining $L_{\mathrm{AP}}$ and $L_{\mathrm{GT}}$ as zero-one loss, an optimal AP model for classification tasks can accurately predict the zero-one loss of annotator's class labels, i.e., whether an annotator $\mathbf{a}$ provides a false, i.e., $y_{\mathrm{AP}}(\mathbf{x}, \mathbf{a}) = 1$, or correct, i.e., $y_{\mathrm{AP}}(\mathbf{x}, \mathbf{a}) = 0$, class label for an instance $\mathbf{x}$.

**Proposition 2.** *Assuming both $L_{\mathrm{AP}}$ and $L_{\mathrm{GT}}$ to be the zero-one loss, as defined in Eq. 2, the Bayes optimal prediction for Eq. 4 is given by:*

$$y_{\mathrm{AP}}(\mathbf{x}, \mathbf{a}) = \delta \left( \sum_{y \in \Omega_Y} \Pr(y \mid \mathbf{x}) \Pr(y \mid \mathbf{x}, \mathbf{a}, y) < 0.5 \right). \tag{5}$$

We refer to Appendix A for the proofs of Proposition 1 and Proposition 2.

## 3 Related Work

This section discusses existing multi-annotator supervised learning techniques targeting our problem setting of Section 2. Since we focus on the AP next to the GT estimation, we restrict our discussion to techniques capable of estimating both target types. In this context, we analyze related research regarding three aspects, i.e., GT models, AP models, and algorithms for training these models.

**Ground truth model:** The first multi-annotator supervised learning techniques employed logistic regression models (Raykar et al., 2010; Kajino et al., 2012; Rodrigues et al., 2013; Yan et al., 2014) for classification. Later, different kernel-based variants of GT models, e.g., Gaussian processes, were developed (Rodrigues et al., 2014; Long et al., 2016; Gil-Gonzalez et al., 2021). Rodrigues et al. (2017) focused on documents and extended topic models to the multi-annotator setting. More recently, several techniques were proposed to train DNNs for large-scale and especially image classification tasks with noisy annotations (Albarqouni et al., 2016; Guan et al., 2018; Khetan et al., 2018; Rodrigues & Pereira, 2018; Yang et al., 2018; Tanno et al., 2019; Cao et al., 2019; Platanios et al., 2020; Zhang et al., 2020; Gil-González et al., 2021; Rühling Cachay et al., 2021; Chu et al., 2021; Li et al., 2022; Wei et al., 2022; Gao et al., 2022). MaDL follows this line of work and also employs a (D)NN as the GT model.

**Annotator performance model:** An AP model is typically seen as an auxiliary part of the GT model since it provides AP estimates for increasing the GT model's performance. In this article, we reframe an AP model's use in a more general context because accurately assessing APs can be crucial in improving several applications, e.g., human-in-the-loop processes (Herde et al., 2021) or knowledge tracing (Piech et al., 2015). For this reason, we analyze existing AP models regarding six properties, which we identified as relevant while reviewing literature about multi-annotator supervised learning.

**(P1) Class-dependent annotator performance:** The simplest AP representation is an overall accuracy value per annotator. On the one hand, AP models estimating such accuracy values have low complexity and thus do not overfit (Rodrigues et al., 2013; Long et al., 2016). On the other hand, they may be overly general and cannot assess APs on more granular levels. Therefore, many other AP models assume a dependency between APs and instances' GT labels. Class-dependent AP models typically estimate confusion matrices (Raykar et al., 2010; Rodrigues et al., 2014; 2017; Khetan et al., 2018; Tanno et al., 2019; Platanios et al., 2020; Gao et al., 2022; Li et al., 2022), which indicate annotator-specific probabilities of mistaking one class for another, e.g., recognizing a jaguar as a leopard. Alternatively, weights of annotation aggregation functions (Cao et al., 2019; Rühling Cachay et al., 2021) or noise-adaption layers (Rodrigues & Pereira, 2018; Chu et al., 2021; Wei et al., 2022) can be interpreted as non-probabilistic versions of confusion matrices. MaDL estimates probabilistic confusion matrices or less complex approximations, e.g., the elements on their diagonals.

**(P2) Instance-dependent annotator performance:** In many real-world applications, APs are additionally instance-dependent (Yan et al., 2014) because instances of the same class can strongly vary in their feature values. For example, recognizing animals in blurry images is more difficult than in high-resolution images. Hence, several AP models estimate the probability of obtaining a correct annotation as a function of instances and annotators (Kajino et al., 2012; Yan et al., 2014; Guan et al., 2018; Yang et al., 2018; Gil-Gonzalez et al., 2021; Gil-González et al., 2021). Combining instance- and class-dependent APs results in the most complex AP models, which estimate a confusion matrix per instance-annotator pair (Platanios et al., 2020; Zhang et al., 2020; Rühling Cachay et al., 2021; Chu et al., 2021; Gao et al., 2022; Li et al., 2022). MaDL also employs an AP model of this type. However, it optionally allows dropping the instance and class dependency, which can benefit classification tasks where each annotator provides only a few annotations.

**(P3) Annotator correlations:** Although most techniques assume that annotators do not collaborate, they can still have correlations regarding their annotation patterns, e.g., by sharing statistically correlated error patterns (Chu et al., 2021). Gil-Gonzalez et al. (2021) proposed a kernel-based approach where a matrix quantifies such correlations for all pairs of annotators. Inspired by weak supervision, Cao et al. (2019) and Rühling Cachay et al. (2021) employ an aggregation function that takes all annotations per instance as input to model annotator correlations. Gil-González et al. (2021) introduce a regularized chained DNN whose weights encode correlations. Wei et al. (2022) jointly model the annotations of all annotators as outputs and thus take account of potential correlated mistakes. Chu et al. (2021) consider common annotation noise through a noise adaptation layer shared across annotators. Moreover, similar to our MaDL framework, they learn embeddings of annotators. Going beyond, MaDL exploits these embeddings to determine annotator correlations.

**(P4) Robustness to spamming annotators:** Especially on crowdsourcing platforms, there have been several reports of workers spamming annotations (Vuurens et al., 2011), e.g., by randomly guessing or permanently providing the same annotation. Such spamming annotators can strongly harm the learning process. As a result, multi-annotator supervised learning techniques are ideally robust against these types of annotation noise. Cao et al. (2019) employ an information-theoretic approach to separate expert annotators from possibly correlated spamming annotators. Rühling Cachay et al. (2021) empirically demonstrated that their weak-supervised learning technique is robust to large numbers of randomly guessing annotators. MaDL ensures this robustness by training via a weighted likelihood function, assigning high weights to independent annotators whose annotation patterns have no or only slight statistical correlations to the patterns of other annotators.

**(P5) Prior annotator information:** On crowdsourcing platforms, requesters may acquire prior information about annotators (Daniel et al., 2018), e.g., through surveys, annotation quality tests, or publicly available profiles. Several existing AP models leverage such information to improve learning. Thereby, conjugate prior probability distributions, e.g., Dirichlet distributions, represent a straightforward way of including prior estimates of class-dependent accuracies (Raykar et al., 2010; Albarqouni et al., 2016; Rodrigues et al., 2017). Other techniques (Platanios et al., 2020; Chu et al., 2021), including our MaDL framework, do not directly expect prior accuracy estimates but work with all types of prior information that can be represented as vectors of annotator features.

**(P6) Inductive learning of annotator performance:** Accurate AP estimates can be beneficial in various applications, e.g., guiding an active learning strategy to select accurate annotators (Yang et al., 2018). For this purpose, it is necessary that a multi-annotator supervised learning technique can inductively infer APs for non-annotated instances. Moreover, an annotation process is often a dynamic system where annotators leave and enter. Hence, it is highly interesting to inductively estimate the performances of newly entered annotators, e.g., through annotator features as used by Platanios et al. 2020 and MaDL.

**Training:** Several multi-annotator supervised learning techniques employ the *expectation-maximization* (EM) algorithm for training (Raykar et al., 2010; Rodrigues et al., 2013; Yan et al., 2014; Long et al., 2016; Albarqouni et al., 2016; Guan et al., 2018; Khetan et al., 2018; Yang et al., 2018; Platanios et al., 2020).

GT labels are modeled as latent variables and estimated during the E step, while the GT and AP models' parameters are optimized during the M step. The exact optimization in the M step depends on the underlying models. Typically, a variant of *gradient descent* (GD), e.g., quasi-Newton methods, is employed, or a closed-form solution exists, e.g., for AP models with instance-independent AP estimates. Other techniques take a Bayesian view of the models' parameters and therefore resort to *expectation propagation* (EP) (Rodrigues et al., 2014; Long et al., 2016) or *variational inference* (VI) (Rodrigues et al., 2017). As approximate inference methods are computationally expensive and may lead to suboptimal results, several end-to-end training algorithms have been proposed. Gil-Gonzalez et al. (2021) introduced a localized kernel alignment-based relevance analysis that optimizes via GD. Through a regularization term, penalizing differences between GT and AP model parameters, Kajino et al. formulated a convex loss function for logistic regression models. Rodrigues & Pereira (2018), Gil-González et al. (2021), and Wei et al. (2022) jointly train the GT and AP models by combining them into a single DNN with noise adaption layers. Chu et al. (2021) follow a similar approach with two types of noise adaption layers: one shared across annotators and one individual for each annotator. Gil-González et al. (2021) employ a regularized chained DNN to estimate GT labels and AP performances jointly. In favor of probabilistic AP estimates, Tanno et al. (2019), Zhang et al. (2020), Li et al. (2022), and MaDL avoid noise adaption layers but employ loss functions suited for end-to-end learning. Cao et al. (2019) and Rühling Cachay et al. (2021) jointly learn an aggregation function in combination with the AP and GT models.

Table 1 summarizes and completes the aforementioned discussion by categorizing multi-annotator supervised learning techniques according to their GT model, AP model, and training algorithm. Thereby, the AP model is characterized by the six previously discussed properties (P1–P6). We assign ✓ if a property is supported, ✗ if not supported, and ◆ if partially supported. More precisely, ◆ is assigned to property P5 if the technique can include prior annotator information but needs a few adjustments and to property P6 if the technique requires some architectural changes to learn the performances of new annotators inductively. For property P4, a ✓ indicates that the authors have shown that their proposed technique learns in the presence of many spamming annotators.

Table 1: Literature categorization of multi-annotator supervised learning techniques.

| Reference | Ground Truth Model | Training | Annotator Performance Model | | | | | |
|---|---|---|---|---|---|---|---|---|
| | | | P1 | P2 | P3 | P4 | P5 | P6 |
| Kajino et al. (2012) | Logistic Regression Model | GD | ✗ | ✓ | ✗ | ✗ | ✗ | ✗ |
| Raykar et al. (2010) | | EM & GD | ✓ | ✗ | ✗ | ✗ | ✓ | ✗ |
| Rodrigues et al. (2013) | | | ✗ | ✗ | ✗ | ✗ | ✗ | ✗ |
| Yan et al. (2014) | | | ✗ | ✓ | ✗ | ✗ | ◆ | ◆ |
| Rodrigues et al. (2017) | Topic Model | VI & GD | ✓ | ✗ | ✗ | ✗ | ✓ | ✗ |
| Rodrigues et al. (2014) | Kernel-based Model | EP | ✓ | ✗ | ✗ | ✗ | ✗ | ✗ |
| Long et al. (2016) | | EM & EP & GD | ✗ | ✗ | ✗ | ✗ | ✗ | ✗ |
| Gil-Gonzalez et al. (2021) | | GD | ✗ | ✓ | ✓ | ✗ | ✗ | ✗ |
| Albarqouni et al. (2016) | (Deep) Neural Network | EM & GD | ✓ | ✗ | ✗ | ✗ | ✓ | ✗ |
| Yang et al. (2018) | | | ✗ | ✓ | ✗ | ✗ | ◆ | ◆ |
| Khetan et al. (2018) | | | ✓ | ✗ | ✗ | ✗ | ◆ | ◆ |
| Platanios et al. (2020) | | | ✓ | ✓ | ✗ | ✗ | ✓ | ✓ |
| Rodrigues & Pereira (2018) | | GD | ✓ | ✗ | ✗ | ✗ | ✗ | ✗ |
| Guan et al. (2018) | | | ✗ | ✓ | ✗ | ✗ | ✗ | ✗ |
| Tanno et al. (2019) | | | ✓ | ✗ | ✗ | ✗ | ◆ | ◆ |
| Cao et al. (2019) | | | ✓ | ✗ | ✓ | ✓ | ✗ | ✗ |
| Zhang et al. (2020) | | | ✓ | ✓ | ✓ | ✗ | ◆ | ◆ |
| Gil-González et al. (2021) | | | ✗ | ✓ | ✓ | ✓ | ✗ | ✗ |
| Rühling Cachay et al. (2021) | | | ✓ | ✓ | ✓ | ✓ | ✗ | ✗ |
| Chu et al. (2021) | | | ✓ | ✓ | ✓ | ✓ | ✓ | ✗ |
| Li et al. (2022) | | | ✓ | ✓ | ✗ | ✗ | ◆ | ◆ |
| Wei et al. (2022) | | | ✓ | ✗ | ✓ | ✗ | ✗ | ✗ |
| Gao et al. (2022) | | | ✓ | ✓ | ✗ | ✗ | ◆ | ◆ |
| MaDL (2023) | | | ✓ | ✓ | ✓ | ✓ | ✓ | ✓ |

# 4 Multi-annotator Deep Learning

In this section, we present our modular probabilistic MaDL framework. We start with a description of its underlying probabilistic model. Subsequently, we introduce its GT and AP models' architectures. Finally, we explain our end-to-end training algorithm.

## 4.1 Probabilistic Model

The four nodes in Fig. 2 depict the random variables of an instance $\mathbf{x}$, a GT label $y$, an annotator $\mathbf{a}$, and an annotation $z$. Thereby, arrows indicate probabilistic dependencies among each other. The random variable of an instance $\mathbf{x}$ and an annotator $\mathbf{a}$ have no incoming arrows and thus no causal dependencies on other random variables. In contrast, the distribution of a latent GT label $y$ depends on its associated instance $\mathbf{x}$. For classification problems, the probability of observing $y = c$ as GT label of an instance $\mathbf{x}$ can be modeled through a categorical distribution:

$$\Pr(y = c \mid \mathbf{x}) \coloneqq \mathrm{Cat}(y = c \mid \boldsymbol{p}(\mathbf{x})) \coloneqq \prod_{k=1}^{C} \left( p^{(k)}(\mathbf{x}) \right)^{\delta(k=c)} = p^{(c)}(\mathbf{x}), \tag{6}$$

where $\mathbf{p} : \Omega_X \to \Delta \coloneqq \{\mathbf{p} \in [0,1]^C \mid \sum_{c=1}^{C} p^{(c)} = 1\}$ denotes the function outputting an instance's true class-membership probabilities. The outcome of an annotation process may depend on the annotator's features, an instance's features, and the latent GT label. A function $\mathbf{P} : \Omega_X \times \Omega_A \to [0,1]^{C \times C}$ outputting a row-wise normalized confusion matrix per instance-annotator pair can capture these dependencies. The probability that an annotator $\mathbf{a}$ annotates an instance $\mathbf{x}$ of class $y = c$ with the annotation $z = k$ can then be modeled through a categorical distribution:

$$\Pr(z = k \mid \mathbf{x}, \mathbf{a}, y = c) \coloneqq \mathrm{Cat}\left(z = k \,\middle|\, \mathbf{P}^{(c,:)}(\mathbf{x}, \mathbf{a})\right) \coloneqq \prod_{l=1}^{C} \left( P^{(c,l)}(\mathbf{x}, \mathbf{a}) \right)^{\delta(l=k)} = P^{(c,k)}(\mathbf{x}, \mathbf{a}), \tag{7}$$

where the column vector $\mathbf{P}^{(c,:)}(\mathbf{x}, \mathbf{a}) \in \Delta$ corresponds to the $c$-th row of the confusion matrix $\mathbf{P}(\mathbf{x}, \mathbf{a})$.

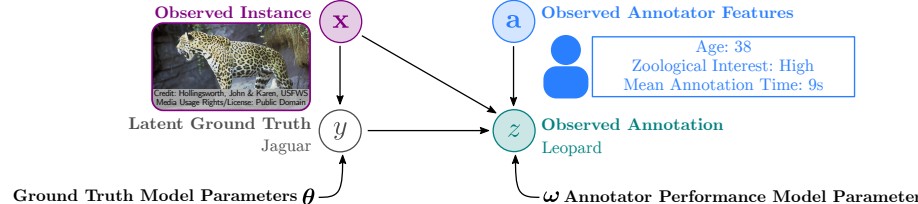

Figure 2: Probabilistic graphical model of MaDL.

## 4.2 Model Architectures

Now, we introduce how MaDL's GT and AP models are designed to approximate the functions of true class-membership probabilities $\mathbf{p}$ and true confusion matrices $\mathbf{P}$ for the respective instances and annotators. Fig. 3 illustrates the architecture of the GT (purple) and AP (green) models within our MaDL framework. Solid lines indicate mandatory components, while dashed lines express optional ones.

The GT model with parameters $\boldsymbol{\theta}$ is a (D)NN (cf. ④ in Fig. 3), which takes an instance $\mathbf{x}$ as input to approximate its true class-membership probabilities $\mathbf{p}(\mathbf{x})$ via $\hat{\mathbf{p}}_{\boldsymbol{\theta}}(\mathbf{x})$. We define its decision function in analogy to the Bayes optimal prediction in Eq. 3 through

$$\hat{y}_{\boldsymbol{\theta}}(\mathbf{x}) \coloneqq \underset{y \in \Omega_Y}{\arg\max} \left( \hat{p}_{\boldsymbol{\theta}}^{(y)}(\mathbf{x}) \right). \tag{8}$$

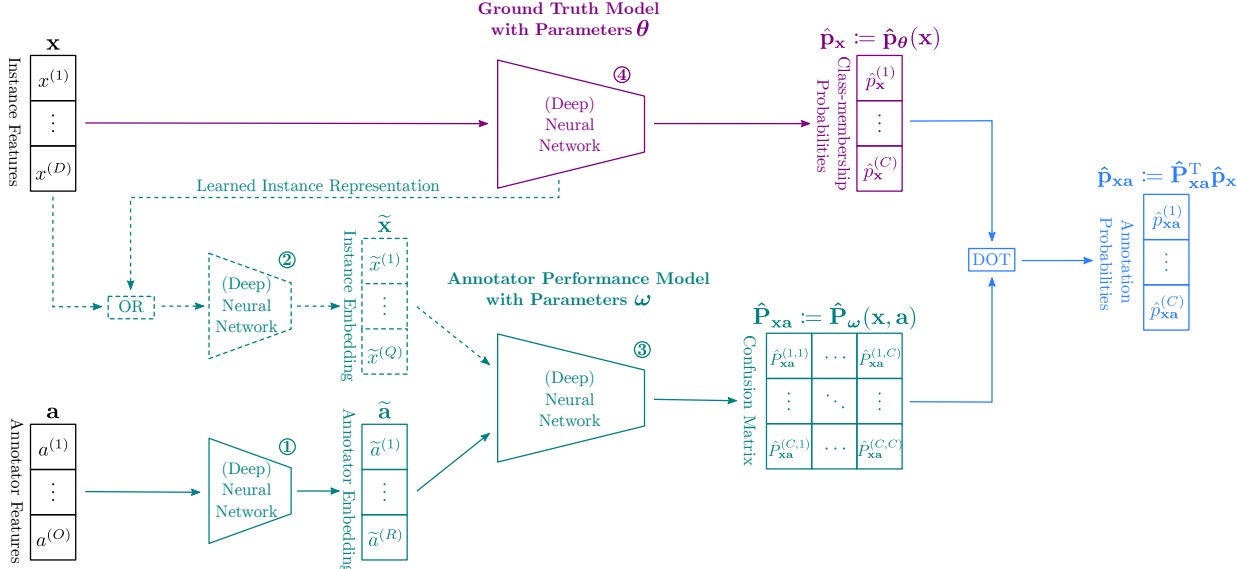

Figure 3: Architectures of MaDL's GT and AP models.

The architecture of the AP model with parameters $\boldsymbol{\omega}$ comprises mandatory and optional components. We start by describing its most general form, which consists of three (D)NNs and estimates annotator-, class-, and instance-dependent APs. Annotator features $\mathbf{a}$ are propagated through a first (D)NN (cf. ① in Fig. 3) to learn an annotator embedding $\widetilde{\mathbf{a}} \in \mathbb{R}^R, R \in \mathbb{N}_{\geq 1}$. During training, we will use such embeddings for quantifying correlations between annotators. Analogously, we propagate raw instance features $\mathbf{x}$ or a representation learned by the GT model's hidden layers through a second (D)NN (cf. ② in Fig. 3) for learning an instance embedding $\widetilde{\mathbf{x}} \in \mathbb{R}^Q, Q \in \mathbb{N}_{\geq 1}$. Subsequently, instance and annotator embeddings $\widetilde{\mathbf{x}}$ and $\widetilde{\mathbf{a}}$ are combined through a third and final (D)NN (cf. ③ in Fig. 3) for approximating the true confusion matrix $\mathbf{P}(\mathbf{x}, \mathbf{a})$ via $\hat{\mathbf{P}}_{\boldsymbol{\omega}}(\mathbf{x}, \mathbf{a})$. Various architectures for combining embeddings have already been proposed in the literature (Fiedler, 2021). We adopt a solution from recommender systems where often latent factors of users and items are combined (Zhang et al., 2019). Concretely, in DNN ③, we use an outer product-based layer outputting $\widetilde{\mathbf{o}} \in \mathbb{R}^F, F \in \mathbb{N}_{\geq 1}$ to model the interactions between instance and annotator embeddings (Qu et al., 2016). The concatenation of

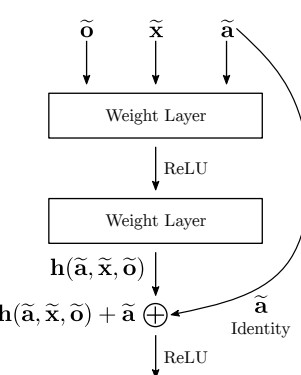

Figure 4: MaDL's residual block combining annotator and instance embedding.

$\widetilde{\mathbf{a}}, \widetilde{\mathbf{x}}$, and $\widetilde{\mathbf{o}}$ is propagated through a residual block (He et al., 2016), whose architecture is visualized in Fig. 4. There, we add only the annotator embedding $\widetilde{\mathbf{a}}$ to the learned mapping $\mathbf{h}(\widetilde{\mathbf{a}}, \widetilde{\mathbf{x}}, \widetilde{\mathbf{o}}) \in \mathbb{R}^R$. The motivation behind this modification is that the annotator embeddings, informing about an annotator's individuality, are likely to be the most influential inputs for estimating confusion matrices as APs. Empirical investigations showed that $R = Q = F = 16$ as the embedding size is a robust default. Finally, we define the AP model's decision function in analogy to the Bayes optimal prediction in Eq. 5 through

$$\hat{y}_{\boldsymbol{\theta}, \boldsymbol{\omega}}(\mathbf{x}, \mathbf{a}) := \delta\left(\sum_{c=1}^{C} \hat{p}_{\boldsymbol{\theta}}^{(c)}(\mathbf{x}) \cdot \hat{P}_{\boldsymbol{\omega}}^{(c,c)}(\mathbf{x}, \mathbf{a}) < 0.5\right) := \delta\left(\underbrace{\hat{p}_{\boldsymbol{\theta}, \boldsymbol{\omega}}(\mathbf{x}, \mathbf{a})}_{\text{predicted correctness probability}} < 0.5\right). \qquad (9)$$

An AP model estimating a confusion matrix per instance-annotator pair can be overly complex if there are only a few annotations per annotator or the number of classes is high (Rodrigues et al., 2013). In such settings, ignoring the instance features as input of the AP model may be beneficial. Alternatively, we can constrain a confusion matrix's degrees of freedom by reducing the number of output neurons of the AP model. For example, we might estimate only the diagonal elements of the confusion matrix and assume

that the remaining probability mass per row is uniformly distributed. Further, we can either estimate each diagonal element individually (corresponding to $C$ output neurons) or approximate them via a single scalar (corresponding to one output neuron). Appendix G illustrates such confusion matrices with varying degrees of freedom.

### 4.3 End-to-end Training

Given the probabilistic model and accompanying architectures of the GT and AP models, we propose an algorithm for jointly learning their parameters. A widespread method for training probabilistic models is to maximize the likelihood of the observed data with respect to the model parameters. Assuming that the joint distributions of annotations $\mathbf{Z}$ are conditionally independent for given instances $\mathbf{X}$, we can specify the likelihood function as follows:

$$\Pr(\mathbf{Z} \mid \mathbf{X}, \mathbf{A}; \boldsymbol{\theta}, \boldsymbol{\omega}) = \prod_{n=1}^{N} \Pr(\mathbf{z}_n \mid \mathbf{x}_n, \mathbf{A}; \boldsymbol{\theta}, \boldsymbol{\omega}). \tag{10}$$

We further expect that the distributions of annotations $\mathbf{z}_n$ for a given instance $\mathbf{x}_n$ are conditionally independent. Thus, we can simplify the likelihood function:

$$\Pr(\mathbf{Z} \mid \mathbf{X}, \mathbf{A}; \boldsymbol{\theta}, \boldsymbol{\omega}) = \prod_{n=1}^{N} \prod_{m \in \mathcal{A}_n} \Pr(z_{nm} \mid \mathbf{x}_n, \mathbf{a}_m; \boldsymbol{\theta}, \boldsymbol{\omega}). \tag{11}$$

Leveraging our probabilistic model in Fig. 2, we can express the probability of obtaining a certain annotation as an expectation with respect to an instance's (unknown) GT class label:

$$\Pr(\mathbf{Z} \mid \mathbf{X}, \mathbf{A}; \boldsymbol{\theta}, \boldsymbol{\omega}) = \prod_{n=1}^{N} \prod_{m \in \mathcal{A}_n} \mathbb{E}_{y_n \mid \mathbf{x}_n; \boldsymbol{\theta}} \left[ \Pr(z_{nm} \mid \mathbf{x}_n, \mathbf{a}_m, y_n; \boldsymbol{\omega}) \right] \tag{12}$$

$$= \prod_{n=1}^{N} \prod_{m \in \mathcal{A}_n} \left( \sum_{y_n=1}^{C} \Pr(y_n \mid \mathbf{x}_n; \boldsymbol{\theta}) \Pr(z_{nm} \mid \mathbf{x}_n, \mathbf{a}_m, y_n; \boldsymbol{\omega}) \right) \tag{13}$$

$$= \prod_{n=1}^{N} \prod_{m \in \mathcal{A}_n} \mathbf{e}_{z_{nm}}^{\mathrm{T}} \underbrace{\hat{\mathbf{P}}_{\boldsymbol{\omega}}^{\mathrm{T}}(\mathbf{x}_n, \mathbf{a}_m) \hat{\mathbf{p}}_{\boldsymbol{\theta}}(\mathbf{x}_n)}_{\text{annotation probabilities}}, \tag{14}$$

where $\mathbf{e}_{z_{nm}}$ denotes the one-hot encoded vector of annotation $z_{nm}$. Taking the logarithm of this likelihood function and converting the maximization into a minimization problem, we get

$$L_{\mathbf{X}, \mathbf{A}, \mathbf{Z}}(\boldsymbol{\theta}, \boldsymbol{\omega}) \coloneqq -\sum_{n=1}^{N} \sum_{m \in \mathcal{A}_n} \ln \left( \mathbf{e}_{z_{nm}}^{\mathrm{T}} \hat{\mathbf{P}}_{\boldsymbol{\omega}}^{\mathrm{T}}(\mathbf{x}_n, \mathbf{a}_m) \hat{\mathbf{p}}_{\boldsymbol{\theta}}(\mathbf{x}_n) \right) \tag{15}$$

as cross-entropy loss function for learning annotation probabilities by combining the outputs of the GT and AP models (cf. blue components in Fig. 3). Yet, directly employing this loss function for learning may result in poor results for two reasons.

**Initialization:** Reason number one has been noted by Tanno et al. (2019), who showed that such a loss function cannot ensure the separation of the AP and GT label distributions. This is because infinite many combinations of class-membership probabilities and confusion matrices perfectly comply with the true annotation probabilities, e.g., by swapping the rows of the confusion matrix as the following example shows:

$$\underbrace{\mathbf{P}^{\mathrm{T}}(\mathbf{x}_n, \mathbf{a}_m) \mathbf{p}(\mathbf{x}_n)}_{\text{true probabilities}} = \begin{pmatrix} 1 & 0 \\ 0 & 1 \end{pmatrix} \begin{pmatrix} 1 \\ 0 \end{pmatrix} = \begin{pmatrix} 1 \\ 0 \end{pmatrix} = \begin{pmatrix} 0 & 1 \\ 1 & 0 \end{pmatrix} \begin{pmatrix} 0 \\ 1 \end{pmatrix} = \underbrace{\hat{\mathbf{P}}_{\boldsymbol{\omega}}^{\mathrm{T}}(\mathbf{x}_n, \mathbf{a}_m) \hat{\mathbf{p}}_{\boldsymbol{\theta}}(\mathbf{x}_n)}_{\text{predicted probabilities}}. \tag{16}$$

Possible approaches aim at resolving this issue by favoring certain combinations, e.g., diagonally dominant confusion matrices. Typically, one can achieve this via regularization (Tanno et al., 2019; Zhang et al., 2020;

Li et al., 2022) and/or suitable initialization of the AP model's parameters (Rodrigues & Pereira, 2018; Wei et al., 2022). We rely on the latter approach because it permits encoding prior knowledge about APs. Concretely, we approximate an initial confusion matrix for any instance-annotator pair $(\mathbf{x}_n, \mathbf{a}_m)$ through

$$\hat{\mathbf{P}}_{\boldsymbol{\omega}}(\mathbf{x}_n, \mathbf{a}_m) := \begin{pmatrix} \texttt{softmax}((\mathbf{v}^{\mathrm{T}}(\mathbf{x}_n, \mathbf{a}_m)\mathbf{W} + \mathbf{B})^{(1,:)}) \\ \vdots \\ \texttt{softmax}((\mathbf{v}^{\mathrm{T}}(\mathbf{x}_n, \mathbf{a}_m)\mathbf{W} + \mathbf{B})^{(C,:)}) \end{pmatrix} \approx \eta \mathbf{I}_C + \frac{(1-\eta)}{C-1}(\mathbf{1}_C - \mathbf{I}_C), \tag{17}$$

where $\mathbf{I}_C \in \mathbb{R}^{C \times C}$ denotes an identity matrix, $\mathbf{1}_C \in \mathbb{R}^{C \times C}$ an all-one matrix, and $\eta \in (0,1)$ the prior probability of obtaining a correct annotation. For example, in a binary classification problem, the initial confusion matrix would approximately take the following values:

$$P_{\boldsymbol{\omega}}(\mathbf{x}_n, \mathbf{a}_m) \approx \begin{pmatrix} \eta & 1-\eta \\ 1-\eta & \eta \end{pmatrix}. \tag{18}$$

The outputs of the $\texttt{softmax}$ functions represent the confusion matrix's rows. Provided that the initial AP model's last layer's weights $\mathbf{W} \in \mathbb{R}^{H \times C \times C}, H \in \mathbb{N}_{>0}$ satisfy $\mathbf{v}^{\mathrm{T}}(\mathbf{x}_n, \mathbf{a}_m)\mathbf{W} \approx \mathbf{0}_C \in \mathbb{R}^{C \times C}$ for the hidden representation $\mathbf{v}(\mathbf{x}_n, \mathbf{a}_m) \in \mathbb{R}^H$ of each instance-annotator pair, we approximate Eq. 17 by initializing the biases $\mathbf{B} \in \mathbb{R}^{C \times C}$ of our AP model's output layer via

$$\mathbf{B} := \ln\left(\frac{\eta \cdot (C-1)}{1-\eta}\right) \mathbf{I}_C. \tag{19}$$

By default, we set $\eta = 0.8$ to assume trustworthy annotators a priori. Accordingly, initial class-membership probability estimates are close to the annotation probability estimates.

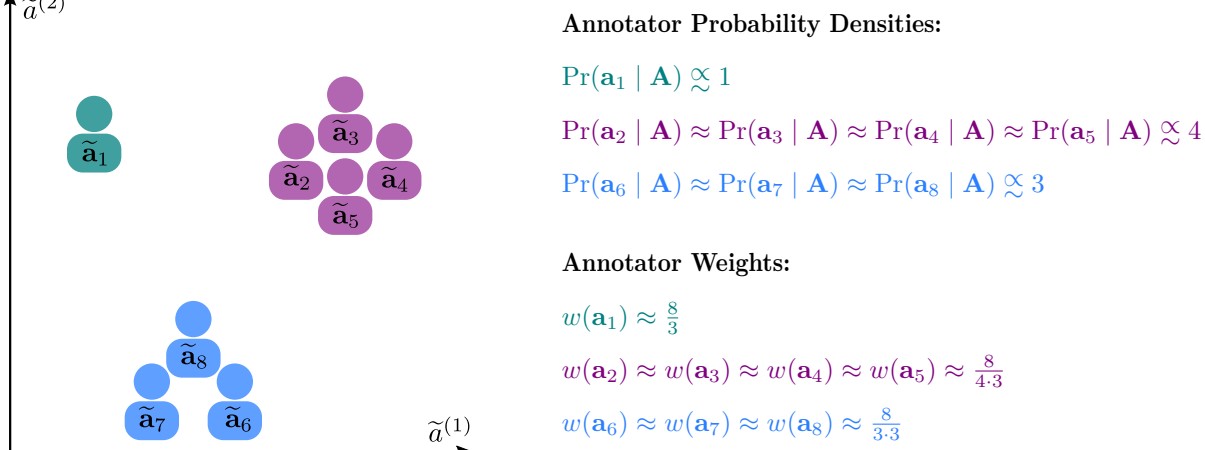

Figure 5: Visualization of annotator embeddings (left) accompanied by an exemplary calculation of annotator probability densities and annotator weights (right).

**Annotator weights:** Reason number two has been noted by Cao et al. (2019), who proved that maximum-likelihood solutions fail when there are strong annotator correlations, i.e., annotators with significant statistical correlations in their annotation patterns. To address this issue, we explore the annotator correlations in the latent space of the learned annotator embeddings. For this purpose, we assume that annotators with similar embeddings share correlated annotation patterns. Recalling our example in Fig. 1, this assumption implies that annotators of the same latent group are located near each other. The left plot of Fig. 5 visualizes this assumption for a two-dimensional embedding space, where the eight annotators are arranged into three clusters as proxies of the three latent annotator groups. We aim to extend our loss function so that its evaluation is independent of the annotator groups' cardinalities. For our example, we view the three annotator groups as three independent annotators of equal importance. To this purpose, we extend the original

likelihood function in Eq. 11 by annotator weights, such that we obtain the weighted likelihood function:

$$\Pr(\mathbf{Z} \mid \mathbf{X}, \mathbf{A}; \boldsymbol{\theta}, \boldsymbol{\omega}, \mathbf{w}) = \prod_{n=1}^{N} \prod_{m \in \mathcal{A}_n} \Pr(z_{nm} \mid \mathbf{x}_n, \mathbf{a}_m; \boldsymbol{\theta}, \boldsymbol{\omega})^{w(\mathbf{a}_m)}, \tag{20}$$

where $\mathbf{w} := (w(\mathbf{a}_1), \ldots, w(\mathbf{a}_M))^{\mathrm{T}} \in \mathbb{R}_{\geq 0}^{M}$ denotes a vector of non-negative annotator weights. From a probabilistic perspective, we can interpret such a weight $w(\mathbf{a}_m)$ as the effective number of observations (or copies) per annotation of annotator $\mathbf{a}_m$. Interpreting the annotators $\mathbf{A}$ as samples from a continuous latent space, we define an annotator weight $w(\mathbf{a}_m)$ to be inversely proportional to an annotator's $\mathbf{a}_m$ probability density:

$$w(\mathbf{a}_m) := \frac{\Pr(\mathbf{a}_m \mid \mathbf{A})^{-1}}{Z}, Z := M^{-1} \left( \sum_{m=1}^{M} \Pr(\mathbf{a}_m \mid \mathbf{A})^{-1} \right) \text{ provided that } \Pr(\mathbf{a}_1 \mid \mathbf{A}), \ldots, \Pr(\mathbf{a}_M \mid \mathbf{A}) > 0. \tag{21}$$

The normalization term $Z \in \mathbb{R}_{>0}$ ensures that the number of effective annotations remains equal to the number of annotators, i.e., $\sum_{m=1}^{M} w(\mathbf{a}_m) = M$. On the right side of our example in Fig. 5, we expect that an annotator's probability density is approximately proportional to the cardinality of the group to which the annotator belongs. As a result, we assign high (low) weights to annotators belonging to small (large) groups. Inspecting the exemplary annotator weights and adding the weights per annotator group, we observe that each group provides the same number of effective annotations, i.e., $8/3$. More generally, we support our definition of the annotator weights by the following theorem, whose proof is given in Appendix A.

---

**Theorem 1.** *Let there be $G \in \{1, \ldots, M\}$ non-empty, disjoint annotator groups, which we denote as sets of indices such that $\mathcal{A}^{(1)} \cup \cdots \cup \mathcal{A}^{(G)} = \{1, \ldots, M\}$. Further assume, the annotators within each group $g \in \{1, \ldots, G\}$ share identical annotation patterns for the observed instances, i.e.,*

$$\forall n \in \{1, \ldots, N\}, \forall m, l \in \mathcal{A}^{(g)} : z_{nm} = z_{nl} \wedge \Pr(z_{nm} \mid \mathbf{x}_n, \mathbf{a}_m) = \Pr(z_{nl} \mid \mathbf{x}_n, \mathbf{a}_l), \tag{$\dagger$}$$

*and the annotators' probability densities are proportional to their respective groups' cardinalities, i.e.,*

$$\forall m \in \{1, \ldots, M\} : \Pr(\mathbf{a}_m \mid \mathbf{A}) \propto \sum_{g=1}^{G} \delta(m \in \mathcal{A}^{(g)}) |\mathcal{A}^{(g)}|. \tag{$\star$}$$

*Then, the true weighted log-likelihood function for all $M$ annotators reduces to the log-likelihood for $G$ annotators:*

$$\sum_{n=1}^{N} \sum_{m=1}^{M} w(\mathbf{a}_m) \ln \left( \Pr(z_{nm} \mid \mathbf{x}_n, \mathbf{a}_m) \right) \propto \sum_{n=1}^{N} \sum_{g=1}^{G} \ln \left( \Pr(z_{nm_g} \mid \mathbf{x}_n, \mathbf{a}_{m_g}) \right),$$

*where $m_g \in \mathcal{A}^{(g)}$ represents the index of an arbitrary annotator of the g-th annotator group.*

---

Intuitively, Theorem 1 confirms that each group $\mathcal{A}^{(g)}$, independent of its cardinality $|\mathcal{A}^{(g)}|$, equally contributes to the weighted log-likelihood function. This way, we avoid any bias toward a large group of highly correlated annotators during learning. Typically, the assumptions ($\dagger$) and ($\star$) of Theorem 1 do not hold in practice because there are no annotator groups with identical annotation patterns. Therefore, we estimate degrees of correlations between annotators by computing similarities between their embeddings $\widetilde{\mathbf{A}} := (\widetilde{\mathbf{a}}_1, \ldots, \widetilde{\mathbf{a}}_M)^{\mathrm{T}}$ as the basis for a nonparametric annotator probability density estimation:

$$\Pr(\mathbf{a}_m \mid \mathbf{A}) \approx \Pr\left(\widetilde{\mathbf{a}}_m \mid \widetilde{\mathbf{A}}, k_\gamma\right) \propto \sum_{l=1}^{M} k_\gamma \left(\texttt{no\_grad}(\widetilde{\mathbf{a}}_l), \texttt{no\_grad}(\widetilde{\mathbf{a}}_m)\right), \tag{22}$$

where $k_\gamma : \mathbb{R}^{R \times R} \to \mathbb{R}_{\geq 0}$ denotes a kernel function and $\gamma \in \mathbb{R}_{>0}$ its kernel scale. The expression $\texttt{no\_grad}(\widetilde{\mathbf{a}}_m) \in \mathbb{R}^{R}$ indicates that no gradient regarding the learned annotator embedding $\widetilde{\mathbf{a}}_m$ is computed,

which is necessary to decouple the learning of embeddings from computing annotator weights. Otherwise, we would learn annotator embeddings, which optimize the annotator weights instead of reflecting the annotation patterns. Although many kernel (or similarity) functions are conceivable, we will focus on the popular Gaussian kernel:

$$k_\gamma(\texttt{no\_grad}\,(\widetilde{\mathbf{a}}_m)\,,\texttt{no\_grad}\,(\widetilde{\mathbf{a}}_l)) \propto \exp\left(-\gamma\,||\texttt{no\_grad}\,(\widetilde{\mathbf{a}}_m) - \texttt{no\_grad}\,(\widetilde{\mathbf{a}}_l)\,||_2^2\right) \tag{23}$$

with $||\cdot||_2$ as Euclidean distance. Typically, the kernel scale $\gamma$ needs to fit the observed data, i.e., annotator embeddings in our case. Therefore, its definition a priori is challenging, such that we define $\gamma$ as a learnable parameter subject to a prior distribution. Concretely, we employ the gamma distribution for this purpose:

$$\Pr\left(\gamma \mid \alpha, \beta\right) \coloneqq \text{Gam}\left(\gamma \mid \alpha, \beta\right) \coloneqq \frac{\beta^\alpha}{\Gamma(\alpha)}\gamma^{\alpha-1}\exp\left(-\beta\gamma\right), \tag{24}$$

where $\Gamma$ is the gamma function and $\alpha \in \mathbb{R}_{>1}, \beta \in \mathbb{R}_{>0}$ are hyperparameters. Based on experiments, we set $\alpha = 1.25, \beta = 0.25$ such that the mode is $(\alpha-1)/\beta = 1$ (defining the initial value of $\gamma$ before optimization) and the variance with $\alpha/\beta^2 = 20$ is high in favor of flexible learning. As a weighted loss function, we finally get

$$L_{\mathbf{X},\mathbf{A},\mathbf{Z},\alpha,\beta}(\boldsymbol{\theta},\boldsymbol{\omega},\gamma) \coloneqq -\frac{1}{|\mathbf{Z}|}\sum_{n=1}^N\sum_{m\in\mathcal{A}_n}\left(\hat{w}_\gamma(\mathbf{a}_m)\ln\left(\mathbf{e}_{z_{nm}}^{\mathrm{T}}\hat{\mathbf{P}}_{\boldsymbol{\omega}}^{\mathrm{T}}(\mathbf{x}_n,\mathbf{a}_m)\hat{\mathbf{p}}_{\boldsymbol{\theta}}(\mathbf{x}_n)\right)\right) - \ln\left(\text{Gam}\left(\gamma \mid \alpha,\beta\right)\right), \tag{25}$$

$$|\mathbf{Z}| \coloneqq \sum_{n=1}^N\sum_{m=1}^M\delta(z_{nm}\in\Omega_Y), \tag{26}$$

where $\hat{w}_\gamma(\mathbf{a}_m)$ denotes that the annotator weights $w(\mathbf{a}_m)$ are estimated by learning the kernel scale $\gamma$. The number of annotations $|\mathbf{Z}|$ is a normalization factor, which accounts for potentially unevenly distributed annotations across mini-batches when using stochastic GD.

Given the loss function in Eq. 25, we present the complete **end-to-end training algorithm** of MaDL in Algorithm 1 and an example in Appendix B. During each training step, we recompute the annotator weights and use them as the basis for the weighted loss function to optimize the AP and GT models' parameters. After training, the optimized model parameters $(\boldsymbol{\theta},\boldsymbol{\omega})$ can be used to make probabilistic predictions, e.g., class-membership probabilities $\hat{\mathbf{p}}_{\boldsymbol{\theta}}(\mathbf{x})$ (cf. Fig. 3) and annotator confusion matrix $\hat{\mathbf{P}}_{\boldsymbol{\omega}}(\mathbf{x},\mathbf{a})$ (cf. Fig. 3), or to decide on distinct labels, e.g., class label $\hat{y}_{\boldsymbol{\theta}}(\mathbf{x})$ (cf. Eq. 8) and annotation error $\hat{y}_{\boldsymbol{\theta},\boldsymbol{\omega}}(\mathbf{x},\mathbf{a})$ (cf. Eq. 9).

---

**Algorithm 1:** End-to-end training algorithm of MaDL.

---

**input:** instances $\mathbf{X}$, annotators $\mathbf{A}$, annotations $\mathbf{Z}$, number of training epochs $E$, mini-batch size $B$,
  initial model parameters $(\boldsymbol{\theta},\boldsymbol{\omega})$, prior annotation accuracy $\eta$, gamma distribution parameters $(\alpha,\beta)$;
**start:** initialize biases $\mathbf{B}$ of the AP model's output layer using $\eta$ (cf. Eq. 19);
  initialize kernel scale $\gamma \coloneqq (\alpha-1)/\beta$ ;
**for** epoch $e \in \{1,\dots,E\}$ **do**
  **for** sampled mini-batch $\overline{\mathbf{X}} \coloneqq (\mathbf{x}_{i_1},\dots,\mathbf{x}_{i_B})^{\mathrm{T}}, \overline{\mathbf{Z}} \coloneqq (\mathbf{z}_{i_1},\dots,\mathbf{z}_{i_B})^{\mathrm{T}}$ with $\{i_1,\dots,i_B\}\subset\{1,\dots,N\}$ **do**
    **for** $b \in \{i_1,\dots,i_B\}$ **do**
      compute class-membership probabilities $\hat{\mathbf{p}}_{\boldsymbol{\theta}}(\mathbf{x}_b)$ (cf. Fig. 3);
      **for** $m \in \{1,\dots,M\}$ **do** compute confusion matrix $\hat{\mathbf{P}}_{\boldsymbol{\omega}}(\mathbf{x}_b,\mathbf{a}_m)$ (cf. Fig. 3); **end**
    **end**
    **for** $(m,l) \in \{1,\dots,M\}^2$ **do** compute similarity $k_\gamma(\texttt{no\_grad}(\widetilde{\mathbf{a}}_m),\texttt{no\_grad}(\widetilde{\mathbf{a}}_l))$ (cf. Eq. 23); **end**
    **for** $m \in \{1,\dots,M\}$ **do** compute annotator weight $w(\mathbf{a}_m) \approx \hat{w}_\gamma(\mathbf{a}_m)$ (cf. Eq. 21 and Eq. 22); **end**
    optimize parameters $\boldsymbol{\theta},\boldsymbol{\omega},\gamma$ with reference to $L_{\overline{\mathbf{X}},\mathbf{A},\overline{\mathbf{Z}},\alpha,\beta}(\boldsymbol{\theta},\boldsymbol{\omega},\gamma)$ (cf. Eq. 25);
  **end**
**end**
**output:** optimized model parameters $(\boldsymbol{\theta},\boldsymbol{\omega})$

---

## 5 Experimental Evaluation

This section investigates three RQs regarding the properties P1–P6 (cf. Section 3) of multi-annotator supervised learning. We divide the analysis of each RQ into four parts, which are (1) a takeaway summarizing the key insights, (2) a setup describing the experiments, (3) a qualitative study, and (4) a quantitative study. The qualitative studies intuitively explain our design choices about MaDL, while the quantitative studies compare MaDL's performance to related techniques. Note that we analyze each RQ in the context of a concrete evaluation scenario. Accordingly, the results provide potential indications for an extension to related scenarios. As this section's starting point, we overview the general experimental setup, whose code base is publicly available at `https://www.github.com/ies-research/multi-annotator-deep-learning`.

### 5.1 Experimental Setup

We base our experimental setup on the problem setting in Section 2. Accordingly, the goal is to evaluate the predictions of GT and AP models trained via multi-annotator supervised learning techniques. For this purpose, we perform experiments on several datasets with class labels provided by error-prone annotators, with models of varying hyperparameters, and in combination with a collection of different evaluation scores.

**Datasets:** We conduct experiments for the tabular and image datasets listed by Table 2. LABELME and MUSIC are actual crowdsourcing datasets, while we simulate annotators for the other five datasets. For the LABELME dataset, Rodrigues & Pereira (2018) performed a crowdsourcing study to annotate a subset of 1000 out of 2688 instances of eight different classes as training data. This dataset consists of images, but due to its small training set size, we follow the idea of Rodrigues & Pereira and transform it into a tabular dataset by utilizing the features of a pretrained VGG-16 (Simonyan & Zisserman, 2015) as inputs. There are class labels obtained from 59 different annotators, and on average, about 2.5 class labels are assigned to an instance. MUSIC is another crowdsourcing dataset, where 700 of 1000 audio files are classified into ten music genres by 44 annotators, and on average, about 2.9 class labels are assigned to a file. We use the features extracted by Rodrigues et al. (2013) from the audio files for training and inference. The artificial TOY dataset with two classes and features serves to visualize our design choices about MaDL. We generate this dataset via a Gaussian mixture model. Frey & Slate (1991) published the LETTER dataset to recognize a pixel display, represented through statistical moments and edge counts, as one of the 26 capital letters in the alphabet for Modern English. The datasets FMNIST, CIFAR10, and SVHN represent typical image benchmark classification tasks, each with ten classes but different object types to recognize. Appendix F presents a separate case study on CIFAR100 to investigate the outcomes on datasets with more classes.

Table 2: Overview of datasets and associated base network architectures.

| Dataset | Annotators | Instances | Classes | Features | Base Network Architecture |
|---|---|---|---|---|---|
| Tabular Datasets | | | | | |
| TOY | simulated | 500 | 2 | 2 | MLP (Rodrigues & Pereira, 2018) |
| LETTER (Frey & Slate, 1991) | simulated | 20000 | 26 | 16 | MLP (Rodrigues & Pereira, 2018) |
| LABELME (Rodrigues & Pereira, 2018) | real-world | 2688 | 8 | 8192 | MLP (Rodrigues & Pereira, 2018) |
| MUSIC (Rodrigues et al., 2013) | real-world | 1000 | 10 | 124 | MLP (Rodrigues & Pereira, 2018) |
| Image Datasets | | | | | |
| FMNIST (Xiao et al., 2017) | simulated | 70000 | 10 | $1 \times 28 \times 28$ | LeNet-5 (LeCun & Cortes, 1998) |
| CIFAR10 (Krizhevsky, 2009) | simulated | 60000 | 10 | $3 \times 32 \times 32$ | ResNet-18 (He et al., 2016) |
| SVHN (Netzer et al., 2011) | simulated | 99289 | 10 | $3 \times 32 \times 32$ | ResNet-18 (He et al., 2016) |

**Network Architectures:** Table 2 lists the base network architectures selected to meet the datasets' requirements. These architectures are starting points for designing the GT and AP models, which we adjust according to the respective multi-annotator supervised learning technique. For the tabular datasets, we follow Rodrigues & Pereira (2018) and train a *multilayer perceptron* (MLP) with a single fully connected layer of 128 neurons as a hidden layer. A modified LeNet-5 architecture (LeCun & Cortes, 1998), a simple convolutional neural network, serves as the basis for FMNIST as a gray-scale image dataset, while we employ a ResNet-18 (He et al., 2016) for CIFAR10 and SVHN as RGB image datasets. We refer to our code base for remaining details, e.g., on the use of *rectified linear units* (ReLU, Glorot et al. 2011) as activation functions.

**Annotator simulation:** For the datasets without real-world annotators, we adopt simulation strategies from related work (Yan et al., 2014; Cao et al., 2019; Rühling Cachay et al., 2021; Wei et al., 2022) and simulate annotators according to the following five types:

*Adversarial* annotators provide false class labels on purpose. In our case, such an annotator provides a correct class label with a probability of 0.05.

*Randomly guessing* annotators provide class labels drawn from a uniform categorical distribution. As a result, such an annotator provides a correct class label with a probability of $1/C$.

*Cluster-specialized* annotators' performances considerably vary across the clusters found by the $k$-means clustering algorithm. For images, we cluster the latent representations of the ResNet-18 pretrained on ImageNet (Russakovsky et al., 2015). In total, there are $k = 10$ clusters. For each annotator, we randomly define five weak and five expert clusters. An annotator provides a correct class label with a probability of 0.95 for an expert cluster and with a probability of 0.05 for a weak cluster.

*Common* annotators are simulated based on the identical clustering employed for the cluster-specialized annotators. However, their APs vary less between the clusters. Concretely, we randomly draw a correctness probability value in the range $[1/C, 1]$ for each cluster-annotator pair.

*Class-specialized* annotators' performances considerably vary across classes to which instances can belong. For each annotator, we randomly define $\lfloor C/2 \rfloor$ weak and $\lceil C/2 \rceil$ expert classes. An annotator provides a correct class label with a probability of 0.95 for an expert class and with a probability of 0.05 for a weak class.

We simulate annotation mistakes by randomly selecting false class labels. Table 3 lists four annotator sets (blueish rows) with varying numbers of annotators per annotator type (first five columns) and annotation ratios (last column). Each annotator set is associated with a concrete RQ. A copy flag indicates that the annotators in the respective types provide identical annotations. This way, we follow Wei et al. (2022), Cao et al. (2019), and Rühling Cachay et al. (2021) to simulate strong correlations between annotators. For example, the entry "1 + 11 copies" of the annotator set CORRELATED indicates twelve cluster-specialized annotators, of which one annotator is independent, while the remaining eleven annotators share identical annotation patterns, i.e., they are copies of each other. The simulated annotator correlations are not directly observable because the copied annotators likely annotate different instances. This is because of the annotation ratios, e.g., a ratio of 0.2 indicates that each annotator provides annotations for only 20 % of randomly chosen instances. The annotation ratios are well below 1.0 because, in practice (especially in crowdsourcing applications), it is unrealistic for every annotator to annotate every instance. We refer to Appendix E presenting the results of a case study with higher annotation ratios for CIFAR10.

Table 3: Simulated annotator sets for each RQ.

| Adversarial | Common | Cluster-specialized | Class-specialized | Random | Annotation Ratio |
|---|---|---|---|---|---|
| INDEPENDENT (RQ1) | | | | | |
| 1 | 6 | 2 | 1 | 0 | 0.2 |
| CORRELATED (RQ2) | | | | | |
| 11 copies | 6 | 1 + 11 copies | 11 copies | 0 | 0.2 |
| RANDOM-CORRELATED (RQ2) | | | | | |
| 1 | 6 | 2 | 1 | 90 copies | 0.2 |
| INDUCTIVE (RQ3) | | | | | |
| 10 | 60 | 20 | 10 | 0 | 0.02 |

**Evaluation scores:** Since we are interested in quantitatively assessing GT and AP predictions, we need corresponding evaluation scores. In this context, we interpret the prediction of APs as a binary classification problem with the AP model predicting whether an annotator provides the correct or a false class label for an instance. Next to categorical predictions, the GT and AP models typically provide probabilistic outputs, which we examine regarding their quality (Huseljic et al., 2021). We list our evaluation scores in the following, where arrows indicate which scores need to be maximized ($\uparrow$) or minimized ($\downarrow$):

*Accuracy* (ACC, ↑) is probably the most popular score for assessing classification performances. For the GT estimates, it describes the fraction of correctly classified instances, whereas it is the fraction of (potential) annotations correctly identified as false or correct for the AP estimates:

$$\text{GT-ACC}(\mathbf{X}, \mathbf{y}, \hat{y}_{\boldsymbol{\theta}}) \coloneqq \frac{1}{N} \sum_{n=1}^{N} \delta\left(y_n = \hat{y}_{\boldsymbol{\theta}}(\mathbf{x}_n)\right), \tag{27}$$

$$\text{AP-ACC}(\mathbf{X}, \mathbf{y}, \mathbf{Z}, \hat{y}_{\boldsymbol{\theta}, \boldsymbol{\omega}}) \coloneqq \frac{1}{|\mathbf{Z}|} \sum_{n=1}^{N} \sum_{m \in \mathcal{A}_n} \delta\left(\delta\left(y_n \neq z_{nm}\right) = \hat{y}_{\boldsymbol{\theta}, \boldsymbol{\omega}}(\mathbf{x}_n, \mathbf{a}_m)\right). \tag{28}$$

Maximizing both scores corresponds to the Bayes optimal predictions in Eq. 3 and Eq. 5.

*Balanced accuracy* (BAL-ACC, ↑) is a variant of ACC designed for imbalanced classification problems (Brodersen et al., 2010). For the GT estimation, the idea is to compute the ACC score for each class of instances separately and then average them. Since our datasets are fairly balanced in their distributions of class labels, we use this evaluation score only for assessing AP estimates. We may encounter highly imbalanced binary classification problems per annotator, where a class represents either a false or correct annotation. For example, an adversarial annotator provides majorly false annotations. Therefore, we extend the definition of BAL-ACC by computing the ACC scores for each annotator-class pair separately to average them.

*Negative log-likelihood* (NLL, ↓) is not only used as a typical loss function for training (D)NNs but can also be used to assess the quality of probabilistic estimates:

$$\text{GT-NLL}(\mathbf{X}, \mathbf{y}, \hat{\mathbf{p}}_{\boldsymbol{\theta}}) \coloneqq -\frac{1}{N} \sum_{n=1}^{N} \ln\left(\hat{p}_{\boldsymbol{\theta}}^{(y_n)}(\mathbf{x}_n)\right), \tag{29}$$

$$\text{AP-NLL}(\mathbf{X}, \mathbf{y}, \mathbf{Z}, \hat{p}_{\boldsymbol{\theta}, \boldsymbol{\omega}}) \coloneqq$$

$$-\frac{1}{|\mathbf{Z}|} \sum_{n=1}^{N} \sum_{m \in \mathcal{A}_n} \left( \delta\left(y_n = z_{nm}\right) \ln\left(\hat{p}_{\boldsymbol{\theta}, \boldsymbol{\omega}}(\mathbf{x}_n, \mathbf{a}_m)\right) + \delta\left(y_n \neq z_{nm}\right) \ln\left(1 - \hat{p}_{\boldsymbol{\theta}, \boldsymbol{\omega}}(\mathbf{x}_n, \mathbf{a}_m)\right) \right). \tag{30}$$

Moreover, NLL is a proper scoring rule (Ovadia et al., 2019) such that the best score corresponds to a perfect prediction.

*Brier score* (BS, ↓), proposed by Brier (1950), is another proper scoring rule, which measures the squared error between predicted probability vectors and one-hot encoded target vectors:

$$\text{GT-BS}(\mathbf{X}, \mathbf{y}, \hat{\mathbf{p}}_{\boldsymbol{\theta}}) \coloneqq \frac{1}{N} \sum_{n=1}^{N} ||\mathbf{e}_{y_n} - \hat{\mathbf{p}}_{\boldsymbol{\theta}}(\mathbf{x}_n)||_2^2, \tag{31}$$

$$\text{AP-BS}(\mathbf{X}, \mathbf{y}, \mathbf{Z}, \hat{p}_{\boldsymbol{\theta}, \boldsymbol{\omega}}) \coloneqq \frac{1}{|\mathbf{Z}|} \sum_{n=1}^{N} \sum_{m \in \mathcal{A}_n} \left(\delta\left(y_n = z_{nm}\right) - \hat{p}_{\boldsymbol{\theta}, \boldsymbol{\omega}}(\mathbf{x}_n, \mathbf{a}_m)\right)^2. \tag{32}$$

In the literature, there exist many further evaluation scores, particularly for assessing probability calibration (Ovadia et al., 2019). As a comprehensive evaluation of probabilities is beyond this article's scope, we focus on the aforementioned proper scoring rules. Accordingly, we have omitted other evaluation scores, such as the expected calibration error (Naeini et al., 2015) being a non-proper scoring rule.

**Multi-annotator supervised learning techniques:** By default, we train MaDL via the weighted loss function in Eq. 25 using the hyperparameter values from Section 4 and the most general architecture depicted by Fig. 3. Next to the ablations as part of analyzing the three RQs, we present an ablation study on the hyperparameters of MaDL in Appendix C and a practitioner's guide with concrete recommendations in Appendix G. We evaluate MaDL compared to a subset of the related techniques presented in Section 3. This subset consists of techniques that (1) provide probabilistic GT estimates for each instance, (2) provide probabilistic AP estimates for each instance-annotator pair, and (3) train a (D)NN as the GT model. Moreover,

we focus on recent techniques with varying training algorithms and properties P1–P6 (cf. Section 3). As a result, we select *crowd layer* (CL, Rodrigues & Pereira, 2018), *regularized estimation of annotator confusion* (REAC, Tanno et al., 2019), *learning from imperfect annotators* (LIA, Platanios et al., 2020), *common noise adaption layers* (CoNAL, Chu et al., 2021), and *union net* (UNION, Wei et al., 2022). Further, we aggregate annotations through the majority rule as a *lower baseline* (LB) and use the GT class labels as an *upper baseline* (UB). We adopt the architectures of MaDL's GT and AP models for both baselines. The GT model then trains via the aggregated annotation (LB) or the GT class labels (UB). The AP model trains using the aggregated annotations (LB) or the GT class labels (UB) to optimize the annotator confusion matrices. Unless explicitly stated, no multi-annotator supervised learning technique can access annotator features containing prior knowledge about the annotators.

**Experiment:** An experiment's run starts by splitting a dataset into train, validation, and test sets. For MUSIC and LABELME, these splits are predefined, while for the other datasets, we randomly select 75 % of the samples for training, 5 % for validation, and 20 % for testing. Following Rühling Cachay et al. (2021), a small validation set with GT class labels allows a fair comparison by finding suitable hyperparameter values for the optimizer of the respective multi-annotator supervised learning technique. We employ the AdamW (Loshchilov & Hutter, 2019) optimizer, where the learning rates $\{0.01, 0.005, 0.001\}$ and weight decays $\{0.0, 0.001, 0.0001\}$ are tested. We decay learning rates via a cosine annealing schedule (Loshchilov & Hutter, 2017) and set the optimizer's mini-batch size to 64. For the datasets MUSIC and LABELME, we additionally perform experiments with 8 and 16 as mini-batch sizes due to their smaller number of instances and, thus, higher sensitivity to the mini-batch size. The number of training epochs is set to 100 for all techniques except for LIA, which we train for 200 epochs due to its EM algorithm. After training, we select the models with the best validation GT-ACC across the epochs. Each experiment is run five times with different parameter initializations and data splits (except for LABELME and MUSIC). We report quantitative results as means and standard deviations over the best five runs determined via the validation GT-ACC.

## 5.2 RQ1: Do class- and instance-dependent modeled APs improve learning? (Properties P1, P2)

> **Takeaway:** Estimating class- (property P1) and instance-dependent (property P2) APs leads to superior performances of the GT and AP models. This observation is especially true for GT models trained on datasets with real-world annotators whose annotation patterns are unknown.

**Setup:** We address RQ1 by evaluating multi-annotator supervised learning techniques with varying AP assumptions. We simulate ten annotators for the datasets without real-world annotators according to the annotator set INDEPENDENT in Table 3. Each simulated annotator provides class labels for 20 % of randomly selected training instances. Next to the related multi-annotator supervised learning techniques and the two baselines, we evaluate six variants of MaDL denoted via the scheme MaDL(P1, P2). Property P1 refers to the estimation of potential class-dependent APs. There, we differentiate between the options class-independent (I), partially (P) class-dependent, and fully (F) class-dependent APs. We implement them by constraining the annotator confusion matrices' degrees of freedom. Concretely, class-independent refers to a confusion matrix approximated by estimating a single scalar, partially class-dependent refers to a confusion matrix approximated by estimating its diagonal elements, and fully class-dependent refers to estimating each matrix element individually (cf. Appendix G). Property P2 indicates whether the APs are estimated as a function of instances (X) or not ($\overline{\text{X}}$). Combining the two options of the properties P1 and P2 represents one variant. For example, MaDL(X, F) is the default MaDL variant estimating instance- and fully class-dependent APs.

**Qualitative study:** Fig. 6 visualizes MaDL's predictive behavior for the artificial dataset TOY. Thereby, each row represents the predictions of a different MaDL variant. Since this is a binary classification problem, the variant MaDL(X, P) is identical to MaDL(X, F), and MaDL($\overline{\text{X}}$, P) is identical to MaDL($\overline{\text{X}}$, F). The first column visualizes instances as circles colored according to their GT labels, plots the class-membership probabilities predicted by the respective GT model as contours across the feature space, and depicts the decision boundary for classification as a black line. The last four columns show the class labels provided by four of the ten simulated annotators. The instances' colors indicate the class labels provided by an annotator, their forms mark whether the class labels are correct (circle) or false (cross) annotations, and the contours

across the feature space visualize the AP model's predicted annotation correctness probabilities. The GT models of the variants MaDL($\overline{X}$, F), MaDL(X, I), and MaDL(X, F) successfully separate the instances of both classes, whereas the GT model of MaDL($\overline{X}$, I) fails in this task. Likely, the missing consideration of instance- and class-dependent APs explains this observation. Further, the class-membership probabilities of the successful MaDL variants reflect instances' actual class labels but exhibit the overconfident behavior typical of deterministic (D)NNs, particularly for feature space regions without observed instances (Huseljic et al., 2021). Investigating the estimated APs for the adversarial annotator (second column), we see that each MaDL variant correctly predicts low APs (indicated by the white-colored contours) across the feature space. When comparing the AP estimates for the class-specialized annotator (fifth column), clear differences between MaDL($\overline{X}$, I) and the other three variants of MaDL are visible. Since MaDL($\overline{X}$, I) ignores any class dependency regarding APs, it cannot differentiate between classes of high and low APs. In contrast, the AP predictions of the other three variants reflect the class structure learned by the respective GT model and thus can separate between weak and expert classes. The performances of the cluster-specialized and common annotator depend on the regions in the feature space. Therefore, only the variants MaDL(X, I) and MaDL(X, F) can separate clusters of low and high APs. For example, both variants successfully identify the two weak clusters of the cluster-specialized annotator. Analogous to the class-membership probabilities, the AP estimates are overconfident for feature space regions without observed instances.

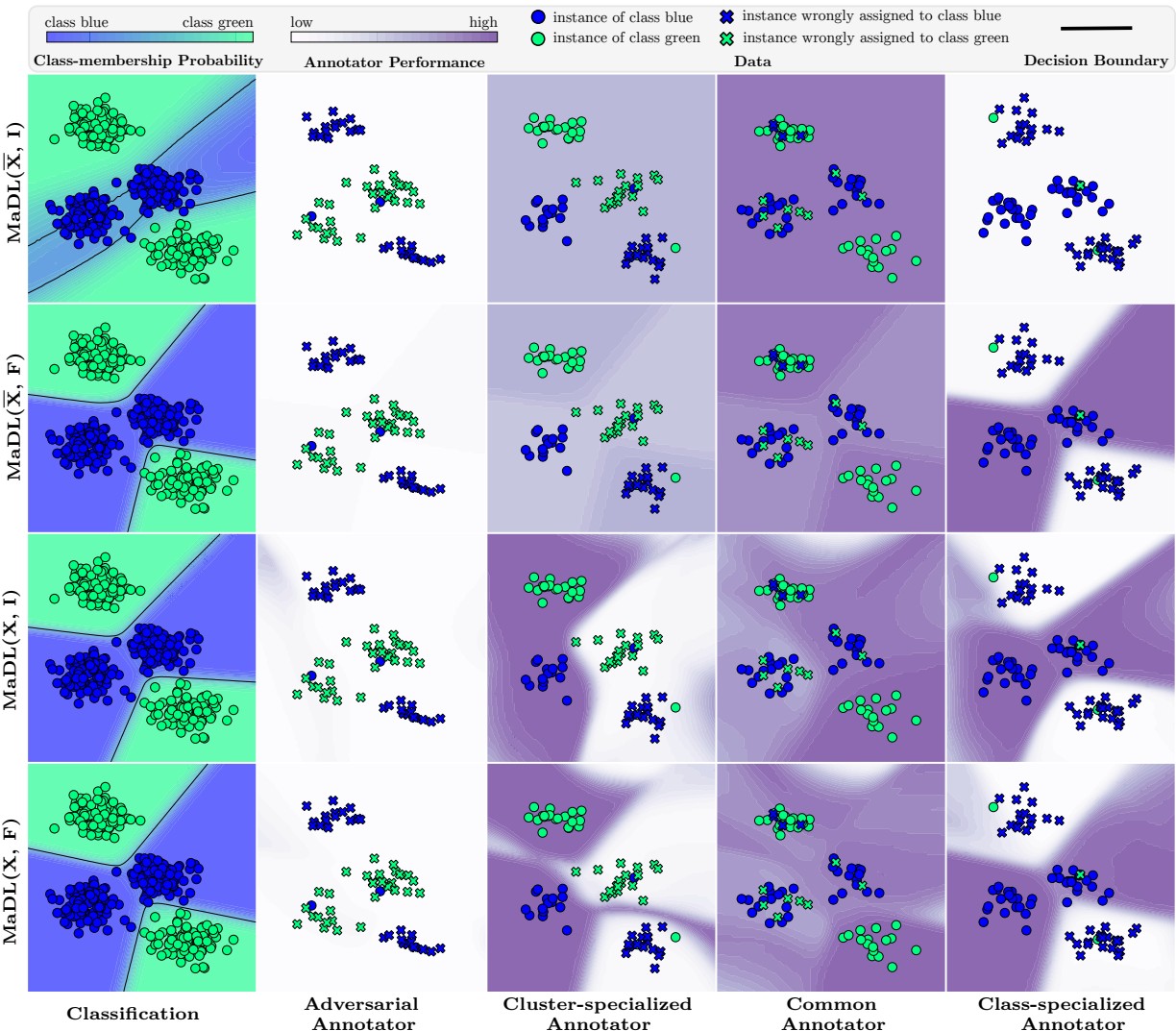

Figure 6: Visualization of MaDL's predictive behavior for the two-dimensional dataset TOY.

**Quantitative study:** Table 4 presents the numerical evaluation results for the two datasets with real-world annotators. There, we only report the GT models' test results since no annotations for the test instances are available to assess the AP models' test performances. Table 5 presents the GT and AP models' test results for the four datasets with simulated annotators. Both tables indicate whether a technique models class-dependent (property P1) and/or instance-dependent (property P2) APs. Generally, training with GT labels as UB achieves the best performances, while the LB with annotations aggregated according to the majority rule leads to the worst ones. The latter observation confirms that leveraging AP estimates during training is beneficial. Moreover, these AP estimates are typically meaningful, corresponding to BAL-ACC values above 0.5. An exception is MaDL($\overline{\mathrm{X}}$, I) because this variant only estimates by design a constant performance per annotator across the feature space. Comparing MaDL(X, F) as the most general variant to related techniques, we observe that it achieves competitive or superior results for all datasets and evaluation scores. Next to MaDL(X, F), CoNAL often delivers better results than the competitors. When we investigate the performances of the MaDL variants with instance-independent APs, we find that MaDL($\overline{\mathrm{X}}$, F) achieves the most robust performances across all datasets. In particular, for the datasets with real-world annotators, the ACC of the respective GT model is superior. This observation suggests that modeling class-dependent APs (property P1) is beneficial. We recognize a similar trend for the MaDL variants with instance-dependent APs (property P2). Comparing each pair of MaDL variants with X and $\overline{\mathrm{X}}$, we observe that instance-dependent APs often improve GT and, in particular, AP estimates. The advantage of class- and instance-dependent APs is confirmed by CoNAL as a strong competitor of MaDL(X, F). LIA's inferior performance contrasts this, although LIA estimates class- and instance-dependent APs. The difference in training algorithms can likely explain this observation. While MaDL(X, F) and CoNAL train via an end-to-end algorithm, LIA trains via the EM algorithm, leading to higher runtimes and introducing additional sensitive hyperparameters, e.g., the number of EM iterations and training epochs per M step.

Table 4: Results regarding RQ1 for datasets with real-world annotators: Best and second best performances are highlighted per dataset and evaluation score while excluding the performances of the UB.

| Technique | P1 | P2 | Ground Truth Model | | | Ground Truth Model | | |
| --- | --- | --- | --- | --- | --- | --- | --- | --- |
| | | | ACC ↑ | NLL ↓ | BS ↓ | ACC ↑ | NLL ↓ | BS ↓ |
| | | | MUSIC | | | LABELME | | |
| UB | ✓ | ✓ | 0.785±0.020 | 0.710±0.037 | 0.314±0.027 | 0.914±0.003 | 0.580±0.112 | 0.150±0.003 |
| LB | ✓ | ✓ | 0.646±0.045 | 1.096±0.103 | 0.492±0.051 | 0.810±0.015 | 0.724±0.155 | 0.294±0.024 |
| CL | ✓ | ✗ | 0.675±0.015 | 1.672±0.400 | 0.524±0.021 | 0.857±0.011 | 1.774±1.155 | 0.250±0.014 |
| REAC | ✓ | ✗ | 0.705±0.023 | 0.893±0.081 | 0.410±0.033 | 0.843±0.006 | 0.833±0.088 | 0.254±0.006 |
| UNION | ✓ | ✗ | 0.682±0.013 | 1.396±0.143 | 0.501±0.027 | 0.855±0.004 | 1.074±0.340 | 0.248±0.011 |
| LIA | ✓ | ✓ | 0.658±0.023 | 1.158±0.047 | 0.498±0.020 | 0.813±0.010 | 0.976±0.234 | 0.295±0.009 |
| CoNAL | ✓ | ✓ | 0.708±0.031 | 0.964±0.081 | 0.423±0.035 | 0.866±0.004 | 2.740±1.304 | 0.247±0.023 |
| MaDL($\overline{\mathrm{X}}$, I) | ✗ | ✗ | 0.718±0.010 | 0.871±0.027 | 0.394±0.009 | 0.815±0.009 | 0.616±0.125 | 0.276±0.017 |
| MaDL($\overline{\mathrm{X}}$, P) | ✦ | ✗ | 0.720±0.018 | 0.871±0.030 | 0.396±0.009 | 0.811±0.012 | 0.630±0.128 | 0.281±0.022 |
| MaDL($\overline{\mathrm{X}}$, F) | ✓ | ✗ | 0.725±0.015 | 0.977±0.064 | 0.403±0.019 | 0.859±0.007 | 1.008±0.278 | 0.240±0.014 |
| MaDL(X, I) | ✗ | ✓ | 0.713±0.027 | 0.876±0.041 | 0.402±0.022 | 0.816±0.008 | 0.559±0.027 | 0.276±0.010 |
| MaDL(X, P) | ✦ | ✓ | 0.714±0.014 | 0.909±0.036 | 0.398±0.013 | 0.811±0.009 | 0.771±0.160 | 0.289±0.016 |
| MaDL(X, F) | ✓ | ✓ | 0.743±0.018 | 0.877±0.030 | 0.381±0.012 | 0.867±0.004 | 0.623±0.124 | 0.214±0.008 |

### 5.3 RQ2: Does modeling correlations between (potentially spamming) annotators improve learning? (Properties P3, P4)

> **Takeaway:** Modeling correlations between annotators leads to better results in scenarios with many correlated spamming annotators (property P4). Capturing the correlations of beneficial annotators does not lead to consistently better results (property P3). However, estimating and leveraging APs during training becomes more critical in scenarios with correlated annotators.

**Setup:** We address RQ2 by evaluating multi-annotator supervised learning techniques with and without modeling annotator correlations. We simulate two annotator sets for each dataset without real-world annotators according to Table 3. The first annotator set CORRELATED consists of the same ten annotators as in RQ1. However, we extend this set by ten additional copies of the adversarial, the class-specialized, and one of the two cluster-specialized annotators, so there are 40 annotators. The second annotator set

RANDOM-CORRELATED also consists of the same ten annotators as in RQ1 but is extended by 90 identical randomly guessing annotators. Each simulated annotator provides class labels for 20 % of randomly selected training instances. Next to the related multi-annotator supervised learning techniques and the two baselines, we evaluate two variants of MaDL denoted via the scheme MaDL(P3). Property P3 refers to the modeling of potential annotator correlations. There, we differentiate between the variant MaDL(W) using annotator weights via the weighted loss function (cf. Eq. 25) and the variant MaDL($\overline{\text{W}}$) training via the loss function without any weights (cf. Eq. 15). MaDL(W) corresponds to MaDL's default variant in this setup.

Table 5: Results regarding RQ1 for datasets with simulated annotators: Best and second best performances are highlighted per dataset and evaluation score while excluding the performances of the UB.

| Technique | P1 | P2 | Ground Truth Model | | | Annotator Performance Model | | | |
|---|---|---|---|---|---|---|---|---|---|
| | | | ACC ↑ | NLL ↓ | BS ↓ | ACC ↑ | NLL ↓ | BS ↓ | BAL-ACC ↑ |
| LETTER (INDEPENDENT) | | | | | | | | | |
| UB | ✓ | ✓ | 0.961±0.003 | 0.130±0.006 | 0.059±0.004 | 0.770±0.001 | 0.488±0.003 | 0.315±0.002 | 0.709±0.001 |
| LB | ✓ | ✓ | 0.878±0.004 | 0.980±0.021 | 0.385±0.008 | 0.664±0.004 | 0.624±0.003 | 0.433±0.003 | 0.666±0.004 |
| CL | ✓ | ✗ | 0.886±0.013 | 1.062±0.145 | 0.181±0.020 | 0.663±0.006 | 0.625±0.013 | 0.430±0.010 | 0.601±0.002 |
| REAC | ✓ | ✗ | 0.936±0.005 | 0.238±0.018 | 0.097±0.007 | 0.685±0.002 | 0.560±0.001 | 0.385±0.001 | 0.604±0.002 |
| UNION | ✓ | ✗ | 0.905±0.016 | 0.906±0.435 | 0.151±0.030 | 0.670±0.004 | 0.589±0.008 | 0.408±0.006 | 0.605±0.002 |
| LIA | ✓ | ✓ | 0.897±0.005 | 0.778±0.052 | 0.305±0.021 | 0.669±0.004 | 0.654±0.010 | 0.447±0.004 | 0.616±0.003 |
| CoNAL | ✓ | ✓ | 0.907±0.016 | 0.813±0.354 | 0.143±0.027 | 0.723±0.018 | 0.555±0.024 | 0.372±0.020 | 0.663±0.017 |
| MaDL($\overline{\text{X}}$, I) | ✗ | ✗ | 0.934±0.003 | 0.269±0.035 | 0.100±0.004 | 0.607±0.001 | 0.627±0.000 | 0.444±0.000 | 0.500±0.000 |
| MaDL($\overline{\text{X}}$, P) | ✦ | ✗ | 0.935±0.005 | 0.235±0.013 | 0.099±0.006 | 0.692±0.001 | 0.556±0.001 | 0.381±0.001 | 0.606±0.003 |
| MaDL($\overline{\text{X}}$, F) | ✓ | ✗ | 0.933±0.005 | 0.255±0.025 | 0.100±0.005 | 0.691±0.002 | 0.556±0.001 | 0.381±0.001 | 0.606±0.002 |
| MaDL(X, I) | ✗ | ✓ | 0.938±0.006 | 0.247±0.043 | 0.092±0.008 | 0.770±0.004 | 0.492±0.016 | 0.316±0.007 | 0.708±0.004 |
| MaDL(X, P) | ✦ | ✓ | 0.940±0.004 | 0.242±0.045 | 0.090±0.004 | 0.770±0.006 | 0.496±0.020 | 0.316±0.009 | 0.708±0.005 |
| MaDL(X, F) | ✓ | ✓ | 0.935±0.006 | 0.303±0.092 | 0.098±0.009 | 0.766±0.004 | 0.491±0.006 | 0.317±0.004 | 0.702±0.005 |
| FMNIST (INDEPENDENT) | | | | | | | | | |
| UB | ✓ | ✓ | 0.909±0.002 | 0.246±0.005 | 0.131±0.003 | 0.756±0.001 | 0.485±0.001 | 0.321±0.001 | 0.704±0.001 |
| LB | ✓ | ✓ | 0.883±0.001 | 0.903±0.003 | 0.385±0.001 | 0.644±0.007 | 0.645±0.005 | 0.453±0.004 | 0.585±0.007 |
| CL | ✓ | ✗ | 0.892±0.002 | 0.312±0.008 | 0.158±0.004 | 0.674±0.002 | 0.580±0.001 | 0.402±0.001 | 0.623±0.001 |
| REAC | ✓ | ✗ | 0.894±0.003 | 0.309±0.011 | 0.155±0.004 | 0.703±0.001 | 0.535±0.001 | 0.364±0.000 | 0.641±0.001 |
| UNION | ✓ | ✗ | 0.893±0.002 | 0.305±0.006 | 0.155±0.003 | 0.674±0.002 | 0.570±0.002 | 0.395±0.002 | 0.622±0.001 |
| LIA | ✓ | ✓ | 0.858±0.002 | 1.017±0.016 | 0.442±0.008 | 0.665±0.024 | 0.628±0.017 | 0.437±0.016 | 0.613±0.027 |
| CoNAL | ✓ | ✓ | 0.894±0.004 | 0.304±0.009 | 0.155±0.004 | 0.725±0.016 | 0.521±0.018 | 0.351±0.016 | 0.679±0.018 |
| MaDL($\overline{\text{X}}$, I) | ✗ | ✗ | 0.896±0.003 | 0.340±0.006 | 0.161±0.004 | 0.590±0.000 | 0.638±0.000 | 0.453±0.000 | 0.500±0.000 |
| MaDL($\overline{\text{X}}$, P) | ✦ | ✗ | 0.894±0.001 | 0.307±0.003 | 0.155±0.001 | 0.705±0.001 | 0.534±0.000 | 0.363±0.000 | 0.640±0.001 |
| MaDL($\overline{\text{X}}$, F) | ✓ | ✗ | 0.894±0.002 | 0.307±0.006 | 0.155±0.003 | 0.705±0.000 | 0.534±0.000 | 0.363±0.000 | 0.640±0.000 |
| MaDL(X, I) | ✗ | ✓ | 0.895±0.003 | 0.291±0.005 | 0.150±0.003 | 0.752±0.004 | 0.490±0.004 | 0.325±0.003 | 0.699±0.004 |
| MaDL(X, P) | ✦ | ✓ | 0.899±0.003 | 0.286±0.006 | 0.147±0.003 | 0.751±0.003 | 0.489±0.004 | 0.324±0.003 | 0.698±0.005 |
| MaDL(X, F) | ✓ | ✓ | 0.896±0.002 | 0.288±0.006 | 0.148±0.003 | 0.750±0.005 | 0.491±0.005 | 0.326±0.005 | 0.697±0.006 |
| CIFAR10 (INDEPENDENT) | | | | | | | | | |
| UB | ✓ | ✓ | 0.933±0.002 | 0.519±0.026 | 0.118±0.004 | 0.710±0.001 | 0.547±0.001 | 0.369±0.001 | 0.658±0.001 |
| LB | ✓ | ✓ | 0.789±0.004 | 1.081±0.031 | 0.460±0.015 | 0.575±0.021 | 0.673±0.006 | 0.481±0.006 | 0.547±0.011 |
| CL | ✓ | ✗ | 0.833±0.003 | 0.536±0.012 | 0.242±0.004 | 0.664±0.001 | 0.604±0.002 | 0.420±0.001 | 0.613±0.001 |
| REAC | ✓ | ✗ | 0.839±0.003 | 0.581±0.010 | 0.245±0.003 | 0.676±0.003 | 0.580±0.006 | 0.397±0.004 | 0.625±0.002 |
| UNION | ✓ | ✗ | 0.834±0.003 | 0.595±0.022 | 0.249±0.005 | 0.668±0.001 | 0.592±0.001 | 0.410±0.001 | 0.617±0.002 |
| LIA | ✓ | ✓ | 0.805±0.003 | 1.102±0.035 | 0.469±0.016 | 0.622±0.024 | 0.645±0.014 | 0.453±0.014 | 0.579±0.019 |
| CoNAL | ✓ | ✓ | 0.838±0.005 | 0.530±0.021 | 0.236±0.008 | 0.668±0.001 | 0.600±0.001 | 0.416±0.001 | 0.616±0.001 |
| MaDL($\overline{\text{X}}$, I) | ✗ | ✗ | 0.832±0.006 | 0.583±0.021 | 0.256±0.009 | 0.576±0.010 | 0.646±0.002 | 0.461±0.002 | 0.500±0.000 |
| MaDL($\overline{\text{X}}$, P) | ✦ | ✗ | 0.844±0.004 | 0.529±0.014 | 0.231±0.004 | 0.682±0.001 | 0.568±0.001 | 0.390±0.001 | 0.630±0.002 |
| MaDL($\overline{\text{X}}$, F) | ✓ | ✗ | 0.840±0.005 | 0.545±0.019 | 0.237±0.006 | 0.681±0.001 | 0.569±0.002 | 0.390±0.001 | 0.630±0.001 |
| MaDL(X, I) | ✗ | ✓ | 0.843±0.005 | 0.555±0.024 | 0.236±0.008 | 0.697±0.002 | 0.559±0.005 | 0.380±0.003 | 0.646±0.002 |
| MaDL(X, P) | ✦ | ✓ | 0.845±0.002 | 0.546±0.027 | 0.232±0.005 | 0.697±0.001 | 0.557±0.002 | 0.380±0.001 | 0.646±0.002 |
| MaDL(X, F) | ✓ | ✓ | 0.846±0.003 | 0.521±0.014 | 0.229±0.005 | 0.697±0.002 | 0.557±0.004 | 0.379±0.002 | 0.646±0.003 |
| SVHN (INDEPENDENT) | | | | | | | | | |
| UB | ✓ | ✓ | 0.965±0.000 | 0.403±0.024 | 0.064±0.001 | 0.675±0.002 | 0.567±0.001 | 0.392±0.001 | 0.590±0.004 |
| LB | ✓ | ✓ | 0.930±0.002 | 0.811±0.030 | 0.332±0.015 | 0.581±0.021 | 0.680±0.008 | 0.487±0.008 | 0.540±0.000 |
| CL | ✓ | ✗ | 0.944±0.001 | 0.237±0.008 | 0.085±0.002 | 0.646±0.001 | 0.598±0.001 | 0.419±0.001 | 0.546±0.001 |
| REAC | ✓ | ✗ | 0.943±0.001 | 0.278±0.048 | 0.096±0.020 | 0.648±0.006 | 0.593±0.015 | 0.414±0.010 | 0.543±0.000 |
| UNION | ✓ | ✗ | 0.942±0.002 | 0.250±0.005 | 0.087±0.001 | 0.646±0.001 | 0.594±0.001 | 0.416±0.000 | 0.544±0.001 |
| LIA | ✓ | ✓ | 0.935±0.002 | 0.809±0.162 | 0.333±0.081 | 0.585±0.016 | 0.667±0.023 | 0.476±0.021 | 0.536±0.004 |
| CoNAL | ✓ | ✓ | 0.944±0.002 | 0.246±0.012 | 0.086±0.002 | 0.688±0.036 | 0.560±0.029 | 0.384±0.026 | 0.602±0.050 |
| MaDL($\overline{\text{X}}$, I) | ✗ | ✗ | 0.942±0.003 | 0.253±0.023 | 0.093±0.008 | 0.613±0.003 | 0.630±0.003 | 0.446±0.003 | 0.500±0.000 |
| MaDL($\overline{\text{X}}$, P) | ✦ | ✗ | 0.940±0.002 | 0.262±0.011 | 0.091±0.003 | 0.652±0.000 | 0.585±0.000 | 0.408±0.000 | 0.544±0.000 |
| MaDL($\overline{\text{X}}$, F) | ✓ | ✗ | 0.940±0.002 | 0.264±0.007 | 0.092±0.002 | 0.652±0.001 | 0.585±0.000 | 0.408±0.000 | 0.543±0.001 |
| MaDL(X, I) | ✗ | ✓ | 0.944±0.003 | 0.240±0.007 | 0.085±0.003 | 0.665±0.001 | 0.575±0.001 | 0.399±0.001 | 0.565±0.001 |
| MaDL(X, P) | ✦ | ✓ | 0.945±0.002 | 0.245±0.010 | 0.084±0.004 | 0.669±0.002 | 0.572±0.002 | 0.396±0.002 | 0.573±0.005 |
| MaDL(X, F) | ✓ | ✓ | 0.943±0.001 | 0.254±0.013 | 0.087±0.002 | 0.668±0.003 | 0.572±0.003 | 0.396±0.003 | 0.570±0.006 |

**Qualitative study:** Fig. 7 visualizes MaDL(W)'s learned annotator embeddings and weights for the dataset LETTER with the two annotator sets, CORRELATED and RANDOM-CORRELATED, after five training epochs. Based on MaDL(W)'s learned kernel function, we create the two scatter plots via multi-dimensional scaling (Kruskal, 1964) for dimensionality reduction. This way, the annotator embeddings, originally located in an ($R = 16$)-dimensional space, are transformed into a two-dimensional space, where each circle represents one annotator embedding. A circle's color indicates to which annotator group the embedding belongs. The two bar plots visualize the mean annotator weight of the different annotator groups, again indicated by their respective color. Analyzing the scatter plot of the annotator set CORRELATED, we observe that the annotator embeddings' latent representations approximately reflect the annotator groups' correlations. Concretely, there are four clusters. The center cluster corresponds to the seven independent annotators, one cluster-specialized annotator and six common annotators. The three clusters in the outer area represent the three groups of correlated annotators. The bar plot confirms our goal to assign lower weights to strongly correlated annotators. For example, the single independent cluster-specialized annotator has a weight of 4.06, while the eleven correlated cluster-specialized annotators have a mean weight of 0.43. We make similar observations for the annotator set RANDOM-CORRELATED. The scatter plot shows that the independent annotators also form a cluster, separated from the cluster of the large group of correlated, randomly guessing annotators. The single adversarial annotator belongs to the cluster of randomly guessing annotators since both groups of annotators make many annotation errors and thus have highly correlated annotation patterns. Again, the bar plot confirms that the correlated annotators get low weights. Moreover, these annotator weights are inversely proportional to the size of a group of correlated annotators. For example, the 90 randomly guessing annotators have a similar weight in sum as the single class-specialized annotator.

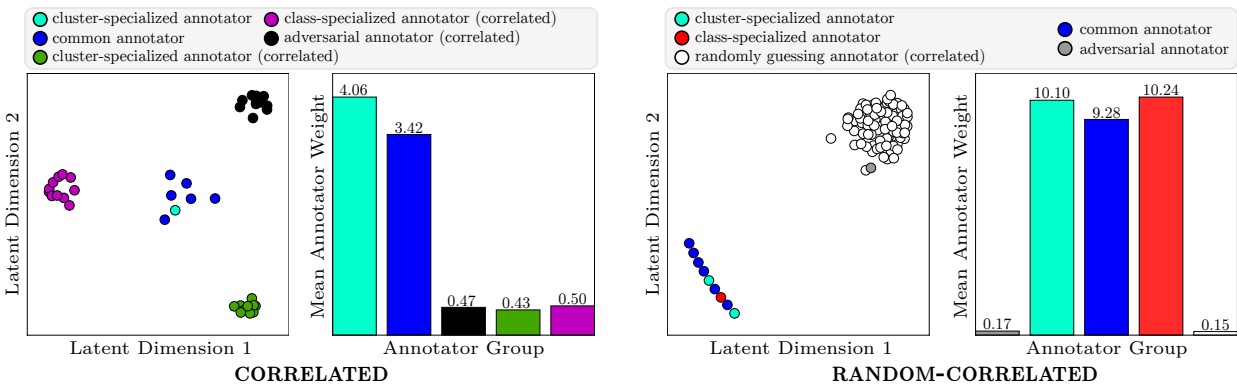

Figure 7: Visualization of MaDL(W)'s learned similarities between annotator embeddings and associated annotator weights.

**Quantitative study:** Table 6 presents the GT and AP models' test performances for the four datasets with the annotator set CORRELATED and Table 7 for the annotator set RANDOM-CORRELATED. Both tables indicate whether a technique models correlations between annotators (property P3) and whether the authors of a technique demonstrated its robustness against spamming annotators (property P4). Analogous to RQ1, training with GT labels achieves the best performances (UB), while annotation aggregation via the majority rule leads to the worst ones (LB). The LB's significant underperformance confirms the importance of modeling APs in scenarios with correlated annotators. MaDL(W), as the default MaDL variant, achieves competitive and often superior results for all datasets and evaluation scores. In particular, for the annotator set RANDOM-CORRELATED, MaDL(W) outperforms the other techniques, which are vulnerable to many randomly guessing annotators. This observation is also confirmed when we compare MaDL(W) to MaDL($\overline{\overline{W}}$). In contrast, there is no consistent performance gain of MaDL(W) over MaDL($\overline{W}$) for the annotator set CORRELATED. While CoNAL is competitive for the annotator set CORRELATED, its performance strongly degrades for the annotator set RANDOM-CORRELATED. The initial E step in LIA's EM algorithm estimates the GT class labels via a probabilistic variant of the majority rule. Similarly to the LB, such an estimate is less accurate for correlated and/or spamming annotators. Besides MaDL(W), only CL and UNION consistently outperform the LB by large margins for the annotator set RANDOM-CORRELATED.

Table 6: Results regarding RQ2 for datasets with simulated annotators: Best and second best performances are highlighted per dataset and evaluation score while excluding the performances of the UB.

| Technique | P3 | P4 | Ground Truth Model | | | Annotator Performance Model | | | |
|---|---|---|---|---|---|---|---|---|---|
| | | | ACC ↑ | NLL ↓ | BS ↓ | ACC ↑ | NLL ↓ | BS ↓ | BAL-ACC ↑ |
| LETTER (CORRELATED) | | | | | | | | | |
| UB | ✗ | ✓ | 0.962±0.004 | 0.129±0.004 | 0.058±0.003 | 0.887±0.002 | 0.305±0.004 | 0.173±0.002 | 0.757±0.002 |
| LB | ✗ | ✗ | 0.762±0.007 | 1.302±0.005 | 0.482±0.004 | 0.682±0.005 | 0.604±0.003 | 0.416±0.002 | 0.602±0.006 |
| CL | ✗ | ✗ | 0.803±0.035 | 2.435±1.218 | 0.318±0.057 | 0.800±0.008 | 0.446±0.016 | 0.285±0.012 | 0.674±0.007 |
| REAC | ✗ | ✗ | 0.922±0.003 | 0.288±0.065 | 0.115±0.007 | 0.815±0.001 | 0.395±0.001 | 0.249±0.001 | 0.684±0.001 |
| UNION | ✓ | ✗ | 0.866±0.019 | 1.668±0.322 | 0.224±0.034 | 0.795±0.007 | 0.432±0.007 | 0.278±0.007 | 0.667±0.006 |
| LIA | ✗ | ✗ | 0.823±0.005 | 1.483±0.018 | 0.569±0.007 | 0.676±0.005 | 0.629±0.004 | 0.436±0.004 | 0.575±0.004 |
| CoNAL | ✓ | ✓ | 0.871±0.015 | 1.380±0.349 | 0.213±0.024 | 0.840±0.014 | 0.390±0.028 | 0.238±0.021 | 0.712±0.014 |
| MaDL($\overline{\text{W}}$) | ✗ | ✗ | 0.946±0.006 | 0.293±0.082 | 0.083±0.009 | 0.883±0.002 | 0.314±0.001 | 0.178±0.002 | 0.751±0.003 |
| MaDL(W) | ✓ | ✓ | 0.947±0.003 | 0.282±0.069 | 0.080±0.004 | 0.887±0.001 | 0.308±0.004 | 0.175±0.002 | 0.756±0.001 |
| FMNIST (CORRELATED) | | | | | | | | | |
| UB | ✗ | ✓ | 0.909±0.002 | 0.246±0.005 | 0.131±0.003 | 0.866±0.002 | 0.333±0.002 | 0.198±0.002 | 0.741±0.002 |
| LB | ✗ | ✗ | 0.787±0.003 | 1.127±0.013 | 0.475±0.007 | 0.668±0.009 | 0.626±0.006 | 0.436±0.006 | 0.580±0.005 |
| CL | ✗ | ✗ | 0.868±0.003 | 0.447±0.020 | 0.217±0.010 | 0.799±0.004 | 0.421±0.004 | 0.270±0.003 | 0.677±0.004 |
| REAC | ✗ | ✗ | 0.873±0.004 | 0.415±0.012 | 0.196±0.006 | 0.828±0.001 | 0.382±0.001 | 0.237±0.001 | 0.697±0.001 |
| UNION | ✓ | ✗ | 0.859±0.006 | 0.411±0.018 | 0.205±0.008 | 0.801±0.009 | 0.420±0.014 | 0.269±0.011 | 0.678±0.009 |
| LIA | ✗ | ✗ | 0.837±0.006 | 1.277±0.008 | 0.553±0.004 | 0.685±0.002 | 0.633±0.001 | 0.441±0.001 | 0.569±0.002 |
| CoNAL | ✓ | ✓ | 0.897±0.002 | 0.299±0.009 | 0.152±0.004 | 0.844±0.001 | 0.356±0.003 | 0.217±0.002 | 0.721±0.001 |
| MaDL($\overline{\text{W}}$) | ✗ | ✗ | 0.904±0.002 | 0.272±0.007 | 0.139±0.003 | 0.863±0.003 | 0.337±0.004 | 0.201±0.004 | 0.737±0.004 |
| MaDL(W) | ✓ | ✓ | 0.903±0.002 | 0.273±0.004 | 0.141±0.002 | 0.863±0.003 | 0.338±0.003 | 0.202±0.003 | 0.738±0.003 |
| CIFAR10 (CORRELATED) | | | | | | | | | |
| UB | ✗ | ✓ | 0.933±0.002 | 0.495±0.017 | 0.118±0.003 | 0.837±0.001 | 0.384±0.001 | 0.235±0.001 | 0.711±0.001 |
| LB | ✗ | ✗ | 0.652±0.014 | 1.309±0.016 | 0.540±0.008 | 0.602±0.011 | 0.623±0.003 | 0.436±0.003 | 0.541±0.008 |
| CL | ✗ | ✗ | 0.850±0.007 | 0.490±0.022 | 0.224±0.011 | 0.799±0.002 | 0.439±0.004 | 0.282±0.003 | 0.674±0.002 |
| REAC | ✗ | ✗ | 0.856±0.003 | 0.600±0.063 | 0.259±0.025 | 0.775±0.017 | 0.445±0.015 | 0.287±0.012 | 0.648±0.017 |
| UNION | ✓ | ✗ | 0.858±0.007 | 0.499±0.024 | 0.211±0.009 | 0.800±0.003 | 0.432±0.002 | 0.276±0.002 | 0.675±0.003 |
| LIA | ✗ | ✗ | 0.776±0.002 | 1.343±0.020 | 0.565±0.009 | 0.741±0.002 | 0.617±0.003 | 0.424±0.003 | 0.617±0.002 |
| CoNAL | ✓ | ✓ | 0.862±0.002 | 0.473±0.005 | 0.213±0.003 | 0.800±0.001 | 0.433±0.003 | 0.277±0.002 | 0.676±0.001 |
| MaDL($\overline{\text{W}}$) | ✗ | ✗ | 0.878±0.004 | 0.439±0.015 | 0.184±0.005 | 0.824±0.004 | 0.398±0.004 | 0.247±0.004 | 0.699±0.004 |
| MaDL(W) | ✓ | ✓ | 0.875±0.008 | 0.434±0.020 | 0.188±0.011 | 0.823±0.002 | 0.397±0.003 | 0.248±0.002 | 0.698±0.002 |
| SVHN (CORRELATED) | | | | | | | | | |
| UB | ✗ | ✓ | 0.966±0.001 | 0.382±0.018 | 0.062±0.001 | 0.794±0.003 | 0.414±0.002 | 0.266±0.002 | 0.657±0.004 |
| LB | ✗ | ✗ | 0.900±0.005 | 1.012±0.038 | 0.420±0.017 | 0.624±0.022 | 0.634±0.008 | 0.444±0.007 | 0.567±0.017 |
| CL | ✗ | ✗ | 0.947±0.001 | 0.314±0.044 | 0.116±0.017 | 0.789±0.009 | 0.433±0.001 | 0.281±0.002 | 0.655±0.012 |
| REAC | ✗ | ✗ | 0.946±0.002 | 0.263±0.012 | 0.097±0.005 | 0.767±0.002 | 0.431±0.001 | 0.283±0.000 | 0.620±0.003 |
| UNION | ✓ | ✗ | 0.947±0.001 | 0.250±0.025 | 0.089±0.010 | 0.767±0.003 | 0.435±0.003 | 0.286±0.002 | 0.621±0.005 |
| LIA | ✗ | ✗ | 0.929±0.002 | 1.123±0.023 | 0.477±0.011 | 0.716±0.013 | 0.623±0.010 | 0.431±0.010 | 0.594±0.013 |
| CoNAL | ✓ | ✓ | 0.952±0.000 | 0.231±0.003 | 0.075±0.001 | 0.835±0.003 | 0.379±0.005 | 0.235±0.004 | 0.702±0.004 |
| MaDL($\overline{\text{W}}$) | ✗ | ✗ | 0.950±0.002 | 0.237±0.006 | 0.078±0.003 | 0.790±0.003 | 0.416±0.002 | 0.269±0.002 | 0.652±0.002 |
| MaDL(W) | ✓ | ✓ | 0.952±0.001 | 0.227±0.006 | 0.075±0.002 | 0.784±0.003 | 0.420±0.002 | 0.273±0.002 | 0.645±0.004 |

## 5.4 RQ3: Do annotator features containing prior information about annotators improve learning and enable inductively learning annotators' performances? (Properties P5, P6)

**Takeaway:** Annotator features containing prior information about annotators improve the learning of GT and AP models (property P5). Furthermore, we can use these annotator features to inductively estimate the performances of annotators unavailable during training (property P6).

**Setup:** We address RQ3 by evaluating multi-annotator supervised learning techniques with and without using annotator features containing prior information. For each dataset, we simulate 100 annotators according to the annotator set INDUCTIVE in Table 3. However, only 75 annotators provide class labels for training. Each of them provides class labels for 2 % of randomly selected training instances. The lower annotation ratio is used to study the generalization across annotators sharing similar features. The remaining 25 annotators form a test set to assess AP predictions. We generate annotator features containing prior information by composing information about annotator type, class-wise APs, and cluster-wise APs. Fig. 8 provides examples for two annotators based on two classes and four clusters. We evaluate two variants of LIA, CoNAL, and MaDL, denoted respectively by the schemes LIA(P5), CoNAL(P5), and MaDL(P5). Property P5 refers to a technique's ability to consider prior information about annotators. We differentiate between the variant with annotator features containing prior information (A) and the one using one-hot encoded features to separate

Table 7: Results regarding RQ2 for datasets with simulated annotators: Best and second best performances are highlighted per dataset and evaluation score while excluding the performances of the UB.

| Technique | P3 | P4 | Ground Truth Model | | | Annotator Performance Model | | | |
|---|---|---|---|---|---|---|---|---|---|
| | | | ACC ↑ | NLL ↓ | BS ↓ | ACC ↑ | NLL ↓ | BS ↓ | BAL-ACC ↑ |
| LETTER (RANDOM-CORRELATED) | | | | | | | | | |
| UB | ✗ | ✓ | 0.960±0.003 | 0.131±0.006 | 0.059±0.003 | 0.937±0.002 | 0.212±0.003 | 0.104±0.002 | 0.516±0.002 |
| LB | ✗ | ✗ | 0.056±0.009 | 3.307±0.049 | 0.965±0.004 | 0.088±0.000 | 9.950±2.090 | 1.816±0.002 | 0.500±0.000 |
| CL | ✗ | ✗ | 0.565±0.028 | 3.519±0.455 | 0.682±0.052 | 0.925±0.000 | 0.237±0.004 | 0.124±0.002 | 0.506±0.000 |
| REAC | ✗ | ✗ | 0.607±0.024 | 1.810±0.127 | 0.561±0.034 | 0.926±0.000 | 0.221±0.004 | 0.116±0.002 | 0.507±0.000 |
| UNION | ✓ | ✗ | 0.615±0.034 | 3.317±0.582 | 0.625±0.065 | 0.925±0.000 | 0.232±0.004 | 0.122±0.002 | 0.506±0.000 |
| LIA | ✗ | ✗ | 0.352±0.010 | 2.960±0.035 | 0.932±0.004 | 0.088±0.000 | 2.131±0.137 | 1.474±0.041 | 0.500±0.000 |
| CoNAL | ✓ | ✓ | 0.581±0.015 | 2.325±0.249 | 0.599±0.027 | 0.925±0.000 | 0.236±0.002 | 0.124±0.001 | 0.507±0.000 |
| MaDL($\overline{\text{W}}$) | ✗ | ✗ | 0.548±0.033 | 1.902±0.215 | 0.673±0.064 | 0.801±0.044 | 0.423±0.033 | 0.265±0.027 | 0.506±0.006 |
| MaDL(W) | ✓ | ✓ | 0.932±0.003 | 0.277±0.038 | 0.101±0.005 | 0.940±0.000 | 0.204±0.003 | 0.101±0.001 | 0.519±0.001 |
| FMNIST (RANDOM-CORRELATED) | | | | | | | | | |
| UB | ✗ | ✓ | 0.909±0.002 | 0.246±0.005 | 0.131±0.003 | 0.888±0.000 | 0.337±0.001 | 0.191±0.000 | 0.520±0.000 |
| LB | ✗ | ✗ | 0.172±0.019 | 2.296±0.005 | 0.899±0.001 | 0.140±0.000 | 21.865±6.169 | 1.703±0.000 | 0.500±0.000 |
| CL | ✗ | ✗ | 0.880±0.003 | 0.462±0.169 | 0.222±0.073 | 0.880±0.003 | 0.347±0.004 | 0.200±0.003 | 0.513±0.002 |
| REAC | ✗ | ✗ | 0.870±0.003 | 0.470±0.009 | 0.204±0.004 | 0.885±0.000 | 0.342±0.000 | 0.194±0.000 | 0.514±0.000 |
| UNION | ✓ | ✗ | 0.884±0.002 | 0.387±0.022 | 0.182±0.007 | 0.881±0.000 | 0.345±0.000 | 0.198±0.000 | 0.514±0.000 |
| LIA | ✗ | ✗ | 0.677±0.008 | 2.094±0.002 | 0.852±0.001 | 0.140±0.000 | 2.067±0.005 | 1.418±0.002 | 0.500±0.000 |
| CoNAL | ✓ | ✓ | 0.858±0.012 | 0.457±0.086 | 0.219±0.031 | 0.882±0.002 | 0.344±0.002 | 0.197±0.002 | 0.516±0.001 |
| MaDL($\overline{\text{W}}$) | ✗ | ✗ | 0.337±0.046 | 2.131±0.090 | 0.855±0.029 | 0.229±0.075 | 1.038±0.146 | 0.814±0.128 | 0.498±0.002 |
| MaDL(W) | ✓ | ✓ | 0.896±0.002 | 0.290±0.003 | 0.150±0.002 | 0.889±0.000 | 0.337±0.000 | 0.191±0.000 | 0.520±0.000 |
| CIFAR10 (RANDOM-CORRELATED) | | | | | | | | | |
| UB | ✗ | ✓ | 0.932±0.002 | 0.519±0.016 | 0.119±0.004 | 0.886±0.000 | 0.340±0.002 | 0.192±0.001 | 0.515±0.000 |
| LB | ✗ | ✗ | 0.141±0.008 | 2.301±0.002 | 0.900±0.000 | 0.139±0.000 | 14.224±6.699 | 1.704±0.001 | 0.500±0.000 |
| CL | ✗ | ✗ | 0.576±0.023 | 1.395±0.090 | 0.576±0.028 | 0.878±0.000 | 0.353±0.002 | 0.204±0.001 | 0.507±0.000 |
| REAC | ✗ | ✗ | 0.462±0.010 | 2.093±0.062 | 0.767±0.011 | 0.875±0.001 | 0.353±0.000 | 0.204±0.000 | 0.505±0.001 |
| UNION | ✓ | ✗ | 0.540±0.049 | 1.517±0.209 | 0.629±0.065 | 0.876±0.002 | 0.355±0.003 | 0.205±0.002 | 0.506±0.002 |
| LIA | ✗ | ✗ | 0.211±0.014 | 2.273±0.007 | 0.894±0.001 | 0.139±0.000 | 2.096±0.007 | 1.429±0.002 | 0.500±0.000 |
| CoNAL | ✓ | ✓ | 0.555±0.020 | 1.379±0.053 | 0.592±0.020 | 0.876±0.001 | 0.355±0.002 | 0.206±0.002 | 0.506±0.001 |
| MaDL($\overline{\text{W}}$) | ✗ | ✗ | 0.217±0.042 | 6.992±0.386 | 1.219±0.087 | 0.872±0.001 | 0.398±0.011 | 0.229±0.009 | 0.502±0.001 |
| MaDL(W) | ✓ | ✓ | 0.822±0.007 | 0.593±0.033 | 0.262±0.010 | 0.885±0.000 | 0.339±0.001 | 0.192±0.001 | 0.514±0.000 |
| SVHN (RANDOM-CORRELATED) | | | | | | | | | |
| UB | ✗ | ✓ | 0.965±0.001 | 0.399±0.017 | 0.064±0.001 | 0.877±0.000 | 0.349±0.000 | 0.201±0.000 | 0.509±0.001 |
| LB | ✗ | ✗ | 0.190±0.000 | 2.298±0.002 | 0.899±0.000 | 0.138±0.000 | 24.019±7.802 | 1.704±0.001 | 0.500±0.000 |
| CL | ✗ | ✗ | 0.908±0.038 | 0.398±0.226 | 0.143±0.056 | 0.873±0.001 | 0.354±0.002 | 0.205±0.001 | 0.505±0.000 |
| REAC | ✗ | ✗ | 0.189±0.001 | 2.294±0.003 | 0.898±0.001 | 0.140±0.000 | 2.262±0.734 | 1.384±0.304 | 0.500±0.000 |
| UNION | ✓ | ✗ | 0.881±0.104 | 0.529±0.553 | 0.179±0.154 | 0.872±0.002 | 0.356±0.008 | 0.206±0.005 | 0.505±0.000 |
| LIA | ✗ | ✗ | 0.192±0.004 | 2.294±0.004 | 0.898±0.001 | 0.138±0.000 | 3.864±3.540 | 1.483±0.111 | 0.500±0.000 |
| CoNAL | ✓ | ✓ | 0.231±0.048 | 2.933±0.526 | 0.956±0.072 | 0.860±0.000 | 0.414±0.008 | 0.242±0.003 | 0.500±0.000 |
| MaDL($\overline{\text{W}}$) | ✗ | ✗ | 0.243±0.102 | 6.055±3.173 | 1.119±0.230 | 0.575±0.352 | 0.702±0.344 | 0.505±0.319 | 0.500±0.001 |
| MaDL(W) | ✓ | ✓ | 0.940±0.002 | 0.244±0.011 | 0.091±0.003 | 0.877±0.000 | 0.349±0.000 | 0.201±0.000 | 0.508±0.000 |

between annotators' identities ($\overline{\text{A}}$). MaDL($\overline{\text{A}}$) corresponds to MaDL's default variant in this setup. We do not evaluate CL, UNION, and REAC since these techniques cannot handle annotator features.

**Qualitative study:** Fig. 8 visualizes AP predictions of MaDL(A) regarding two exemplary annotators for the dataset TOY. The visualization of these AP predictions is analogous to Fig. 6. Neither of the two annotators provides class labels for the training, and the plotted training instances show only potential annotations to visualize the annotation patterns. The vectors at the right list the annotator features containing prior information for both annotators. The colors reveal the meanings of the respective feature values. These meanings are unknown to MaDL(A), such that its AP predictions exclusively result from generalizing similar annotators' features and their annotations available during training. MaDL(A) correctly identifies the left annotator as adversarial because it predicts low (white) AP scores across the feature space regions close to training instances. For the right cluster-specialized annotator, MaDL(A) accurately separates the two weak clusters (feature space regions with predominantly crosses) with low AP estimates from the two expert clusters (feature space regions with predominantly circles) with high AP estimates.

**Quantitative study:** Table 8 presents the GT and AP models' test performances for the four datasets with the simulated annotator set INDUCTIVE. The table further indicates whether a technique processes prior information as annotator features (property P5) and whether a technique can inductively estimate the performances of annotators unavailable during the training phase (property P6). Note that the AP results refer to the aforementioned 25 test annotators. Hence, there are no results (marked as –) for techniques

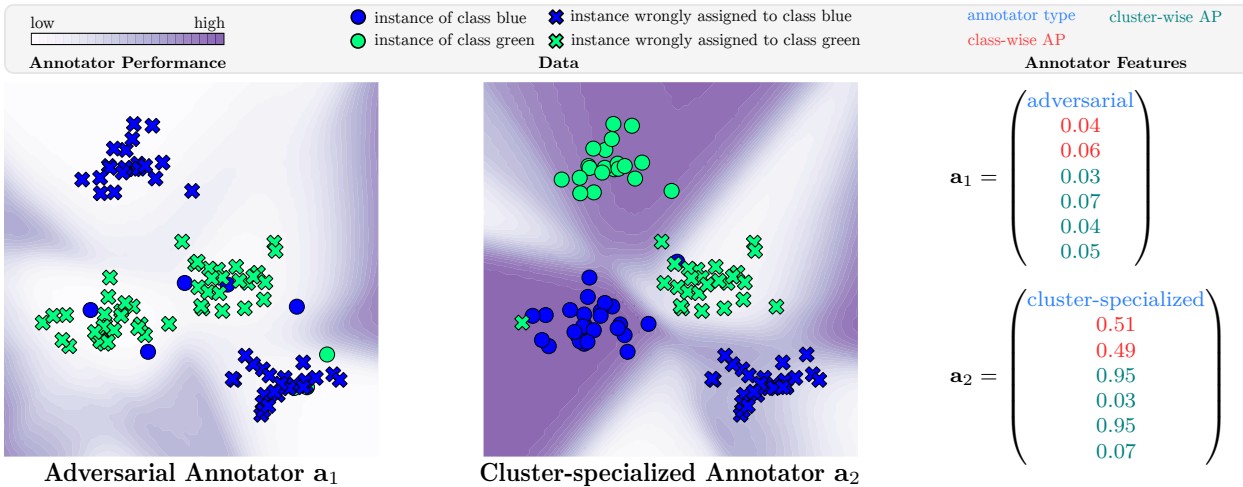

Figure 8: Visualization of MaDL(A)'s inductive AP estimates for two unknown annotators.

with AP models not fulfilling property P6. For completeness, we provide the results for the 75 annotators providing class labels for training in Appendix D. As for RQ1 and RQ2, training with GT labels leads to the best performance results (UB), whereas learning from annotations aggregated via the majority rule mostly results in the worst performances (LB). Inspecting the results of MaDL(A)'s GT model compared to the other techniques, we observe competitive or partially superior results across all four datasets. Concerning its AP model, we further note that MaDL(A) provides meaningful AP estimates, indicated by BAL-ACC values greater than 0.5. Comparing the GT models' results of each pair of variants, performance gains for LIA and MaDL demonstrate the potential benefits of learning from annotator features containing prior annotator information. In contrast, the GT models' results of CoNAL(A) and CoNAL($\overline{\text{A}}$) hardly differ.

## 6 Conclusion

In this article, we made three main contributions. (1) We started with a formalization of the objectives in multi-annotator supervised learning. Focusing on AP estimation, we then presented six relevant properties (cf. P1–P6 in Section 3) for categorizing related techniques in this research area. (2) Considering these six properties, we proposed our framework MaDL. A modular, probabilistic design and a weighted loss function modeling annotator correlations characterize its novelties. (3) We experimentally investigated the six properties via three RQs. The results confirmed MaDL's robust and often superior performance to related multi-annotator supervised learning techniques. The findings of this article, with a focus on AP estimation, provide a starting point for several aspects of future research, some examples of which are given below.

Although the annotator embeddings already contain information about the annotation patterns concerning instances and classes, MaDL is currently limited to computing annotator correlations on a global level, i.e., annotator weights are not an explicit function of instance-annotator pairs. For example, an extension in this direction may be valuable to quantify correlations in certain regions of the feature space. Leveraging AP estimates for additional applications, e.g., selecting the best crowdworkers to obtain high-quality annotations during a crowdsourcing campaign (Herde et al., 2023), is also of great value. Another neglected aspect is the study of epistemic uncertainty (Huseljic et al., 2021). For example, the visualizations for the two-dimensional dataset in Fig. 6 show high certainty of the GT and AP models in feature space regions with no observed instances. However, meaningful epistemic uncertainty estimates are essential in many (safety-critical) applications (Hüllermeier & Waegeman, 2021) and would improve the characterization of annotators' knowledge. During our experiments, we showed the potential benefit of annotator features. We had no access to a dataset with prior information from real-world annotators, so we needed a suitable simulation for these features. Therefore, and also noted by Zhang et al. (2023), future research may acquire such prior information via crowdsourcing to verify their benefit. As the concentration of annotators may fluctuate or annotators may learn during the annotation process, taking time-varying APs into account is another potential avenue

Table 8: Results regarding RQ3 for datasets with simulated annotators: Best and second best performances are highlighted per dataset and evaluation score while excluding the performances of the UB. The AP models' results refer to the 25 test annotators providing no class labels for training. An entry – marks a technique whose AP model cannot make predictions for such test annotators.

| Technique | P5 | P6 | Ground Truth Model | | | Annotator Performance Model | | | |
|---|---|---|---|---|---|---|---|---|---|
| | | | ACC ↑ | NLL ↓ | BS ↓ | ACC ↑ | NLL ↓ | BS ↓ | BAL-ACC ↑ |
| LETTER (INDUCTIVE) | | | | | | | | | |
| UB | ✓ | ✓ | 0.962±0.002 | 0.129±0.003 | 0.058±0.002 | 0.672±0.005 | 0.745±0.047 | 0.457±0.011 | 0.612±0.005 |
| LB | ✓ | ✓ | 0.861±0.005 | 1.090±0.017 | 0.429±0.008 | 0.569±0.008 | 0.730±0.011 | 0.522±0.007 | 0.537±0.006 |
| LIA(Ā) | ✗ | ✗ | 0.875±0.006 | 0.901±0.060 | 0.350±0.024 | – | – | – | – |
| LIA(A) | ✓ | ✓ | 0.876±0.006 | 1.006±0.177 | 0.397±0.074 | 0.609±0.017 | 1.447±0.845 | 0.597±0.105 | 0.545±0.033 |
| CoNAL(Ā) | ✗ | ✗ | 0.875±0.009 | 0.804±0.119 | 0.186±0.010 | – | – | – | – |
| CoNAL(A) | ✓ | ✗ | 0.874±0.007 | 0.808±0.116 | 0.186±0.011 | – | – | – | – |
| MaDL(Ā) | ✗ | ✗ | 0.911±0.006 | 0.334±0.026 | 0.129±0.008 | – | – | – | – |
| MaDL(A) | ✓ | ✓ | 0.914±0.004 | 0.303±0.009 | 0.124±0.005 | 0.668±0.007 | 0.813±0.115 | 0.471±0.015 | 0.600±0.010 |
| FMNIST (INDUCTIVE) | | | | | | | | | |
| UB | ✓ | ✓ | 0.909±0.002 | 0.246±0.005 | 0.131±0.003 | 0.730±0.008 | 0.536±0.019 | 0.357±0.010 | 0.656±0.009 |
| LB | ✓ | ✓ | 0.881±0.002 | 0.876±0.005 | 0.370±0.002 | 0.590±0.023 | 0.681±0.005 | 0.487±0.006 | 0.537±0.010 |
| LIA(Ā) | ✗ | ✗ | 0.852±0.003 | 1.011±0.020 | 0.436±0.010 | – | – | – | – |
| LIA(A) | ✓ | ✓ | 0.855±0.002 | 0.972±0.012 | 0.417±0.006 | 0.674±0.036 | 0.626±0.026 | 0.436±0.024 | 0.601±0.027 |
| CoNAL(Ā) | ✗ | ✗ | 0.889±0.002 | 0.322±0.005 | 0.163±0.003 | – | – | – | – |
| CoNAL(A) | ✓ | ✗ | 0.890±0.002 | 0.323±0.011 | 0.163±0.005 | – | – | – | – |
| MaDL(Ā) | ✗ | ✗ | 0.895±0.002 | 0.297±0.004 | 0.152±0.002 | – | – | – | – |
| MaDL(A) | ✓ | ✓ | 0.893±0.004 | 0.297±0.008 | 0.153±0.004 | 0.723±0.004 | 0.538±0.003 | 0.362±0.003 | 0.649±0.005 |
| CIFAR10 (INDUCTIVE) | | | | | | | | | |
| UB | ✓ | ✓ | 0.931±0.002 | 0.527±0.022 | 0.122±0.003 | 0.686±0.006 | 0.646±0.101 | 0.409±0.016 | 0.613±0.006 |
| LB | ✓ | ✓ | 0.781±0.003 | 1.054±0.035 | 0.447±0.016 | 0.583±0.009 | 0.684±0.004 | 0.490±0.004 | 0.521±0.003 |
| LIA(Ā) | ✗ | ✗ | 0.798±0.004 | 1.072±0.014 | 0.455±0.006 | – | – | – | – |
| LIA(A) | ✓ | ✓ | 0.804±0.004 | 1.056±0.022 | 0.447±0.011 | 0.607±0.020 | 0.670±0.017 | 0.477±0.016 | 0.544±0.010 |
| CoNAL(Ā) | ✗ | ✗ | 0.835±0.002 | 0.576±0.016 | 0.245±0.005 | – | – | – | – |
| CoNAL(A) | ✓ | ✗ | 0.834±0.006 | 0.574±0.017 | 0.248±0.007 | – | – | – | – |
| MaDL(Ā) | ✗ | ✗ | 0.811±0.008 | 0.626±0.036 | 0.277±0.014 | – | – | – | – |
| MaDL(A) | ✓ | ✓ | 0.837±0.003 | 0.557±0.028 | 0.242±0.006 | 0.698±0.003 | 0.567±0.015 | 0.383±0.004 | 0.617±0.004 |
| SVHN (INDUCTIVE) | | | | | | | | | |
| UB | ✓ | ✓ | 0.965±0.001 | 0.393±0.015 | 0.063±0.002 | 0.613±0.004 | 0.943±0.113 | 0.511±0.015 | 0.524±0.006 |
| LB | ✓ | ✓ | 0.927±0.002 | 0.805±0.016 | 0.328±0.009 | 0.588±0.010 | 0.704±0.007 | 0.509±0.006 | 0.511±0.007 |
| LIA(Ā) | ✗ | ✗ | 0.929±0.003 | 0.818±0.133 | 0.336±0.068 | – | – | – | – |
| LIA(A) | ✓ | ✓ | 0.932±0.001 | 0.754±0.152 | 0.303±0.079 | 0.603±0.013 | 0.671±0.024 | 0.478±0.022 | 0.513±0.008 |
| CoNAL(Ā) | ✗ | ✗ | 0.941±0.001 | 0.258±0.009 | 0.090±0.003 | – | – | – | – |
| CoNAL(A) | ✓ | ✗ | 0.942±0.001 | 0.260±0.012 | 0.090±0.002 | – | – | – | – |
| MaDL(Ā) | ✗ | ✗ | 0.928±0.002 | 0.299±0.019 | 0.109±0.005 | – | – | – | – |
| MaDL(A) | ✓ | ✓ | 0.935±0.001 | 0.256±0.009 | 0.098±0.002 | 0.624±0.007 | 0.632±0.013 | 0.444±0.008 | 0.521±0.006 |

for future research (Donmez et al., 2010). Furthermore, there are already crowdsourcing approaches (Chang et al., 2017) and concepts (Calma et al., 2016) supporting collaboration between annotators. Thus, developing techniques considering or recommending such collaborations is of practical value (Fang et al., 2012).

Finally, we limited ourselves to empirical performance results and classification tasks with class labels as annotations. Future investigations on theoretical performance guarantees of MaDL and the learning with different annotation types, such as class labels with confidence scores (Berthon et al., 2021) or partial labels (Yu et al., 2022), are apparent. Furthermore, the extension to related supervised learning tasks, such as semantic segmentation, sequence classification, and regression, is of interest. The goal of semantic segmentation is to classify individual pixels (Minaee et al., 2021). A potential approach to extend MaDL would be to implement its GT model through a U-Net (Ronneberger et al., 2015) and feed its latent representations as input to the AP model for estimating pixel-wise confusion matrices per annotator. Likewise, we may adapt MaDL to be applied to sequence classification tasks, such as named entity recognition (Li et al., 2020). Concretely, we could implement the GT model through a BiLSTM-network with softmax outputs (Reimers & Gurevych, 2017) and feed its latent word representations as inputs to the AP model for estimating word-wise confusion matrices per annotator. Since both extensions involve higher computational costs than standard classification tasks, one may alternatively investigate the estimation of a single (pixel- or word-independent) confusion matrix per annotator. Regression tasks expect the prediction of continuous target variables. Therefore, the probabilistic model of MaDL has to be adapted. For example, the GT model could estimate the mean and variance of an instance's target variable, while the AP model learns annotators' biases and variances.

## Broader Impact Statement

Big data is a driving force behind the success of machine learning (Zhou et al., 2017). Reducing the effort and cost required for annotating this data is essential for its ongoing development In this context, MaDL is a possible tool to leverage the workforce of cost-efficient but error-prone annotators. Yet, as a central resource for data annotation, crowdsourcing can negatively impact individuals or even entire communities. Some of these impacts include exploiting vulnerable individuals who participate in low-wage crowdsourcing tasks (Schlagwein et al., 2019), producing low-quality data (Daniel et al., 2018), and outsourcing jobs (Howe, 2008). On the one hand, multi-annotator supervised learning techniques can improve data quality and support awarding well-performing crowdworkers. On the other hand, such a technique may intensify the already existing competition between crowdworkers (Schlagwein et al., 2019). It also requires tight monitoring to ensure fair assessments of crowdworkers. Besides the benefits of annotator features containing prior information about annotators, there are several risks. Collecting and leaking potentially sensitive personal data about the annotators is such a significant risk (Xia & McKernan, 2020). Thus, the annotator features must contain only information relevant to the learning task. Further, a lack of control over this or other processes can lead to discrimination and bias based on gender, origin, and other factors (Goel & Faltings, 2019). For these reasons, it is crucial to consider and address the potential risks via responsible policies and practices when employing multi-annotator supervised learning techniques.

## Acknowledgments

We thank Lukas Rauch for the insightful discussions and comments, which greatly improved this article.

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

# A Proofs

*Proof of Proposition 1.* Minimizing the loss in Eq. 1 results in the following Bayes optimal prediction:

$$y_{\mathrm{GT}}(\mathbf{x}) = \underset{y' \in \Omega_Y}{\arg\min} \left( \mathbb{E}_{y|\mathbf{x}} \left[ \delta(y \neq y') \right] \right) = \underset{y' \in \Omega_Y}{\arg\min} \left( \sum_{y \in \Omega_Y} \Pr(y \mid \mathbf{x}) \delta(y \neq y') \right)$$

$$= \underset{y' \in \Omega_Y}{\arg\min} \left( \sum_{y \in \Omega_Y \setminus \{y'\}} \Pr(y \mid \mathbf{x}) \right) = \underset{y' \in \Omega_Y}{\arg\min} \left( 1 - \Pr(y' \mid \mathbf{x}) \right) = \underset{y' \in \Omega_Y}{\arg\max} \left( \Pr(y' \mid \mathbf{x}) \right). \qquad \square$$

*Proof of Proposition 2.* Minimizing the loss in Eq. 4 results in the following Bayes optimal prediction:

$$
y_{\text{AP}}(\mathbf{x}, \mathbf{a}) = \underset{y' \in \{0,1\}}{\arg\min} \left( \mathbb{E}_{y|\mathbf{x}} \left[ \mathbb{E}_{z|\mathbf{x},\mathbf{a},y} \left[ \delta\left(y' \neq \delta\left(y \neq z\right)\right)\right]\right]\right)
$$

$$
= \underset{y' \in \{0,1\}}{\arg\min} \left( \sum_{y \in \Omega_Y} \Pr(y \mid \mathbf{x}) \left( \sum_{z \in \Omega_Y} \Pr(z \mid \mathbf{x}, \mathbf{a}, y)\delta(y' \neq \delta(y \neq z)) \right) \right)
$$

$$
= \underset{y' \in \{0,1\}}{\arg\min} \left( \sum_{y \in \Omega_Y} \Pr(y \mid \mathbf{x}) \left( \sum_{z \in \Omega_Y \setminus \{y\}} \Pr(z \mid \mathbf{x}, \mathbf{a}, y)\delta(y' \neq 1) + \Pr(y \mid \mathbf{x}, \mathbf{a}, y)\delta(y' \neq 0) \right) \right)
$$

$$
= \underset{y' \in \{0,1\}}{\arg\min} \left( \sum_{y \in \Omega_Y} \Pr(y \mid \mathbf{x}) \Big( (1 - \Pr(y \mid \mathbf{x}, \mathbf{a}, y))\delta(y' \neq 1) + \Pr(y \mid \mathbf{x}, \mathbf{a}, y)\delta(y' \neq 0) \Big) \right)
$$

$$
= \delta \left( \sum_{y \in \Omega_Y} \Pr(y \mid \mathbf{x}) \Pr(y \mid \mathbf{x}, \mathbf{a}, y) < \sum_{y \in \Omega_Y} \Pr(y \mid \mathbf{x}) \left( 1 - \Pr(y \mid \mathbf{x}, \mathbf{a}, y) \right) \right)
$$

$$
= \delta \left( \sum_{y \in \Omega_Y} \Pr(y \mid \mathbf{x}) \Pr(y \mid \mathbf{x}, \mathbf{a}, y) < \sum_{y \in \Omega_Y} \Pr(y \mid \mathbf{x}) - \sum_{y \in \Omega_Y} \Pr(y \mid \mathbf{x}) \Pr(y \mid \mathbf{x}, \mathbf{a}, y) \right)
$$

$$
= \delta \left( \sum_{y \in \Omega_Y} \Pr(y \mid \mathbf{x}) \Pr(y \mid \mathbf{x}, \mathbf{a}, y) < 1 - \sum_{y \in \Omega_Y} \Pr(y \mid \mathbf{x}) \Pr(y \mid \mathbf{x}, \mathbf{a}, y) \right)
$$

$$
= \delta \left( \sum_{y \in \Omega_Y} \Pr(y \mid \mathbf{x}) \Pr(y \mid \mathbf{x}, \mathbf{a}, y) < 0.5 \right). \qquad \square
$$

*Proof of Theorem 1.* Applying assumption $(\star)$ of Theorem 1 to Eq. 21, the weight $w(\mathbf{a}_m)$ for an annotator $\mathbf{a}_m$ is given by:

$$
w(\mathbf{a}_m) \overset{(\star)}{=} \frac{M}{G} \sum_{g=1}^{G} \frac{\delta(m \in \mathcal{A}^{(g)})}{|\mathcal{A}^{(g)}|}.
$$

Accordingly, the sums of the annotator weights are uniformly distributed across the $G$ groups:

$$
\sum_{m \in \mathcal{A}^{(1)}} w(\mathbf{a}_m) = \cdots = \sum_{m \in \mathcal{A}^{(G)}} w(\mathbf{a}_m) = \frac{M}{G}. \tag{$\diamond$}
$$

Inserting these annotator weights into the weighted log-likelihood function and making use of assumption $(\dagger)$ in Theorem 1, we get

$$
\sum_{n=1}^{N} \sum_{m=1}^{M} w(\mathbf{a}_m) \ln\left(\Pr(z_{nm} \mid \mathbf{x}_n, \mathbf{a}_m)\right) = \sum_{n=1}^{N} \sum_{g=1}^{G} \sum_{m \in \mathcal{A}^{(g)}} w(\mathbf{a}_m) \ln\left(\Pr(z_{nm} \mid \mathbf{x}_n, \mathbf{a}_m)\right)
$$

$$
\overset{(\dagger)}{=} \sum_{n=1}^{N} \sum_{g=1}^{G} \left( \sum_{m \in \mathcal{A}^{(g)}} w(\mathbf{a}_m) \right) \ln\left(\Pr(z_{nm_g} \mid \mathbf{x}_n, \mathbf{a}_{m_g})\right)
$$

$$
\overset{(\diamond)}{=} \sum_{n=1}^{N} \sum_{g=1}^{G} \frac{M}{G} \ln\left(\Pr(z_{nm_g} \mid \mathbf{x}_n, \mathbf{a}_{m_g})\right)
$$

$$
\propto \sum_{n=1}^{N} \sum_{g=1}^{G} \ln\left(\Pr(z_{nm_g} \mid \mathbf{x}_n, \mathbf{a}_{m_g})\right). \qquad \square
$$

# B  End-to-end Training Algorithm Example

This appendix illustrates the steps of Algorithm 1 for a sampled mini-batch. Let us assume a classification problem with $C = 2$ classes, $M = 2$ annotators, and a mini-batch size of $B = 1$. We further suppose there is no prior information about annotators such that we represent the two annotators via the one-hot encoded vectors

$$\mathbf{a}_1 = (1,0)^{\mathrm{T}} \text{ and } \mathbf{a}_2 = (0,1)^{\mathrm{T}}. \tag{33}$$

In the first step, we compute the instances' class-membership probabilities. For a mini-batch size of $B = 1$, we exemplarily assume we have a single arbitrary instance $\mathbf{x}_n$, for which we obtain

$$\hat{\mathbf{p}}_{\boldsymbol{\theta}}(\mathbf{x}_n) = (0.8, 0.2)^{\mathrm{T}} \tag{34}$$

as class-membership probabilities outputted by the GT model with parameters $\boldsymbol{\theta}$. In the second step, we compute the confusion matrix for each instance-annotator pair. For a mini-batch size of $B = 1$ and $M = 2$ annotators, we obtain

$$\hat{\mathbf{P}}_{\boldsymbol{\omega}}(\mathbf{x}_n, \mathbf{a}_1) = \begin{pmatrix} 1 & 0 \\ 0 & 1 \end{pmatrix} \text{ and } \hat{\mathbf{P}}_{\boldsymbol{\omega}}(\mathbf{x}_n, \mathbf{a}_2) = \begin{pmatrix} 0.5 & 0.5 \\ 0.5 & 0.5 \end{pmatrix} \tag{35}$$

as two exemplary confusion matrices outputted by the AP model with parameters $\boldsymbol{\omega}$. Thus, we currently expect annotator $\mathbf{a}_1$ to be error-free and $\mathbf{a}_2$ to randomly guess. In the third step, we determine the similarities between all pairs of annotator embeddings to compute their weights in the fourth step. Since the Gaussian kernel $k_\gamma$ is symmetric and there are only two annotators, we obtain

$$\hat{w}_\gamma(\mathbf{a}_1) = \hat{w}_\gamma(\mathbf{a}_2) = 1 \tag{36}$$

as annotator weights in our example. We refer to Fig. 7 for a more complex example of computing annotator weights. Before evaluating the loss function, we assume $\mathbf{z}_n = (2,2)^{\mathrm{T}}$ as the vector of class labels assigned by the annotators $\mathbf{a}_1$ and $\mathbf{a}_2$ to instance $\mathbf{x}_n$. Moreover, we take $\gamma = 1$ as an example bandwidth and $\alpha = 2, \beta = 1$ as parameters of the gamma distribution. Now, we have all the ingredients to evaluate the loss function in Eq. 25:

$$L_{\mathbf{x}_n, \mathbf{a}_1, \mathbf{a}_2, \mathbf{z}_n, \alpha, \beta}(\boldsymbol{\theta}, \boldsymbol{\omega}, \gamma) = \underbrace{-\frac{1}{2} \cdot 1 \cdot \ln\left( (0,1) \begin{pmatrix} 1 & 0 \\ 0 & 1 \end{pmatrix}^{\mathrm{T}} \begin{pmatrix} 0.8 \\ 0.2 \end{pmatrix} \right)}_{\text{prediction loss for annotator } \mathbf{a}_1} \tag{37}$$

$$\underbrace{-\frac{1}{2} \cdot 1 \cdot \ln\left( (0,1) \begin{pmatrix} 0.5 & 0.5 \\ 0.5 & 0.5 \end{pmatrix}^{\mathrm{T}} \begin{pmatrix} 0.8 \\ 0.2 \end{pmatrix} \right)}_{\text{prediction loss for annotator } \mathbf{a}_2} \tag{38}$$

$$\underbrace{-\ln\left(\mathrm{Gam}\left(1 \mid 2, 1\right)\right)}_{\text{regularization term for the bandwidth } \gamma} \tag{39}$$

$$= -\frac{1}{2}\ln(0.2) - \frac{1}{2}\ln(0.5) + 1. \tag{40}$$

Eq. 37 and Eq. 38 compute the cross-entropy loss between the estimated annotation probabilities

$$\hat{\mathbf{p}}_{\boldsymbol{\theta}, \boldsymbol{\omega}}(\mathbf{x}_n, \mathbf{a}_1) = \begin{pmatrix} 1 & 0 \\ 0 & 1 \end{pmatrix}^{\mathrm{T}} \begin{pmatrix} 0.8 \\ 0.2 \end{pmatrix} = \begin{pmatrix} 0.8 \\ 0.2 \end{pmatrix}, \qquad \hat{\mathbf{p}}_{\boldsymbol{\theta}, \boldsymbol{\omega}}(\mathbf{x}_n, \mathbf{a}_1) = \begin{pmatrix} 0.5 & 0.5 \\ 0.5 & 0.5 \end{pmatrix}^{\mathrm{T}} \begin{pmatrix} 0.8 \\ 0.2 \end{pmatrix} = \begin{pmatrix} 0.5 \\ 0.5 \end{pmatrix} \tag{41}$$

and the two provided annotations as one-hot encoded targets to learn annotation patterns. In contrast, Eq. 39 computes the logarithmic probability density of the current bandwidth value in relation to the gamma distribution to regularize the possible bandwidth values. Finally, we can use a common optimizer for DNNs to update the parameters $\boldsymbol{\theta}, \boldsymbol{\omega}, \gamma$.

## C  Ablation Study

This appendix presents the results of an ablation study regarding MaDL's hyperparameters. Table 9 provides the results regarding the two datasets MUSIC and LABELME with real-world annotators and Table 10 presents the results for the dataset LETTER with the four annotator sets simulated according to Table 3. We design the ablation study following a one-factor-at-a-time approach in favor of reducing the computational cost. This means we define a default MaDL variant for each experiment and change the value of only one hyperparameter at a time. For example, we study the effect of the AP prior $\eta \in (0,1)$ by taking the default MaDL variant and changing only the value of this hyperparameter. Our default MaDL variant corresponds to the hyperparameter values described in Section 4. For the combination of the dataset LETTER with the annotator set INDUCTIVE, the default MaDL variant gets annotator features containing prior information instead of one-hot encoded annotator features as input. The general setup of an experiment, e.g., the number of repeated runs, the splits into the training, test, and validation sets, etc., is identical to the one described in Section 5. In the following, we analyze the effects of the individual hyperparameters, whose default values are given in brackets:

*Embedding size* ($Q = R = 16$)*:* The embedding size controls the dimensionality of the instance and annotator embeddings learned by the AP model. For the two datasets with real-world annotators in Table 9, an embedding size of $Q = R = 16$ clearly works best. In contrast, the results of Table 10 indicate that an embedding size of $Q = R = 8$ is superior for the dataset LETTER with simulated annotator sets. Although $Q = R = 16$ is a robust default embedding size for the tested learning tasks, it is crucial to consider the characteristics of each learning task individually. Thereby, the number of annotators and instance features are of particular importance.

*AP prior* ($\eta = 0.8$)*:* The AP prior controls the initialization of the AP output layer's biases (cf. Eq. 19) and thus the parametrization of initial annotator confusion matrices. This hyperparameter is relevant for the identifiability of the class-membership probabilities and annotator confusion matrices. The results of Table 9 and 10 confirm this importance. A low value of $\eta = 0.1$ leads to poor performance across all tested datasets since it cannot identify the annotation noise. Selecting high values, e.g., $\eta \in \{0.7, 0.8, 0.9\}$, resolves such an issue and thus leads to much better performances.

*Outer product* (True)*:* The outer product layer is one option, adopted from literature (Qu et al., 2016), to model the interactions between instance and annotator embeddings. Training MaDL with such a product layer leads to performance gains for MUSIC and LABELME as datasets with real-world annotators (cf. Table 9). In contrast, there are no clear performance differences for the dataset LETTER with the four simulated annotator sets (cf. Table 10). Since this modeling of interactions resembles recommender systems, testing alternatives, e.g., computing the outer product of instance and annotator embeddings as input to a convolutional layer (He et al., 2018), may be worthwhile.

*Residual block* (True)*:* This hyperparameter determines whether we implement the residual connection (True) or not (False) into the block shown in Fig. 4. The idea of this connection is to prioritize the annotator embeddings when computing APs. Inspecting the results, we see clear performance gains for MUSIC and LABELME as datasets with real-world annotators (cf. Table 9). In contrast, the residual connection leads for two of the simulated annotator sets to better and for the other two to worse results for the dataset LETTER. Similar to investigating alternatives to the outer product layer, optimizing the entire architecture in Fig. 4 may be valuable in future work.

*Gamma prior* ($\alpha = 1.25, \beta = 0.25$)*:* This pair of hyperparameters specifies the prior gamma distribution of the bandwidth $\gamma$ used for computing kernel values between annotator embeddings. Table 9 and 10 list the evaluated pairs of $\alpha$ and $\beta$, including the option of no annotator weights. The values for $\alpha$ and $\beta$ are selected to test different combinations of modes, i.e., $(\alpha-1)/\beta \in \{0.5, 1, 2\}$, and variances, i.e., $\alpha/\beta^2 \in \{6, 20\}$, for the gamma distribution. Table 10 indicates no large performance difference of varying value pairs of $\alpha$ and $\beta$. As previously shown for RQ2 in Section 5, the weak results of the option "No Weights" for the dataset LETTER with RANDOM-CORRELATED as annotator set demonstrate the importance of modeling annotator correlations. Table 9 indicates larger performance differences

for the varying parametrizations. Here, a mode of $(\alpha-1)/\beta = 1$, corresponding to $(\alpha = 1.25, \beta = 0.25)$ and $(\alpha = 1.5, \beta = 0.5)$, leads to competitive performances across both datasets. In general, one could also replace the Gaussian kernel with other common kernels or similarity functions to estimate annotators' densities as a basis for quantifying their correlations. A popular alternative would be the cosine similarity function. For example, Wojke & Bewley (2018) demonstrate how to learn an embedding space where this similarity function is optimized by re-parametrizing the conventional softmax classification output.

Table 9: Ablation study on MaDL's hyperparameters for the datasets MUSIC and LABELME with real-world annotators: Best and second best performances are highlighted per annotator set and evaluation score.

| Parameter | Value | Ground Truth Model | | | Ground Truth Model | | |
|---|---|---|---|---|---|---|---|
| | | ACC ↑ | NLL ↓ | BS ↓ | ACC ↑ | NLL ↓ | BS ↓ |
| | | MUSIC | | | LABELME | | |
| Default: MaDL | | 0.743±0.020 | 0.877±0.034 | 0.381±0.013 | 0.867±0.004 | 0.623±0.138 | 0.214±0.009 |
| Embedding Size | $Q = R = 8$ | 0.742±0.011 | 0.907±0.048 | 0.378±0.010 | 0.852±0.010 | 0.728±0.146 | 0.235±0.018 |
| | $Q = R = 32$ | 0.724±0.020 | 0.889±0.067 | 0.388±0.029 | 0.856±0.012 | 0.940±0.280 | 0.242±0.026 |
| AP Prior | $\eta = 0.1$ | 0.200±0.101 | 5.652±1.197 | 1.305±0.213 | 0.139±0.055 | 10.514±5.138 | 1.483±0.351 |
| | $\eta = 0.7$ | 0.734±0.027 | 0.935±0.071 | 0.377±0.020 | 0.851±0.012 | 0.882±0.309 | 0.248±0.026 |
| | $\eta = 0.9$ | 0.735±0.005 | 0.907±0.047 | 0.395±0.008 | 0.854±0.004 | 1.000±0.277 | 0.246±0.012 |
| Outer Product | False | 0.725±0.030 | 0.896±0.076 | 0.394±0.034 | 0.860±0.003 | 1.014±0.221 | 0.238±0.006 |
| Residual Block | False | 0.734±0.015 | 0.911±0.065 | 0.387±0.032 | 0.852±0.006 | 1.121±0.461 | 0.249±0.022 |
| Gamma Prior | No Weights | 0.736±0.014 | 0.857±0.033 | 0.377±0.011 | 0.870±0.008 | 0.634±0.177 | 0.211±0.014 |
| | $\alpha = 1.118, \beta = 0.236$ | 0.731±0.026 | 0.922±0.071 | 0.387±0.021 | 0.853±0.011 | 1.077±0.218 | 0.253±0.018 |
| | $\alpha = 1.226, \beta = 0.452$ | 0.722±0.015 | 0.948±0.086 | 0.399±0.023 | 0.852±0.011 | 0.902±0.316 | 0.245±0.027 |
| | $\alpha = 1.5, \beta = 0.5$ | 0.748±0.009 | 0.869±0.078 | 0.372±0.025 | 0.864±0.001 | 0.685±0.153 | 0.222±0.010 |
| | $\alpha = 1.56, \beta = 0.28$ | 0.722±0.018 | 0.847±0.026 | 0.381±0.012 | 0.860±0.012 | 0.678±0.211 | 0.225±0.019 |
| | $\alpha = 2.22, \beta = 0.61$ | 0.716±0.006 | 0.947±0.068 | 0.405±0.023 | 0.857±0.008 | 0.785±0.244 | 0.236±0.013 |

The above analyses show that the chosen default hyperparameters do not always give the best results but are competitive for most datasets studied. They also indicate that having a (small) validation set with GT labels is required if maximum performance is crucial. Obtaining such a validation set in a setting with error-prone annotators can be expensive. Therefore, future research may examine methods to design such validation sets cost-efficiently. In this context, it is also essential to elaborate further on the theoretical foundations of multi-annotator supervised learning techniques, e.g., by deriving theoretical guarantees for specific GT and AP distribution types.

## D   Extended Results regarding Research Question 3

Table 11 extends Table 8 by the AP models' results regarding the 75 annotators providing class labels for training. The GT models' results of both tables are identical and are given to avoid switching between the two tables. The additional AP models' results confirm the main takeaway for property P5. Excluding the UB, MaDL(A), which uses annotator features including prior information, makes the most accurate AP predictions. This means it also outperforms its counterpart MaDL($\overline{A}$), which has no prior information about the annotators. LIA(A) estimates the APs more accurately than LIA($\overline{A}$) for three of the four datasets. Comparing the AP models of CoNAL(A) and CoNAL($\overline{A}$), performance gains of using annotator features are observable for two of the four datasets, while there are no notable differences for the other two datasets.

## E   A Case Study on Varying Annotation Ratios

This appendix presents a case study on the impact of varying annotation ratios on multi-annotator supervised learning techniques' performances. Fig. 9 displays evaluation results for four different annotation ratios, i.e., 0.2, 0.4, 0.6, and 0.8, as curves for the dataset CIFAR10 with the simulated annotator set INDEPENDENT. For a better interpretation of the results, the upper right plot displays the means and standard deviations over five runs for two descriptive statistics of the training data. On the one hand, we calculate the *annotation accuracy* (ANNOT-ACC), and on the other hand, we calculate the accuracy of the *observed annotations*

Table 10: Ablation study on MaDL's hyperparameters for the dataset LETTER and four simulated annotated sets: Best and second best performances are highlighted per annotator set and evaluation score.

| Parameter | Value | Ground Truth Model | | | Annotator Performance Model | | | |
|---|---|---|---|---|---|---|---|---|
| | | ACC ↑ | NLL ↓ | BS ↓ | ACC ↑ | NLL ↓ | BS ↓ | BAL-ACC ↑ |
| LETTER (INDEPENDENT) | | | | | | | | |
| Default: MaDL | | 0.935±0.006 | 0.303±0.102 | 0.098±0.010 | 0.766±0.004 | 0.491±0.007 | 0.317±0.005 | 0.702±0.005 |
| Embedding Size | $Q = R = 8$ | 0.937±0.004 | 0.285±0.116 | 0.097±0.006 | **0.773±0.003** | **0.476±0.006** | **0.309±0.002** | **0.708±0.003** |
| | $Q = R = 32$ | 0.935±0.007 | 0.327±0.125 | 0.099±0.011 | 0.763±0.004 | 0.501±0.005 | 0.322±0.003 | 0.698±0.005 |
| AP Prior | $\eta = 0.1$ | 0.587±0.048 | 7.686±1.229 | 0.766±0.095 | 0.656±0.018 | 1.194±0.139 | 0.554±0.037 | 0.614±0.015 |
| | $\eta = 0.7$ | 0.938±0.005 | 0.273±0.047 | 0.095±0.007 | 0.767±0.002 | 0.487±0.003 | 0.314±0.002 | 0.702±0.004 |
| | $\eta = 0.9$ | 0.935±0.007 | **0.250±0.027** | 0.095±0.009 | **0.768±0.002** | 0.486±0.003 | 0.315±0.002 | **0.703±0.003** |
| Outer Product | False | 0.933±0.010 | 0.365±0.177 | 0.103±0.018 | 0.765±0.003 | 0.499±0.012 | 0.321±0.006 | 0.701±0.004 |
| Residual Block | False | 0.935±0.003 | 0.326±0.131 | 0.099±0.007 | 0.762±0.004 | 0.513±0.014 | 0.326±0.005 | 0.699±0.004 |
| Gamma Prior | No Weights | **0.939±0.007** | 0.302±0.097 | **0.094±0.011** | 0.767±0.002 | 0.491±0.007 | 0.316±0.003 | 0.702±0.002 |
| | $\alpha = 1.118, \beta = 0.236$ | 0.937±0.005 | 0.285±0.053 | 0.096±0.008 | 0.768±0.004 | **0.484±0.003** | **0.314±0.002** | 0.701±0.004 |
| | $\alpha = 1.226, \beta = 0.452$ | 0.936±0.003 | 0.307±0.081 | 0.097±0.008 | 0.767±0.004 | 0.486±0.006 | 0.315±0.003 | 0.702±0.006 |
| | $\alpha = 1.5, \beta = 0.5$ | **0.940±0.002** | 0.274±0.042 | **0.092±0.004** | 0.767±0.003 | 0.489±0.003 | 0.317±0.003 | 0.703±0.004 |
| | $\alpha = 1.56, \beta = 0.28$ | 0.936±0.003 | **0.266±0.046** | 0.095±0.006 | 0.767±0.001 | 0.491±0.006 | 0.316±0.002 | 0.702±0.002 |
| | $\alpha = 2.22, \beta = 0.61$ | 0.936±0.002 | 0.283±0.048 | 0.097±0.005 | 0.766±0.003 | 0.493±0.005 | 0.318±0.002 | 0.703±0.002 |
| LETTER (CORRELATED) | | | | | | | | |
| Default: MaDL | | 0.947±0.003 | 0.282±0.077 | 0.080±0.004 | 0.887±0.001 | **0.308±0.004** | 0.175±0.002 | 0.756±0.001 |
| Embedding Size | $Q = R = 8$ | **0.953±0.003** | **0.219±0.017** | **0.072±0.004** | **0.888±0.002** | **0.303±0.003** | **0.172±0.002** | 0.757±0.002 |
| | $Q = R = 32$ | 0.949±0.005 | 0.303±0.089 | 0.080±0.008 | 0.885±0.002 | 0.324±0.018 | 0.178±0.004 | 0.754±0.002 |
| AP Prior | $\eta = 0.1$ | 0.588±0.089 | 9.442±2.241 | 0.773±0.166 | 0.782±0.031 | 0.894±0.186 | 0.377±0.062 | 0.659±0.030 |
| | $\eta = 0.7$ | 0.947±0.006 | 0.292±0.080 | 0.078±0.008 | 0.887±0.002 | 0.309±0.005 | 0.174±0.003 | 0.756±0.003 |
| | $\eta = 0.9$ | 0.948±0.003 | 0.255±0.079 | 0.079±0.006 | 0.887±0.001 | 0.311±0.001 | 0.175±0.002 | 0.757±0.002 |
| Outer Product | False | 0.948±0.003 | 0.305±0.109 | 0.081±0.005 | 0.887±0.001 | 0.314±0.005 | 0.177±0.002 | 0.757±0.001 |
| Residual Block | False | 0.936±0.016 | 0.541±0.486 | 0.101±0.031 | 0.881±0.005 | 0.339±0.032 | 0.185±0.010 | 0.751±0.005 |
| Gamma Prior | No Weights | 0.946±0.007 | 0.293±0.092 | 0.083±0.010 | 0.883±0.003 | 0.314±0.002 | 0.178±0.002 | 0.751±0.003 |
| | $\alpha = 1.118, \beta = 0.236$ | 0.950±0.004 | 0.260±0.089 | 0.077±0.007 | 0.887±0.001 | 0.308±0.004 | 0.174±0.003 | **0.758±0.001** |
| | $\alpha = 1.226, \beta = 0.452$ | 0.951±0.004 | **0.250±0.070** | 0.075±0.006 | **0.888±0.002** | 0.309±0.005 | **0.174±0.003** | **0.758±0.002** |
| | $\alpha = 1.5, \beta = 0.5$ | **0.951±0.005** | 0.271±0.093 | **0.075±0.007** | 0.887±0.001 | 0.311±0.003 | 0.174±0.002 | 0.757±0.002 |
| | $\alpha = 1.56, \beta = 0.28$ | 0.947±0.004 | 0.286±0.072 | 0.082±0.008 | 0.885±0.002 | 0.318±0.013 | 0.177±0.004 | 0.754±0.002 |
| | $\alpha = 2.22, \beta = 0.61$ | 0.947±0.002 | 0.289±0.052 | 0.082±0.003 | 0.885±0.002 | 0.313±0.001 | 0.177±0.001 | 0.754±0.003 |
| LETTER (RANDOM-CORRELATED) | | | | | | | | |
| Default: MaDL | | 0.932±0.004 | 0.277±0.043 | 0.101±0.006 | 0.940±0.000 | 0.204±0.004 | 0.101±0.001 | **0.519±0.001** |
| Embedding Size | $Q = R = 8$ | 0.935±0.003 | **0.232±0.012** | 0.097±0.004 | **0.940±0.000** | **0.202±0.001** | **0.101±0.000** | 0.519±0.000 |
| | $Q = R = 32$ | 0.935±0.003 | 0.266±0.057 | **0.096±0.007** | 0.939±0.000 | 0.214±0.003 | 0.104±0.000 | 0.519±0.000 |
| AP Prior | $\eta = 0.1$ | 0.598±0.069 | 7.537±1.922 | 0.744±0.136 | 0.931±0.002 | 0.251±0.012 | 0.120±0.004 | 0.512±0.001 |
| | $\eta = 0.7$ | 0.934±0.009 | 0.263±0.035 | 0.098±0.012 | 0.940±0.000 | 0.204±0.003 | 0.101±0.000 | 0.519±0.000 |
| | $\eta = 0.9$ | 0.932±0.003 | **0.247±0.015** | 0.101±0.004 | 0.939±0.000 | 0.205±0.002 | 0.102±0.001 | 0.519±0.000 |
| Outer Product | False | 0.932±0.003 | 0.292±0.038 | 0.102±0.006 | 0.939±0.000 | 0.207±0.004 | 0.102±0.001 | 0.519±0.000 |
| Residual Block | False | **0.936±0.005** | 0.269±0.034 | 0.098±0.006 | 0.939±0.000 | 0.207±0.001 | 0.103±0.000 | 0.518±0.000 |
| Gamma Prior | No Weights | 0.548±0.037 | 1.902±0.241 | 0.673±0.071 | 0.801±0.049 | 0.423±0.037 | 0.265±0.031 | 0.506±0.007 |
| | $\alpha = 1.118, \beta = 0.236$ | 0.935±0.001 | 0.270±0.034 | **0.097±0.004** | 0.940±0.000 | **0.204±0.001** | 0.101±0.000 | 0.519±0.000 |
| | $\alpha = 1.226, \beta = 0.452$ | 0.934±0.002 | 0.255±0.039 | 0.097±0.005 | **0.940±0.000** | 0.204±0.002 | **0.101±0.000** | **0.519±0.000** |
| | $\alpha = 1.5, \beta = 0.5$ | **0.935±0.006** | 0.252±0.047 | 0.098±0.009 | 0.940±0.000 | 0.204±0.003 | 0.101±0.001 | 0.519±0.000 |
| | $\alpha = 1.56, \beta = 0.28$ | 0.932±0.004 | 0.306±0.087 | 0.103±0.010 | 0.939±0.000 | 0.210±0.003 | 0.103±0.000 | 0.519±0.000 |
| | $\alpha = 2.22, \beta = 0.61$ | 0.932±0.005 | 0.279±0.048 | 0.101±0.008 | 0.939±0.000 | 0.209±0.003 | 0.103±0.001 | 0.519±0.000 |
| LETTER (INDUCTIVE) | | | | | | | | |
| Default: MaDL(A) | | 0.914±0.004 | 0.303±0.010 | 0.124±0.006 | 0.718±0.004 | 0.583±0.007 | 0.383±0.004 | 0.665±0.005 |
| Embedding Size | $Q = R = 8$ | **0.919±0.004** | **0.287±0.018** | **0.120±0.005** | 0.704±0.012 | **0.569±0.015** | 0.384±0.012 | 0.645±0.012 |
| | $Q = R = 32$ | 0.916±0.005 | 0.330±0.041 | 0.122±0.006 | 0.712±0.007 | 0.632±0.028 | 0.398±0.012 | 0.657±0.008 |
| AP Prior | $\eta = 0.1$ | 0.642±0.065 | 4.982±1.017 | 0.630±0.127 | 0.652±0.017 | 0.999±0.148 | 0.525±0.042 | 0.605±0.015 |
| | $\eta = 0.7$ | 0.912±0.006 | 0.332±0.030 | 0.128±0.009 | 0.704±0.004 | 0.588±0.005 | 0.390±0.003 | 0.650±0.005 |
| | $\eta = 0.9$ | 0.913±0.003 | 0.306±0.021 | 0.127±0.003 | 0.717±0.005 | 0.582±0.005 | 0.383±0.003 | 0.665±0.005 |
| Outer Product | False | 0.914±0.008 | 0.403±0.151 | 0.127±0.011 | **0.725±0.003** | 0.596±0.013 | 0.381±0.004 | **0.676±0.003** |
| Residual Block | False | **0.921±0.003** | **0.286±0.032** | **0.116±0.006** | **0.721±0.004** | 0.593±0.032 | 0.378±0.011 | **0.668±0.004** |
| Gamma Prior | No Weights | 0.915±0.004 | 0.301±0.016 | 0.124±0.003 | 0.719±0.007 | **0.577±0.010** | **0.377±0.006** | 0.663±0.007 |
| | $\alpha = 1.118, \beta = 0.236$ | 0.917±0.006 | 0.308±0.033 | 0.123±0.008 | 0.718±0.004 | 0.593±0.015 | 0.384±0.006 | 0.667±0.006 |
| | $\alpha = 1.226, \beta = 0.452$ | 0.918±0.002 | 0.298±0.024 | 0.121±0.002 | 0.719±0.006 | 0.583±0.016 | 0.381±0.007 | 0.666±0.006 |
| | $\alpha = 1.5, \beta = 0.5$ | 0.914±0.000 | 0.304±0.006 | 0.126±0.003 | 0.716±0.007 | 0.581±0.018 | 0.383±0.010 | 0.663±0.005 |
| | $\alpha = 1.56, \beta = 0.28$ | 0.910±0.006 | 0.341±0.059 | 0.132±0.007 | 0.711±0.008 | 0.588±0.009 | 0.389±0.007 | 0.659±0.006 |
| | $\alpha = 2.22, \beta = 0.61$ | 0.917±0.007 | 0.334±0.072 | 0.124±0.009 | 0.714±0.007 | 0.586±0.015 | 0.385±0.008 | 0.660±0.005 |

*aggregated via the majority rule* (MR-ACC). Mathematically, both statistics can be expressed as follows:

$$\text{ANNOT-ACC}(\mathbf{y}, \mathbf{Z}) \coloneqq \frac{1}{|\mathbf{Z}|} \sum_{n=1}^{N} \sum_{m \in \mathcal{A}_n} \delta\left(y_n = z_{nm}\right), \tag{42}$$

$$\text{MR-ACC}(\mathbf{y}, \bar{\mathbf{z}}) \coloneqq \frac{1}{N} \sum_{n=1}^{N} \delta\left(y_n = \bar{z}_n\right), \tag{43}$$

Table 11: Additional results regarding RQ3 for datasets with simulated annotators: Best and second best performances are highlighted per dataset and evaluation score while excluding the performances of the UB. In contrast to Table 8, the AP models' results refer to the 75 annotators providing class labels for training.

| Technique | P5 | P6 | Ground Truth Model | | | Annotator Performance Model | | | |
|---|---|---|---|---|---|---|---|---|---|
| | | | ACC ↑ | NLL ↓ | BS ↓ | ACC ↑ | NLL ↓ | BS ↓ | BAL-ACC ↑ |
| LETTER (INDUCTIVE) | | | | | | | | | |
| UB | ✓ | ✓ | 0.962±0.002 | 0.129±0.003 | 0.058±0.002 | 0.729±0.004 | 0.565±0.010 | 0.369±0.005 | 0.677±0.004 |
| LB | ✓ | ✓ | 0.861±0.005 | 1.090±0.017 | 0.429±0.008 | 0.572±0.008 | 0.719±0.008 | 0.515±0.006 | 0.552±0.004 |
| LIA($\overline{\text{A}}$) | ✗ | ✗ | 0.875±0.006 | 0.901±0.060 | 0.350±0.024 | 0.622±0.002 | 0.712±0.015 | 0.495±0.007 | 0.593±0.003 |
| LIA(A) | ✓ | ✓ | 0.876±0.006 | 1.006±0.177 | 0.397±0.074 | 0.618±0.028 | 1.441±0.925 | 0.581±0.129 | 0.572±0.038 |
| CoNAL($\overline{\text{A}}$) | ✗ | ✗ | 0.875±0.009 | 0.804±0.119 | 0.186±0.010 | 0.646±0.001 | 0.666±0.019 | 0.455±0.007 | 0.595±0.001 |
| CoNAL(A) | ✓ | ✗ | 0.874±0.007 | 0.808±0.116 | 0.186±0.011 | 0.646±0.001 | 0.662±0.015 | 0.453±0.006 | 0.595±0.001 |
| MaDL($\overline{\text{A}}$) | ✗ | ✗ | 0.911±0.006 | 0.334±0.026 | 0.129±0.008 | 0.685±0.011 | 0.601±0.008 | 0.403±0.009 | 0.626±0.012 |
| MaDL(A) | ✓ | ✓ | 0.914±0.004 | 0.303±0.009 | 0.124±0.005 | 0.718±0.004 | 0.583±0.006 | 0.383±0.004 | 0.665±0.004 |
| FMNIST (INDUCTIVE) | | | | | | | | | |
| UB | ✓ | ✓ | 0.909±0.002 | 0.246±0.005 | 0.131±0.003 | 0.759±0.001 | 0.486±0.002 | 0.320±0.001 | 0.693±0.002 |
| LB | ✓ | ✓ | 0.881±0.002 | 0.876±0.005 | 0.370±0.002 | 0.621±0.020 | 0.662±0.006 | 0.469±0.006 | 0.562±0.009 |
| LIA($\overline{\text{A}}$) | ✗ | ✗ | 0.852±0.003 | 1.011±0.020 | 0.436±0.010 | 0.657±0.022 | 0.641±0.013 | 0.449±0.013 | 0.589±0.015 |
| LIA(A) | ✓ | ✓ | 0.855±0.002 | 0.972±0.012 | 0.417±0.006 | 0.685±0.031 | 0.623±0.023 | 0.432±0.021 | 0.610±0.025 |
| CoNAL($\overline{\text{A}}$) | ✗ | ✗ | 0.889±0.002 | 0.322±0.005 | 0.163±0.003 | 0.705±0.009 | 0.547±0.010 | 0.372±0.009 | 0.653±0.010 |
| CoNAL(A) | ✓ | ✗ | 0.890±0.002 | 0.323±0.011 | 0.163±0.005 | 0.713±0.011 | 0.540±0.011 | 0.366±0.010 | 0.661±0.013 |
| MaDL($\overline{\text{A}}$) | ✗ | ✗ | 0.895±0.002 | 0.297±0.004 | 0.152±0.002 | 0.753±0.003 | 0.496±0.004 | 0.328±0.003 | 0.683±0.005 |
| MaDL(A) | ✓ | ✓ | 0.893±0.004 | 0.297±0.008 | 0.153±0.004 | 0.755±0.003 | 0.492±0.004 | 0.325±0.003 | 0.686±0.005 |
| CIFAR10 (INDUCTIVE) | | | | | | | | | |
| UB | ✓ | ✓ | 0.931±0.002 | 0.527±0.022 | 0.122±0.003 | 0.712±0.001 | 0.555±0.002 | 0.372±0.002 | 0.655±0.001 |
| LB | ✓ | ✓ | 0.781±0.003 | 1.054±0.035 | 0.447±0.016 | 0.584±0.006 | 0.682±0.004 | 0.489±0.003 | 0.534±0.004 |
| LIA($\overline{\text{A}}$) | ✗ | ✗ | 0.798±0.008 | 1.072±0.014 | 0.455±0.006 | 0.598±0.013 | 0.676±0.014 | 0.482±0.013 | 0.547±0.008 |
| LIA(A) | ✓ | ✓ | 0.804±0.004 | 1.056±0.022 | 0.447±0.011 | 0.602±0.022 | 0.673±0.017 | 0.479±0.016 | 0.549±0.014 |
| CoNAL($\overline{\text{A}}$) | ✗ | ✗ | 0.835±0.002 | 0.576±0.016 | 0.245±0.005 | 0.670±0.002 | 0.599±0.002 | 0.415±0.002 | 0.618±0.001 |
| CoNAL(A) | ✓ | ✗ | 0.834±0.006 | 0.574±0.017 | 0.248±0.007 | 0.670±0.001 | 0.599±0.001 | 0.415±0.001 | 0.618±0.002 |
| MaDL($\overline{\text{A}}$) | ✗ | ✗ | 0.811±0.008 | 0.626±0.036 | 0.277±0.014 | 0.690±0.002 | 0.565±0.002 | 0.386±0.001 | 0.624±0.003 |
| MaDL(A) | ✓ | ✓ | 0.837±0.003 | 0.557±0.028 | 0.242±0.006 | 0.703±0.004 | 0.561±0.015 | 0.378±0.005 | 0.640±0.005 |
| SVHN (INDUCTIVE) | | | | | | | | | |
| UB | ✓ | ✓ | 0.965±0.001 | 0.393±0.015 | 0.063±0.002 | 0.650±0.003 | 0.589±0.002 | 0.410±0.002 | 0.578±0.004 |
| LB | ✓ | ✓ | 0.927±0.002 | 0.805±0.016 | 0.328±0.009 | 0.579±0.006 | 0.709±0.005 | 0.513±0.004 | 0.531±0.003 |
| LIA($\overline{\text{A}}$) | ✗ | ✗ | 0.929±0.003 | 0.818±0.133 | 0.336±0.068 | 0.548±0.043 | 0.695±0.029 | 0.502±0.027 | 0.528±0.009 |
| LIA(A) | ✓ | ✓ | 0.932±0.001 | 0.754±0.152 | 0.303±0.079 | 0.600±0.009 | 0.672±0.027 | 0.480±0.024 | 0.539±0.004 |
| CoNAL($\overline{\text{A}}$) | ✗ | ✗ | 0.941±0.001 | 0.258±0.009 | 0.090±0.003 | 0.627±0.005 | 0.608±0.002 | 0.428±0.002 | 0.551±0.006 |
| CoNAL(A) | ✓ | ✗ | 0.942±0.001 | 0.260±0.012 | 0.090±0.002 | 0.632±0.007 | 0.605±0.003 | 0.425±0.003 | 0.556±0.007 |
| MaDL($\overline{\text{A}}$) | ✗ | ✗ | 0.928±0.002 | 0.299±0.019 | 0.109±0.005 | 0.631±0.004 | 0.599±0.006 | 0.418±0.002 | 0.548±0.002 |
| MaDL(A) | ✓ | ✓ | 0.935±0.001 | 0.256±0.009 | 0.098±0.002 | 0.641±0.002 | 0.592±0.001 | 0.413±0.001 | 0.562±0.002 |

where $\overline{\mathbf{z}} = (\overline{z}_1, \ldots, \overline{z}_N)^{\mathrm{T}} \in \Omega_Y^N$ denotes the vector of observed annotations aggregated per instance via the majority rule. The ANNOT-ACC is almost constant across the different annotation ratios, as the selection of which annotations are observed is made randomly. In comparison, the MR-ACC increases because, on average, the annotators are independent and better than random guessing. Therefore, the probability of correctly aggregated annotation increases with more observed annotations per instance.

The curves in the other seven plots report multi-annotator supervised learning techniques' mean evaluation scores and standard deviations over five runs. The evaluated MaDL variant corresponds to its default hyperparameter configuration. The GT models' evaluation curves in the first row of Fig. 9 confirm our intuition that more annotations lead to better results. Further, we observe superior performances of the UB regarding ACC and BS, while the UB repeatedly provides inferior NLL scores. Since the NLL per instance is unbounded, the latter observation is likely caused by highly overconfident predictions. As expected, the GT model of the LB performs worst in terms of the ACC for each tested annotation ratio and thus confirms the general usefulness of multi-annotator supervised learning. However, its results for the NLL and BS are surprisingly better than LIA, possibly due to the difficulties of training via an EM algorithm (cf. Section 3). The GT model of MaDL outperforms other multi-annotator supervised learning techniques regarding the ACC and BS results, while it is approximately on par with CoNAL regarding the NLL results. As the annotation ratio increases, the ACC of MaDL approaches that of the UB. Still, MaDL cannot reach the ACC results of the UB, even for the highest tested annotation ratio of 0.8. This is because the negative impact of annotation noise is problem-dependent (Gu et al., 2022), e.g., the impact may depend on the data distribution. The annotator simulation is an additional impactful aspect in our concrete case. For CIFAR10,

the negative impact of annotation noise is more significant than for the other datasets, particularly evident in the large gap in the ACC between the LB and the UB. The AP models' evaluation curves in the second row of Fig. 9 show similar trends to the ones of the GT models. In particular, there are two noteworthy aspects. First, the AP estimates of MaDL are close to the UB. Second, the AP estimates of the LB are better than the ones of the other multi-annotator supervised learning techniques (excluding MaDL and UB) for annotation ratios of 0.6 and 0.8. The well-performing AP model architecture adopted from MaDL likely leads to these results.

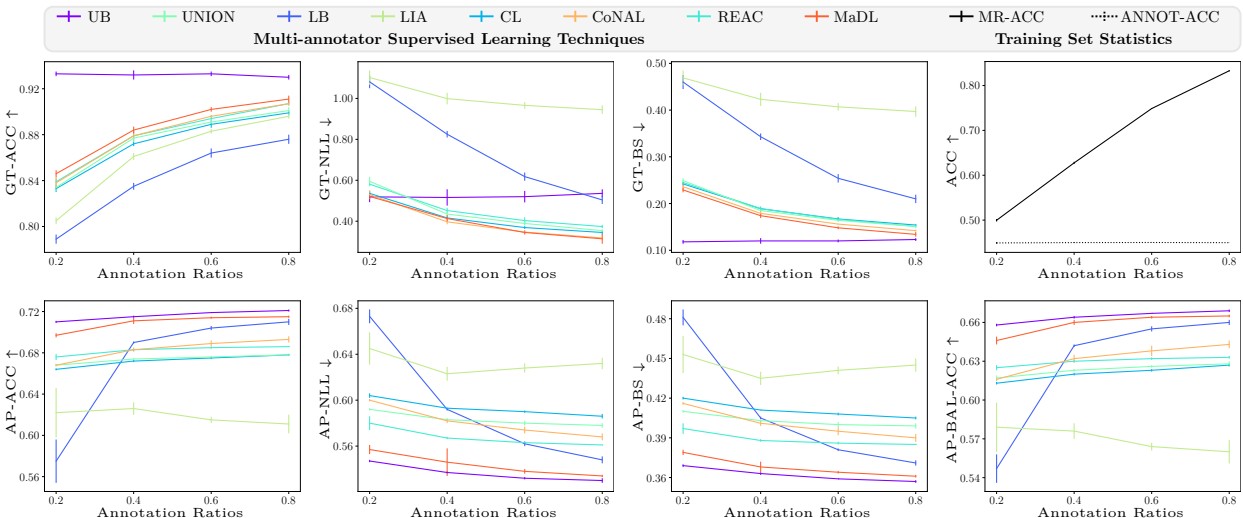

Figure 9: Evaluation results for four different annotation ratios on CIFAR10 (INDEPENDENT).

# F  A Case Study on CIFAR100

This appendix presents a case study analyzing multi-annotator supervised learning techniques' performances on classification tasks with more classes than in the main experiments. For this purpose, we use the dataset CIFAR100, which is similar to the CIFAR10 dataset (cf. Table 2), with the major difference that $C = 100$ classes occur (Krizhevsky, 2009). The general design of the experiments follows that of RQ1 (cf. Section 5.1), except that we adopt the *wide residual network* (WRN-28-10, Zagoruyko & Komodakis, 2016) as the basic network architecture and stochastic GD as the optimizer with a mini-batch size of $B = 128$. For a fairer comparison, we perform a small grid search with the learning rates $\{0.1, 0.01\}$ and weight decays $\{0.0005, 0.0\}$ for each technique. Thereby, we reduce the learning rate with cosine annealing (Loshchilov & Hutter, 2017) over the 100 training epochs.

Table 12 presents the obtained GT and AP models' test performances, where the evaluated MaDL variant corresponds to its default hyperparameter configuration. As expected, training with GT labels as the UB achieves the best performance in terms of GT and AP estimates. The performance gap between the UB and other techniques is substantial, highlighting the challenges of learning the CIFAR100 dataset with partially erroneous annotations. However, the benefits of employing multi-annotator supervised learning techniques are also significant. This advantage becomes particularly apparent when comparing MaDL with the LB because MaDL improves the GT-ACC by approximately 19 %. The comparison with the other multi-annotator supervised learning techniques further confirms the state-of-the-art results of MaDL for CIFAR100. We note that only LIA and REAC appear to roughly learn the classification task among the related techniques. A conceivable explanation might be that the default values of the hyperparameters of CL, UNION, and CoNAL are only well-suited for datasets with fewer classes.

Table 12: Results on the CIFAR100 dataset with simulated annotators: Best and second best performances are highlighted per evaluation score while excluding the performances of the UB.

| Technique | Ground Truth Model | | | Annotator Performance Model | | | |
|---|---|---|---|---|---|---|---|
| | ACC ↑ | NLL ↓ | BS ↓ | ACC ↑ | NLL ↓ | BS ↓ | BAL-ACC ↑ |
| CIFAR100 (INDEPENDENT) | | | | | | | |
| UB | 0.795 ± 0.001 | 0.818 ± 0.008 | 0.297 ± 0.003 | 0.727 ± 0.002 | 0.530 ± 0.001 | 0.355 ± 0.001 | 0.665 ± 0.003 |
| LB | 0.430 ± 0.010 | 2.593 ± 0.038 | 0.762 ± 0.012 | 0.599 ± 0.015 | 0.665 ± 0.006 | 0.472 ± 0.005 | 0.577 ± 0.011 |
| CL | 0.077 ± 0.013 | 10.555 ± 0.802 | 1.616 ± 0.059 | 0.594 ± 0.000 | 1.466 ± 0.013 | 0.764 ± 0.001 | 0.500 ± 0.000 |
| REAC | 0.553 ± 0.006 | 2.037 ± 0.175 | 0.630 ± 0.023 | 0.629 ± 0.029 | 0.657 ± 0.071 | 0.465 ± 0.060 | 0.543 ± 0.035 |
| UNION | 0.033 ± 0.003 | 4.815 ± 0.067 | 1.031 ± 0.014 | 0.594 ± 0.000 | 1.838 ± 0.008 | 0.794 ± 0.001 | 0.500 ± 0.000 |
| LIA | 0.560 ± 0.008 | 1.995 ± 0.051 | 0.619 ± 0.009 | 0.672 ± 0.004 | 0.608 ± 0.003 | 0.417 ± 0.003 | 0.611 ± 0.006 |
| CoNAL | 0.168 ± 0.008 | 7.762 ± 0.344 | 1.283 ± 0.016 | 0.594 ± 0.000 | 1.209 ± 0.021 | 0.700 ± 0.003 | 0.500 ± 0.000 |
| MaDL | 0.621 ± 0.003 | 1.628 ± 0.011 | 0.508 ± 0.005 | 0.665 ± 0.006 | 0.646 ± 0.015 | 0.442 ± 0.011 | 0.628 ± 0.003 |

## G   Practitioner's Guide

This appendix provides an overview of various aspects to consider when employing MaDL in practical classification tasks. Note that these aspects are closely related to each other and that the associated recommendations do not have general validity.

**Data modalities:** We can train MaDL with different modalities of data. Depending on the instances' data modality, we need to modify the GT model's architecture. For example, we may use TabNet (Arik & Pfister, 2021) for tabular data, ResNet (He et al., 2016) for image data, or BERT (Devlin et al., 2018) for text data. These considerations apply analogously to the annotator features and the architecture of the AP model. However, annotator features are commonly tabular since they are collected via surveys, or only anonymized identifiers of the annotators are known if there is no prior annotator information. In the latter case, one-hot encoding converts these identifiers into tabular annotator features. *Recommendation: Use common architectures from the literature to fit the data modality.*

**Number of instances:** The number of annotated instances available for training is critical to DNNs' generalization performances (Hestness et al., 2017) and, consequently, to MaDL's generalization performance. Typically, the more annotated instances we have, the better MaDL can learn the underlying annotation patterns as a basis for inferring the GT class labels and APs. Learning curves allow us to study this behavior (Hoiem et al., 2021). Fig. 10 shows such exemplary learning curves of MaDL's GT-ACC and AP-ACC compared to the LB and UB for the dataset LETTER. Each curve represents the means and standard deviations over five runs. The trend of improving performance with an increasing number of annotated instances is confirmed for both evaluation scores. Further, we observe that MaDL improves upon the LB, even for smaller training sets. Our findings, coupled with the case study on varying annotation ratios in Appendix E, suggest that an increase in both the number of annotated instances and the number of annotations per instance generally contributes to improved results. *Recommendation: If the annotation budget allows, increase the number of annotated instances for better results.*

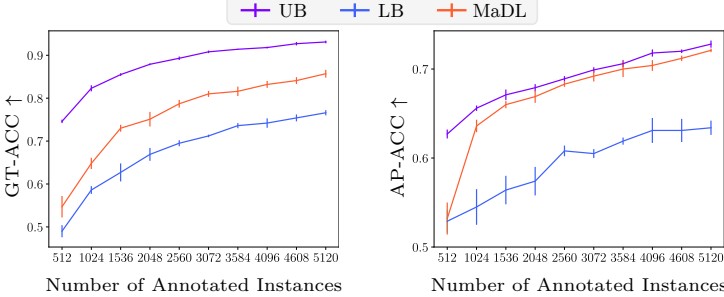

Figure 10: Exemplary learning curves on the LETTER dataset.

**Number of classes:** With an increasing number of classes, there is a growing chance that some classes will be easy to separate and some classes will be highly confusable (Gupta et al., 2014). Detecting such pairs of confusable classes requires the accurate estimation of annotators' confusion matrices, whose concept is illustrated in Fig. 11. The three matrices represent three variants of MaDL with varying assumptions about APs (cf. property P1 in Section 3). The class-independent variant corresponds to estimating a single scalar as the only degree of freedom, i.e., $\nu = 1$, to build the entire confusion matrix. In our example, this scalar corresponds to 0.80 as an annotator's correctness probability. Accordingly, this variant cannot distinguish between easy- and difficult-to-recognize classes. The degree of freedom of the partially class-dependent variant equals the number of classes, i.e., $\nu = C$, because there is an individual correctness probability for each class. These probabilities build the diagonal of the confusion matrix, which consists of the values $0.60, 0.80$, and $1.00$ in our example. Only the fully class-dependent variant can capture pairs of highly confusable classes. As a result, the degree of freedom is considerably higher with $\nu = C \cdot (C-1)$. On the one hand, our experiments show that this variant is robust across different datasets (cf. Section 5), and even works for datasets with $C = 100$ classes (cf. Appendix F). On the other hand, the complexity increases quadratically with the number of classes, so this variant is computationally costly for datasets with hundreds or thousands of classes. Note that MaDL can estimate these confusion matrices not only annotator-dependent but also instance-dependent. *Recommendation: If no assumptions about the annotators are known, and the computational effort is feasible, use the fully class-dependent MaDL variant.*

| **Class-independent (I)** $\nu = 1$ | | | |
|---|---|---|---|
| | Annotated Class | | |
| GT Class | $z = 1$ | $z = 2$ | $z = 3$ |
| $y = 1$ | 0.80 | 0.10 | 0.10 |
| $y = 2$ | 0.10 | 0.80 | 0.10 |
| $y = 3$ | 0.10 | 0.10 | 0.80 |

| **Partially Class-dependent (P)** $\nu = C = 3$ | | | |
|---|---|---|---|
| | Annotated Class | | |
| GT Class | $z = 1$ | $z = 2$ | $z = 3$ |
| $y = 1$ | 0.60 | 0.20 | 0.20 |
| $y = 2$ | 0.10 | 0.80 | 0.10 |
| $y = 3$ | 0.00 | 0.00 | 1.00 |

| **Fully Class-dependent (F)** $\nu = C \cdot (C-1) = 3 \cdot 2 = 6$ | | | |
|---|---|---|---|
| | Annotated Class | | |
| GT Class | $z = 1$ | $z = 2$ | $z = 3$ |
| $y = 1$ | 0.60 | 0.35 | 0.05 |
| $y = 2$ | 0.20 | 0.80 | 0.00 |
| $y = 3$ | 0.00 | 0.00 | 1.00 |

Figure 11: Illustrative confusion matrices for $C = 3$ classes.

**Number of annotators:** As the number of annotators increases, the AP model usually needs to learn more varying annotation patterns. For this, the AP model's size must be sufficiently large, e.g., by increasing the annotator embeddings' dimensionality. Moreover, in settings with many annotators, prior information about the annotators can be particularly helpful for learning correlations between them. *Recommendation: Ensure that the AP model has a sufficient size to learn annotation patterns.*

**Training, validation, and testing:** In principle, MaDL can be trained without a validation or test set, demonstrating robust results with default hyperparameters across diverse datasets. Nevertheless, like most DNNs, careful selection of the optimizer's hyperparameters, such as the learning rate, is essential. Without a validation set, the training loss curve can serve as a rudimentary guide to assess if the loss is being minimized effectively. Still, better results typically stem from hyperparameter tuning using a validation set. Next to popular regularization techniques, e.g., dropout and weight decay, such a validation set is also essential to detect and avoid overfitting. Finally, a separate test set allows us to assess MaDL's final performance reliably. The setting of error-prone annotators makes obtaining a validation or test set challenging. Ideally, we obtain the GT labels from experts. Alternatively, we may assume the majority votes of numerous annotators as GT labels. *Recommendation: For hyperparameter tuning, avoidance of overfitting, and a reliable performance assessment, consider acquiring a validation and test set before deploying MaDL.*

**Deployment:** After training, MaDL's GT and AP models offer various deployment options: We can

- directly apply the GT model as a downstream classifier to the associated learning task,
- use the GT model's predictions to improve the dataset's label quality,
- study the annotation patterns and correlations between annotators via the AP model,
- or leverage the AP model to assess which annotator is best for annotating a particular instance.

