# OpenReview forum: "Multi-annotator Deep Learning: A Probabilistic Framework for Classification"
_TMLR — Accepted by TMLR_

### Review · Reviewer_Jzj1 · 2023-05-01

**Summary Of Contributions:**

In this paper, the authors study Multi-annotator Deep Learning (MaDL) from a probabilistic perspective. Machine learning datasets today are usually crowd-sourced and the annotators might not be reliable. So it is important to make sure that the models can still learn from these ambiguous labels. The framework proposed in this paper learns a ground truth model (GT model) and an annotator performance model (AP model), in an attempt to maximize the likelihood of real ground truth labels. Evaluated on both real world crowdsourced datasets and simulated academic datasets, the authors demonstrate that MaDL outperforms all existing multi-annotator deep learning methods and partially closes the gap between fully supervised learning on real ground truth labels and aggregated crowdsourced labels.

**Audience:**

Yes

**Claims And Evidence:**

No

**Requested Changes:**

To get this paper accepted into TMLR, it is necessary to restructure the paper and make it more intuitive. Despite being a long submission, 24 pages is far too long and the authors need to consider what part of the main context can be moved to the appendix. Besides, the authors need to justify the importance of the MaDL task itself and the severe performance drop on very small scale datasets such as CIFAR 10.

**Strengths And Weaknesses:**

**Strengths:**

- The authors present a nice probabilistic framework for the multi-annotator deep learning problems and provide a good amount of discussion / analysis on what kind of properties people need to consider when developing MaDL algorithms (p1-p6);
- The mathematical derivations are very detailed and helpful for the readers to understand the paper;
- The experiment results outperform existing baselines under the same setting.

**Weaknesses:**

- The paper is extremely lengthy. A normal TMLR paper usually finishes in 12-15 pages but the paper is almost twice as long as a regular paper submission. I understood that the authors submit the paper as a "long submission", but I still believe that moving a good amount of detail to the appendix might be a better idea.
- It's still about the writing of the paper. I feel that the authors prefer to directly jump into mathematical formulations and derivations without providing intuition first. This will make the paper very hard to follow. In fact, I will suggest the authors to try summarizing what message each formula is supposed to deliver and express them in plain language before diving into the details. It is also very important to provide concrete numerical examples on how key formulae are calculated (e.g. Eqn 25). It's even better if the authors could provide an overview figure for Algorithm 1.
- I am curious about the necessity of the AP model in Figure 3. Is it possible for us to use a simple statistical model to measure the reliability of different annotators on different classes? For example, we can just use 1 - error_percentage as the entries for confusion matrices. I feel that the authors gave too little example on how this confusion matrix looks like in real world datasets and how it correlates with the simplest error percentage statistics.
- Regarding evaluation, it seems to me that the authors are using very small datasets to perform proof-of-concept experiments. While it is usually OK for most theory-oriented papers, it might be problematic for this particular paper. From my understanding, one of the most important motivations for MaDL is that deriving a "golden standard" ground truth label might be very costly. But if you conduct experiments on very small datasets, the cost will not be a problem. Besides, it is true that the performance drop of MaDL from UB is slower than other papers in the field, but the gap is still considerably large. Take CIFAR10 as an example, the performance almost dropped from 93.3% accuracy to 85% accuracy, which is the difference between a very simple 3-5 layer CNN and a relatively strong, ResNet-like design. This also makes me question whether the proposed MaDL framework could have any practical value in the current stage.

---

> ### Author Response · Authors · 2023-07-12
> **Respone to Reviewer Jzj1 (Part 1 of 2)**
>
> Dear Reviewer Jzj1,
>
> We are grateful for your time and effort in reviewing our paper. We aim to address your identified shortcomings and implement the suggested changes. For better orientation, we have marked your comments in **bold** and our changes/additions within the paper in red.
>
> **The paper is extremely lengthy. A normal TMLR paper usually finishes in 12-15 pages but the paper is almost twice as long as a regular paper submission. I understood that the authors submit the paper as a "long submission", but I still believe that moving a good amount of detail to the appendix might be a better idea.**
>
> We agree on the substantial length of our paper, primarily attributed to our thorough related work, comprehensive methodology explanation, and extensive evaluations addressing various research questions. Nevertheless, we have already condensed the paper's main body by relocating proofs to the appendix. If required, we could also move parts of the tables to the appendix, such as results for specific datasets. Potentially, an entire research question could also be moved to the appendix. However, we believe the key takeaways provide a succinct overview of the results, allowing readers to skim certain parts of the evaluation if necessary. Please let us know if we should make any such or other shortenings of the main body of the paper.
>
> **It's still about the writing of the paper. I feel that the authors prefer to directly jump into mathematical formulations and derivations without providing intuition first. This will make the paper very hard to follow. In fact, I will suggest the authors to try summarizing what message each formula is supposed to deliver and express them in plain language before diving into the details. It is also very important to provide concrete numerical examples on how key formulae are calculated (e.g. Eqn 25). It's even better if the authors could provide an overview figure for Algorithm 1.**
>
> For a better understanding, we added a concrete numerical example of how our proposed algorithm works (cf. Appendix B). For this purpose, we exemplify each step of the algorithm individually. Furthermore, our provided code and notebooks allow readers to gain more insights about MaDL by testing and visualizing MaDL's predictions for different hyperparameter configurations and datasets.
>
> **I am curious about the necessity of the AP model in Figure 3. Is it possible for us to use a simple statistical model to measure the reliability of different annotators on different classes? For example, we can just use 1 - error_percentage as the entries for confusion matrices. I feel that the authors gave too little example on how this confusion matrix looks like in real world datasets and how it correlates with the simplest error percentage statistics.**
>
> In the context of RQ1, we have tested the need for the complex AP model. We think the simple statistical model described by you corresponds to the variant $\text{MaDL}(\overline{X}, I)$, which estimates an instance- and class-independent correctness probability per annotator. The GT-ACC results for two of the four datasets with simulated annotators are competitive with $\text{MaDL}(X, F)$ as the most complex AP model. However, on the datasets with real-world annotators, there is a considerable performance drop when using the simple AP model. Similar observations hold when we estimate instance-dependent and class-independent annotation correctness probabilities (cf. performances of $\text{MaDL}(X, I)$). Further, we added illustrative examples to the new practitioner's guide to better understand the confusion matrices with varying degrees of freedom (cf. Appendix G).

---

> ### Author Response · Authors · 2023-07-12
> **Respone to Reviewer Jzj1 (Part 2 of 2)**
>
> **Regarding evaluation, it seems to me that the authors are using very small datasets to perform proof-of-concept experiments. While it is usually OK for most theory-oriented papers, it might be problematic for this particular paper. From my understanding, one of the most important motivations for MaDL is that deriving a "golden standard" ground truth label might be very costly. But if you conduct experiments on very small datasets, the cost will not be a problem. Besides, it is true that the performance drop of MaDL from UB is slower than other papers in the field, but the gap is still considerably large. Take CIFAR10 as an example, the performance almost dropped from 93.3% accuracy to 85% accuracy, which is the difference between a very simple 3-5 layer CNN and a relatively strong, ResNet-like design. This also makes me question whether the proposed MaDL framework could have any practical value in the current stage.**
>
> We generally based our data selection for evaluation on related and well-known works in the field [1, 2]. However, we agree it would be interesting to investigate the results on more complex datasets. For this purpose, we added a case study, which presents the empirical results of MaDL and the other techniques for CIFAR100 with the wide residual network WRN-28-10. Overall, the state-of-the-art performance of MaDL compared to related techniques is confirmed even for datasets with significantly more classes (previously, the LETTER dataset had the most classes with 26).
> From a practical point of view, a comparison with the UB is of limited value. The performance of MaDL and the other techniques depends to a large extent on the distribution and ratio of erroneous annotations. For example, if we have a data set with 80% erroneous annotations, it is challenging to approach the accuracy of the UB. This applies not only to learning techniques with multiple error-prone annotators but also to training DNNs with erroneous annotations in general [3]. To illustrate the effect of noise, we tested different annotation ratios for CIFAR10 (cf. Appendix E). With an increasing annotation ratio, the accuracy of the annotations aggregated via the majority rule increases, and thus, MaDL approaches the performance of the UB.
> Furthermore, we emphasize that the annotators were simulated for CIFAR10. For another simulation (e.g., with better performances of the annotators), MaDL's results would also be significantly closer to the UB's results. A comparison with the LB, on the other hand, often provides more practical insights since aggregating class labels via a majority rule provides a lower bound to be exceeded by more sophisticated techniques. Furthermore, the LB allows assessing the difficulty of a dataset. For example, if we compare the performances between the LB and UB for the datasets in Table 5, we see that the difference is most prominent for CIFAR10. Accordingly, this is an indicator of a more challenging learning task.
> Regarding the cost issue and practical relevance, we remark that GT labels cannot be requested for every dataset, e.g., crowdsourced datasets. In this case, expensive experts can be recruited, or many annotators can be requested for each instance. If we look at the case study on CIFAR10 as an example, we see that even with an annotation rate of 0.8 (which implies eight annotations on average per instance), GT labels cannot be inferred for all instances. Furthermore, even for this high annotation rate, MaDL performs significantly better than LB and thus demonstrates its practical value.
>
> [1] Filipe Rodrigues and Francisco Pereira. Deep Learning from Crowds. In AAAI Conf. Artif. Intell., pp. 1611–1618, New Orleans, LA, 2018.
>
> [2] Zhendong Chu, Jing Ma, and Hongning Wang. Learning from Crowds by Modeling Common Confusions. In AAAI Conf. Artif. Intell., pp. 5832–5840, Virtual Conf., 2021.
>
> [3] Li, Junnan, Richard Socher, and Steven CH Hoi. DivideMix: Learning with Noisy Labels as Semi-supervised Learning. In Int. Conf. Learn. Representations., Virtual Conf., 2020.
>
> Feel free to contact us any time if you have further questions, change requests, or remarks regarding our response! We are pleased and grateful for any further feedback!

---

### Review · Reviewer_YPct · 2023-05-29

**Summary Of Contributions:**

The authors investigate multi-annotator deep learning (assuming noisy class labels given that the annotators could produce errors due to various reasons like expertise and carelessness) by proposing the approach where a ground-truth model (predicting the correct label of the classification problem) and an annotator performance model are jointly trained end-to-end using a weighted maximum-likelihood approach.

The training in prior work usually uses an EM approach, where the ground-truth labels are estimated during the expectation step, and the ground-truth model and annotator performance models’ parameters are optimized during the minimization step. Variational inference and expectation propagation are also used in past approaches. The authors’ approach satisfies all six proposed criteria (for the annotator performance model), but prior approaches don’t. The authors’ approach relies on two models as well: the ground-truth model and the annotator performance model. The authors only consider classification, so the two models can be modeled as two categorical distributions as shown in Equations (6) and (7).

Figure 3 (on page 8 in the submission version) illustrates the model architecture. The architecture is very clear. The ground-truth model takes in the input (of the example to be classified). The annotator performance model takes in one or optionally two things: the first thing being the annotator embedding; the second thing (optionally) being the instance embedding (of the given example – either using the original instance feature or using the learned instance representation from the ground-truth model). The output of the ground-truth model and the output of the annotator performance model are then multiplied (by dot product) into the final annotation probability.

Experiments are done on seven datasets: Toy, Letter, Labelme, Msic, FMNIST, CIFAR10, SVHN. Two (LabelMe, Music) datasets are using real-world annotators. The rest of the datasets are using simulated annotators.

Three research questions:

(1) Do class-dependent and instance-dependent modeled annotator performances improve learning? The answer is yes. Performances of ground-truth and annotator performance models are improved.

(2) Does modeling correlations between annotators (who are potentially spamming) improve learning? The answer is yes, when there are many correlated spamming annotators. But no (at least not consistently), when capturing correlations of beneficial annotators.

(3) Do annotator features containing prior information improve learning and enable inductively learning annotators’ performances? The answer is yes.


**Audience:**

Yes

**Broader Impact Concerns:**

The statement is present and well-written.

**Claims And Evidence:**

Yes

**Requested Changes:**

- Answer the above questions.
- Include more information on the applicability of the approach (see above).

**Strengths And Weaknesses:**

I learned a lot from the related work section. The related work in multi-annotator deep learning is split into six categories: class-dependent annotator performance, instance-dependent annotator performance, annotator correlations, robustness to spamming annotators, annotator prior information, and inductive learning of annotator performance. Table 1 is quite helpful.

I really appreciate the “takeaway” blobs right under the research questions.

The paper is generally sound and detailed. The research questions and takeaways are clear.

---

A general comment: What’s the takeaway for general researchers or practitioners? What circumstances would the approach work well / would the approach be more useful in the real world?
- For example, what if the number of classes is very large?
- What if the number of annotators is very small, or very large?
- Would the approach be more or less useful when there are a lot more examples in the dataset? What if there are only a small number of examples in the dataset?
- Would the approach extend onto sequence generation tasks (e.g., text generation, where at each time-step, we choose one word out of the vocabulary of, say, 10k possibilities)? If not, what are the difficulties?

An unclear assumption in Section 2: “We further assume that there is a subset of annotators whose annotated instances are sufficient to approximate the ground-truth label distribution.” It’s not clear to me what the authors mean by “sufficient” – does the sufficiency depend on the class of algorithms?

Potential issue on the derivation of Proposition 2: how did the authors go from the third-to-last line [ delta ( \sum_y p(y|x) p(y|x,a,y) < \sum p(y|x) (1 - p(y|x,a,y)) ) ] to the second-to-last line [ delta ( \sum_y p(y|x) p(y|x,a,y) < 1 - \sum_y p(y|x) p(y|x,a,y) ) ]? If p(y | x) p(y | x, a, y) is between the right-hand-side (of the inequality) of the third-to-last line and the right-hand-side of the second-to-last line, would the equality still be true? Please let me know if I’m missing something.

In Equation 14 and Equation 15, what is e?

---

> ### Author Response · Authors · 2023-07-12
> **Response to Reviewer YPct (Part 1 of 2)**
>
> Dear Reviewer YPct,
>
> We thank you for taking the time and effort to read our paper and to create a thorough review. We aim to address all your questions and requested changes in the following. For better orientation, we highlight your comments in **bold** and mark our changes/additions in the paper in red.
>
> **What's the takeaway for general researchers or practitioners? What circumstances would the approach work well / would the approach be more useful in the real world?**
>
> We have added the takeaways and recommendations for practitioners in a corresponding guide (cf. Appendix G). This guide overviews the practical aspects you and other reviewers identified.
>
> **For example, what if the number of classes is very large?**
>
> We address this aspect in the new practitioner's guide (cf. Appendix G) and a new case study (cf. Appendix F). Specifically, this case study shows the empirical results of MaDL and the other techniques for CIFAR100 with the wide residual network WRN-28-10. Overall, the state-of-the-art performance of MaDL compared to related techniques is confirmed even for datasets with significantly more classes (previously, the LETTER dataset had the most classes with 26). In the guide, we also show how to control the complexity of the confusion matrices and, thus, the computational costs to apply MaDL to problems with significantly more classes.
>
> **What if the number of annotators is very small, or very large?**
> We address this aspect in the practitioner's guide (cf. Appendix G). As the number of annotators increases, so does (usually) the number of different annotation patterns to be learned. To ensure that the AP model still learns these annotation patterns, one could increase the dimensionality of the annotator embeddings.
>
> **Would the approach be more or less useful when there are a lot more examples in the dataset? What if there are only a small number of examples in the dataset?**
>
> We address this aspect by showing two types of learning curves. A case study on CIFAR10 (cf. Appendix E) shows how MaDL performs when we vary the annotation ratio, i.e., the average number of annotations per instance. For example, an annotation ratio of 0.2 for ten annotators implies (on average) two annotations per instance. The results of this case study demonstrate that more annotations per instance (higher annotation ratios) lead to better results. Furthermore, MaDL mostly yields superior results for the tested annotation ratios compared to other techniques. As part of the practitioner's guide (cf. Appendix G), we present learning curves as a function of the number of annotated instances on the dataset LETTER. The results indicate that more annotated instances strongly improve MaDL's performances. Further, they confirm the benefit of using MaDL over naive techniques such as the majority rule for a rather low number of annotated instances.
>
> **Would the approach extend onto sequence generation tasks (e.g., text generation, where at each time-step, we choose one word out of the vocabulary of, say, 10k possibilities)? If not, what are the difficulties?**
>
> We added a paragraph in our paper's conclusion (cf. Section 6) to detail the extension of MaDL to potential other supervised learning tasks, such as semantic segmentation, sequence classification, and regression. Both classification tasks require adjustments to the architecture of the GT and AP models, while regression tasks require adjustments to the distributional assumptions in the probabilistic model. As for your question, we thus expect that MaDL can be extended to the classification of text sequences. However, a general extension to generative tasks is not simply possible from our point of view. This is because one typically needs objective GT labels. For example, as we understand it, we cannot say which words must follow each other. Thus, estimating whether an annotator is right or wrong would be tricky.
>
> **An unclear assumption in Section 2: "We further assume that there is a subset of annotators whose annotated instances are sufficient to approximate the ground-truth label distribution." It's not clear to me what the authors mean by "sufficient" – does the sufficiency depend on the class of algorithms?**
>
> This is an important hint for us. We have now supplemented this passage with the addition that these annotated instances allow a correct distinction between all classes. A technical formalization is impossible for us at this point since the theoretical requirements for learning with noisy annotations with a wide variety of noise distributions are still the subject of current research. Accordingly, we have added this limitation to the paper's conclusion (cf. Section 6) and formulated it as an open research task for the future. This also includes deriving theoretical performance guarantees of MaDL.

---

> ### Author Response · Authors · 2023-07-12
> **Response to Reviewer YPct (Part 2 of 2)**
>
> **Potential issue on the derivation of Proposition 2: how did the authors go from the third-to-last line [ delta ( \sum_y p(y|x) p(y|x,a,y) < \sum p(y|x) (1 - p(y|x,a,y)) ) ] to the second-to-last line [ delta ( \sum_y p(y|x) p(y|x,a,y) < 1 - \sum_y p(y|x) p(y|x,a,y) ) ]? If p(y | x) p(y | x, a, y) is between the right-hand-side (of the inequality) of the third-to-last line and the right-hand-side of the second-to-last line, would the equality still be true? Please let me know if I'm missing something.**
>
> In the step from the third-to-last line to the second-to-last line, we resolved the parenthesis in $\sum_y \Pr(y|\mathbf{x}) (1 - \Pr(y|\mathbf{x}, \mathbf{a}, y))$ to $\sum_y \Pr(y|\mathbf{x}) - \sum_y \Pr(y|\mathbf{x})\Pr(y|\mathbf{x}, \mathbf{a}, y)$. Due to the normalization of $\Pr(y|\mathbf{x})$ over the class labels $y$, we get $\sum_y \Pr(y|x) = 1$. We have included this intermediate step in the proof, which is now part of Appendix A, to shorten the paper's main content (according to the request of another reviewer).
>
> **In Equation 14 and Equation 15, what is e?**
>
> $\mathbf{e}\_{z\_{nm}}$ denotes the one-hot encoded vector of annotation $z_{nm}$. In the context of annotator features, we introduced this notation of one-hot encoded vectors in the problem setting (cf. Section 2). Since this definition can be easily overlooked, we define it now after Eq. 14 in the context of annotations.
>
> Feel free to contact us any time if you have further questions, change requests, or remarks regarding our response! We are pleased and grateful for any further feedback!

---

### Review · Reviewer_HnGV · 2023-06-29

**Summary Of Contributions:**

The author proposed a probabilistic training framework named multi-annotator deep learning (MaDL) which contains one model to predict the ground truth (GT) labels and one model to estimate annotator performance (AP). The framework could be applied to complex classification tasks requiring crowdsourcing, and multiple annotators label each instance. The AP model provides predictions on six properties including class-dependent, instance-dependent annotation performance, annotator correlations, robustness to spamming, etc. The training procedure is end-to-end. The authors also tested the efficacy of this framework on public classification datasets including images, audio, and tabular data.

**Audience:**

Yes

**Broader Impact Concerns:**

N.A.

**Claims And Evidence:**

Yes

**Requested Changes:**


-	Elaborate on what features are suitable or feasible to obtain for both instances and annotators
-	Explain whether the overfitting could be an issue in the test cases
-	Extend the studies on applications of MaDT to improve labeled dataset quality and downstream classifier training
-	Discuss how the framework could be extended to complex annotations such as high dimensional classification, keypoint, bounding boxes, segmentation, and so on

**Strengths And Weaknesses:**

The strengths of this work include:

-	The paper is well-written, and the figures are clear and easy-to-understand
-	Section 3 provides a comprehensive review of the existing multi-annotator supervised learning techniques including GT and AP models.  Table 1 is impressive as it summarizes the past works’ predictability on the six annotator properties
-	The authors also benchmarked the framework’s performance on public datasets against other approaches such as majority vote, ground truth, union set and etc. MaDL performance well in terms of accuracy, balanced accuracy, and negative log-likelihood
-	The authors provide detailed mathematical equations for each new concept introduced ranging from metrics to loss function, which makes the paper theoretically sound

The weaknesses of this work include:

-	Both instance and annotator features are crucial pieces in the framework. As the author stated, annotator features should contain prior knowledge. However, the authors did not elaborate on how the features are generated or obtained in the case studies. In a real-life annotation case, we could measure annotation time, and potentially collect annotator age or interest from surveys (annotator features in Figure 1). Still, some features could be hard to obtain due to privacy or resource constraints.
-	Certain datasets have a smaller amount of instances and a larger number of classes (for instance, LABELME). Could overfitting be an issue for those use cases? The authors should further elaborate.
-	The authors answered a few research questions on how to improve the AP or GT model. But how can their predictions be used in removing low-quality labels and training downstream classifiers? The paper barely touches on the applications of AP/GT models. It seems unclear how the framework could help boost the performance of downstream models.
-	The scope of this paper is limited to classification tasks in a relatively low dimension (<30 classes). How could the framework scale to high dimensional classification problems (> 100 classes)? In addition, in real life, more tasks using crowd workers are on complex annotations such as keypoints, bounding boxes, and image segmentation.

---

> ### Author Response · Authors · 2023-07-12
> **Response to Reviewer HnGV (Part 1 of 2)**
>
> Dear Reviewer HnGV,
>
> We appreciate your time and commitment to reading our paper and providing an insightful review. We aim to address each of your identified weaknesses and requested changes in the following. For better orientation, we highlight your comments in **bold** and mark our changes/additions in the paper in red.
>
> **Both instance and annotator features are crucial pieces in the framework. As the author stated, annotator features should contain prior knowledge. However, the authors did not elaborate on how the features are generated or obtained in the case studies. In a real-life annotation case, we could measure annotation time, and potentially collect annotator age or interest from surveys (annotator features in Figure 1). Still, some features could be hard to obtain due to privacy or resource constraints. Elaborate on what features are suitable or feasible to obtain for both instances and annotators.**
>
> The instance and annotator features are indeed essential aspects of our framework. However, MaDL does not make any specific assumptions regarding the characteristics or modality of the instances. In a new practitioner's guide (cf. Appendix G), we added a passage highlighting this and explaining how to learn with different modalities, e.g., tabular, image, or text data. We can combine MaDL with varying architectures from the literature depending on the modality.
> For annotator features, we introduce two variants within the problem setting (cf. Section 2). On the one hand, we can use features with background information about the annotators for learning. The broader impact statement discusses the privacy issue of acquiring such data. Since we could not access a dataset with background information, we artificially generated it using information about an annotator's simulation. A more detailed explanation, including visualization, can be found in the setup of RQ3 and Fig. 8. On the other hand, MaDL can also be trained entirely without this background information about the annotators. As described in the problem setting (cf. Section 2) and now also in the guide (cf. Appendix G) and an example of the MaDL's training algorithm (cf. Appendix B), we encode the anonymous identity of each annotator by a one-hot-encoded vector. Given three annotators, we would then have the vectors $(1, 0, 0)^\mathrm{T}, (0, 1, 0)^\mathrm{T}$, and $(0, 0, 1)^\mathrm{T}$. The dimensionality of the vectors depends on the number of annotators. We used these one-hot encoded vectors in almost all experiments and RQs. The only exception is RQ3, where we explicitly investigate the benefit of annotator features containing background information about the annotators.
>
> **Certain datasets have a smaller amount of instances and a larger number of classes (for instance, LABELME). Could overfitting be an issue for those use cases? The authors should further elaborate. Explain whether the overfitting could be an issue in the test cases.**
>
> We now explicitly address the critical aspect of overfitting in a paragraph of the practitioner's guide (cf. Appendix G). Generally, MaDL shares the risk of overfitting with typical DNNs. However, we can mitigate such overfitting using standard regularization techniques from the literature, such as weight decay and dropout. We also recommend gathering a small validation and test dataset, if possible. The validation dataset allows tuning hyperparameters and detecting overfitting, while the test dataset evaluates MaDL's final performance before deployment. Further, we exemplary show via learning curves on the LETTER dataset that we can train MaDL on datasets with very few annotated instances.
>
> **The authors answered a few research questions on how to improve the AP or GT model. But how can their predictions be used in removing low-quality labels and training downstream classifiers? The paper barely touches on the applications of AP/GT models. It seems unclear how the framework could help boost the performance of downstream models. Extend the studies on applications of MaDT to improve labeled dataset quality and downstream classifier training.**
>
> This aspect may need to be clarified. Throughout the paper, we evaluate the GT and AP models on separate test sets, as our focus is on downstream models. For example, we can directly apply MaDL's GT model as a downstream classifier to a learning task. For improved clarity, we have now included the keyword downstream in the abstract and in the definition of the GT model. In addition, we have further listed other deployment options of MaDL's GT and AP model in the practitioner's guide (cf. Appendix G). This list also includes the option to improve the label quality on the training data. For this purpose, we can use the GT model's predictions as proxies of the training instances' GT labels. Again, we note that such a step is not required since we can directly employ GT models as downstream classifiers. In case we misunderstood your question, please let us know.

---

> ### Author Response · Authors · 2023-07-12
> **Response to Reviewer HnGV (Part 2 of 2)**
>
> **The scope of this paper is limited to classification tasks in a relatively low dimension (<30 classes). How could the framework scale to high dimensional classification problems (> 100 classes)? In addition, in real life, more tasks using crowd workers are on complex annotations such as keypoints, bounding boxes, and image segmentation. Discuss how the framework could be extended to complex annotations such as high dimensional classification, keypoint, bounding boxes, segmentation, and so on.**
>
> We address datasets with more classes in the new practitioner's guide (cf. Appendix G) and in a new case study (cf. Appendix F). This case study shows the empirical results of MaDL and the other techniques for CIFAR100 with the wide residual network WRN-28-10. Overall, the state-of-the-art performance of MaDL compared to related techniques is confirmed even for datasets with significantly more classes (previously, the LETTER dataset had the most classes with 26). In the guide, we also illustrate how to control the complexity of the confusion matrices and, thus, the computational costs to apply MaDL to problems with significantly more classes. Furthermore, we added a paragraph in our paper's conclusion (cf. Section 6) to detail the extension of MaDL to potential other supervised learning tasks, such as semantic segmentation, sequence classification, and regression tasks. Both classification tasks require adjustments to the architecture of the GT and AP models, while regression tasks require adjustments to the distributional assumptions in the probabilistic model.
>
> Feel free to contact us any time if you have further questions, change requests, or remarks regarding our response! We are pleased and grateful for any further feedback!

---

### Decision · Action_Editors · 2023-08-28

**Recommendation:** Accept as is

**Comment:**

This paper presents a method for explicitly modeling the quality of individual labelers, alongside the training of deep networks, to deal with per-annotator noise.  The authors use this to incorporate the probability of the annotation, along in the predictive likelihood of the label, in the loss.  In the paper, the authors experiment with learning the annotator model using one-hot labels for the annotators or explicit annotator features.  This is a rather long submission for TMLR and the reviewers seemed to find it somewhat dense.  Reviewers found the paper well-written, interesting, and sound.  One reviewer commented that the related work section is quite thorough and educational.  As weaknesses, the reviewers pointed out that they had trouble understanding how the authors proposed to use annotator features, and that the scope of the paper was limited to relatively smaller classification tasks.  The reviewers noted in discussion that the author rebuttal was strong and convincing, and helped to answer their concerns.  Ultimately, all reviewers voted to accept the paper, with two leaning accept and one accept.  Given that the reviewers all agreed on acceptance, found the paper relevant and interesting, and agreed that the claims were substantiated, the recommendation is to accept the paper.

**Audience:**

Noisy labels, arising from differing interpretation, performance, ability, etc. of the annotators seems like a prevalent issue in the practical application of deep learning.  Therefore it would seem that there is a potentially large audience for this kind of work.

**Claims And Evidence:**

The reviewers all found that the claims are sufficiently supported by evidence in the paper.

---

> ### Author Response · Authors · 2023-09-05
> **Camera-ready version**
>
> Dear Action Editor,
>
> We've uploaded the camera-ready version of our paper. If there are any further requirements, please do not hesitate to inform us. We thank the anonymous reviewers and you for the effort and the insightful comments, which enhanced our work.
>
> Best regards,
> Authors